



# Norwegian Sea net community production estimated from $O_2$ and prototype $CO_2$ optode measurements on a Seaglider

Luca Possenti[1], Ingunn Skjelvan[2], Dariia Atamanchuk[3], Anders Tengberg[4], Matthew P. Humphreys[5], Socratis Loucaides[6], Liam Fernand[7], Jan Kaiser[1]

[1]Centre for Ocean and Atmospheric Sciences, School of Environmental Sciences, University of East Anglia, Norwich, UK

[2]NORCE Norwegian Research Centre, Bjerknes Centre for Climate Research, Bergen, Norway

[3]Dalhousie University, Halifax, Canada

[4]University of Gothenburg, Sweden

[5]NIOZ Royal Netherlands Institute for Sea Research, Department of Ocean Systems (OCS), and Utrecht University, Texel, the Netherlands

[6]National Oceanography Centre, European Way, Southampton, SO14 3ZH, UK

[7]Centre for Environment, Fisheries and Aquaculture Sciences, Lowestoft, UK, NR33 0HT

*Correspondence to:* Luca Possenti (L.Possenti@uea.ac.uk)

**Abstract.** We report on a pilot study using a $CO_2$ optode deployed on a Seaglider in the Norwegian Sea for 8 months (March to October 2014). The optode measurements required drift- and lag-correction, and in situ calibration using discrete water samples collected in the vicinity. We found the optode signal correlated better with the concentration of $CO_2$, $c(CO_2)$, than with its partial pressure, $p(CO_2)$. Using the calibrated $c(CO_2)$ and a regional parameterisation of total alkalinity ($A_T$) as a function of temperature and salinity, we calculated total dissolved inorganic carbon concentrations, $C_T$, which had a standard deviation of 10 µmol kg$^{-1}$ compared with direct $C_T$ measurements. The glider was also equipped with an oxygen ($O_2$) optode. The $O_2$ optode was drift-corrected and calibrated using a $c(O_2)$ climatology for deep samples ($R^2 = 0.89$; RMSE = 0.009 µmol kg$^{-1}$). The calibrated data enabled the calculation of $C_T$- and oxygen-based net community production, $N(C_T)$ and $N(O_2)$. To derive $N$, $C_T$ and $O_2$ inventory changes over time were combined with estimates of air-sea gas exchange and entrainment of deeper waters. Glider-based observations captured two periods of increased Chl $a$ inventory in late spring (May) and a second one in summer (June). For the May period, we found $N(C_T) = (24\pm5)$ mmol m$^{-2}$ d$^{-1}$, $N(O_2) = (61\pm14)$ mmol m$^{-2}$ d$^{-1}$ and an (uncalibrated) Chl $a$ peak concentration of $c_{raw}$(Chl $a$) = 3 mg m$^{-3}$. During the June period, $c_{raw}$(Chl $a$) increased to a summer maximum of 4 mg m$^{-3}$, which drove $N(C_T)$ to $(64\pm67)$ mmol m$^{-2}$ d$^{-1}$ and $N(O_2)$ to $(166\pm75)$ mmol m$^{-2}$ d$^{-1}$. The high-resolution dataset allowed for quantification of the changes in $N$ before, during and after the periods of increased Chl $a$ inventory. After the May period, the remineralisation of the material produced during the period of increased Chl $a$ inventory decreased $N(C_T)$ to (-80$\pm$107) mmol m$^{-2}$ d$^{-1}$ and $N(O_2)$ to $(-15\pm27)$ mmol m$^{-2}$ d$^{-1}$. The survey area was a source of $O_2$ and a sink of $CO_2$ for most of the summer. The deployment captured two different surface waters: the Norwegian Atlantic





Current (NwAC) and the Norwegian Coastal Current (NCC). The NCC was characterised by lower $c(O_2)$ and $C_T$
than the NwAC, as well as lower $N(O_2)$, $N(C_T)$ and $c_{raw}(\text{Chl } a)$. Our results show the potential of glider data to
simultaneously capture time and depth-resolved variability in $C_T$ and $O_2$.
**1 Introduction**
Climate models project an increase in the atmospheric $CO_2$ mole fraction driven by anthropogenic emissions
from a preindustrial value of 280 µmol mol$^{-1}$ (Neftel et al., 1982) to 538-936 µmol mol$^{-1}$ by 2100 (Pachauri and
Reisinger, 2007). The ocean is known to be a major $CO_2$ sink (Sabine et al., 2004; Le Quéré et al., 2009; Sutton
et al., 2014), in fact has taken up approximately 25 % of this anthropogenic $CO_2$ with a rate of (2.5±0.6) Gt a$^{-1}$
(in C equivalents) (Friedlingstein et al., 2019). This uptake alters the carbonate system of seawater and is
causing a decrease in seawater pH, a process known as ocean acidification (Gattuso and Hansson, 2011). The
processes affecting the marine carbonate system include air-sea gas exchange, photosynthesis and respiration,
transport and vertical and horizontal mixing, and $CaCO_3$ formation and dissolution. For that reason, it is
important to develop precise, accurate and cost-effective tools to observe $CO_2$ variability and related processes
in the ocean. Provided that suitable sensors are available, autonomous ocean glider measurements may help
resolve these processes.
To quantify the marine carbonate system, four variables are commonly measured: total dissolved inorganic
carbon concentration ($C_T$), pH, total alkalinity ($A_T$) and the fugacity of $CO_2$ ($f(CO_2)$). At thermodynamic
equilibrium, knowledge of two of the four variables is sufficient to calculate the other two. Marine carbonate
system variables are primarily measured on research ships, commercial ships of opportunity, moorings, buoys
and floats (Hardman-Mountford et al., 2008; Monteiro et al., 2009; Takahashi et al., 2009; Olsen et al., 2016;
Bushinsky et al., 2019). Moorings equipped with submersible sensors often provide limited vertical and
horizontal, but good long-term temporal resolution (Hemsley, 2015). In contrast, ship-based surveys have higher
vertical and spatial resolution than moorings but limited repetition frequency because of the expense of ship
operations. Ocean gliders have the potential to replace some ship surveys because they are much cheaper to
operate and will increase our coastal and regional observational capacity. However, the slow glider speed of 1-2
km h$^{-1}$ only allows a smaller spatial coverage than ship surveys and the sensors require careful calibration to
match the quality of data provided by ship-based sampling.
Carbonate system sensors suitable for autonomous deployment have been developed in the past decades, in
particular pH sensors (Martz et al., 2010; Rérolle et al., 2013; Seidel et al., 2008) and $p(CO_2)$ sensors (Goyet et
al., 1992; Degrandpre, 1993; Körtzinger et al., 1996; Bittig et al., 2012; Atamanchuk, 2013). One of these





sensors is the $CO_2$ optode (Atamanchuk et al., 2014) which has been successfully deployed to monitor an
artificial $CO_2$ leak on the Scottish west coast (Atamanchuk, et al., 2015b), on a cabled underwater observatory
(Atamanchuk, et al., 2015a), to measure lake metabolism (Peeters et al., 2016), for fish transportation (Thomas
et al., 2017) and on a moored profiler (Chu et al., 2020).
Oxygen and $C_T$ can be used to calculate net community production ($N$), which is defined as the difference
between gross primary production ($G$) and community respiration ($R$). At steady-state, $N$ is equal to the rate of
organic carbon export and transfer from the surface into the mesopelagic and deep waters (Lockwood et al.,
2012). $N$ is derived by vertical integration to a specific depth, that is commonly defined relative to the mixed
layer depth ($z_{mix}$) or the bottom of the euphotic zone (Plant et al., 2016). A system is defined as autotrophic
when $G$ is larger than $R$ (i.e. $N$ is positive) and as heterotrophic when $R$ is larger than $G$ (i.e. $N$ is negative)
(Ducklow and Doney, 2013).
$N$ can be quantified using bottle incubations, isotope methods ($^{14}C$, $^{15}N$, $^{16}O/^{17}O/^{18}O$) (Sharples et al., 2006;
Quay, et al, 2012; Seguro et al., 2019) or in situ biogeochemical budgets. Bottle incubations involve measuring
oxygen concentration driven by production and respiration in vitro under dark and light conditions.
Biogeochemical budgets combine $O_2$ and $C_T$ inventory changes with estimates of air-sea gas exchange,
entrainment, advection and vertical mixing (Alkire et al., 2014; Binetti et al., 2020; Neuer et al., 2007).
The Norwegian Sea is a complex environment due to the interaction between the Atlantic Water (NwAC)
entering from the south-west, Arctic Water coming from north and the Norwegian Coastal Current (NCC)
flowing along the Norwegian coast (Nilsen and Falck, 2006). In particular, Atlantic Water enters the Norwegian
Sea through the Faroe-Shetland Channel and Iceland-Faroe Ridge (Hansen and Østerhus, 2000) with $S$ between
35.1 and 35.3 and temperatures warmer than 6 ˚C (Swift, 1986). Furthermore, the NCC water mass differs from
the NwAC with a surface $S < 35$ (Saetre and Ljoen, 1972) and a seasonal $\theta$ signal (Nilsen and Falck, 2006).
Biological production in the Norwegian Sea varies during the year and can be divided into 5 periods (Rey,
2001): (1) winter with the smallest productivity and phytoplankton biomass; (2) a pre-bloom period; (3) the
spring bloom when productivity increases and phytoplankton biomass reaches the annual maximum; (4) a post-
bloom period with productivity mostly based on regenerated nutrients; (5) autumn with smaller blooms than in
summer. Previous estimates of $N(C_T)$ were based on discrete $C_T$ samples (Falck and Anderson, 2005) or were
calculated from oxygen-based measurements and converted to C equivalents assuming Redfield stoichiometry
of production/respiration (Falck and Gade, 1999; Kivimäe, 2007; Skjelvan et al., 2001). Glider measurements
have been used to estimate $N$ in other ocean regions (Nicholson et al., 2008; Alkire et al., 2014; Haskell et al.,





2019; Binetti et al., 2020); however, as far as we know, this is the first study of net community production in the
Norwegian Sea using a high-resolution glider dataset (>$10^6$ data points; 40 s time resolution) and the first
anywhere estimating $N$ from a glider-mounted sensor directly measuring the marine carbonate system.
**2 Material and methods**
**2.1 Glider sampling**
Kongsberg Seaglider 564 was deployed in the Norwegian Sea on the 16 March 2014 at 63.00° N, 3.86° E and
recovered on the 30 October 2014 at 62.99° N, 3.89° E. The Seaglider was equipped with a prototype Aanderaa
4797 $CO_2$ optode, an Aanderaa 4330F oxygen optode (Tengberg et al., 2006), a Seabird CTD (GPCTD) and a
combined backscatter/chlorophyll $a$ fluorescence sensor (Wetlabs Eco Puck BB2FLVMT). The mean time
needed by the sensor to reach a stable value for an in situ measurement ($t$) varied with depth (Table 1). On
average in the top 100 m the CTD performed an in situ measurement every 24 s, the $O_2$ optode every 49 s, the
$CO_2$ optode every 106 s and the fluorescence sensor every 62 s. The time to perform an in situ measurement
increased in depths between 100 to 500 m to 31 s for the CTD, 153 s for the $O_2$ optode and 233 s for the $CO_2$
optode. This measurement time reached its maximum at depths between 500 to 1000 m where was 42 s for the
CTD, 378 s for the $O_2$ optode and 381 d for the $CO_2$ optode.
**Table 1.** Average time needed by the Seabird CTD (GPCTD), Aanderaa 4330F oxygen optode, Aanderaa 4797
$CO_2$ optode and a combined backscatter/chlorophyll $a$ fluorescence sensor (Wetlabs Eco Puck BB2FLVMT) to
perform an in situ measurement in the top 100 m, from 100 to 500 and from 500 to 1000 m.

| Depth / m | $t$(CTD) / s | $t$($O_2$) / s | $t$($CO_2$) / s | $t$(Chl $a$) / s |
|---|---|---|---|---|
| 0 – 100 m | 24 | 49 | 106 | 62 |
| 100 – 500 m | 31 | 153 | 233 | - |
| 500 – 1000 m | 42 | 378 | 381 | - |


The deployment followed the Svinøy trench, from the open sea towards the Norwegian coast. The glider
covered a 536 km long transect 8 times (4 times in each direction) for a total of 703 dives (Figure 1).

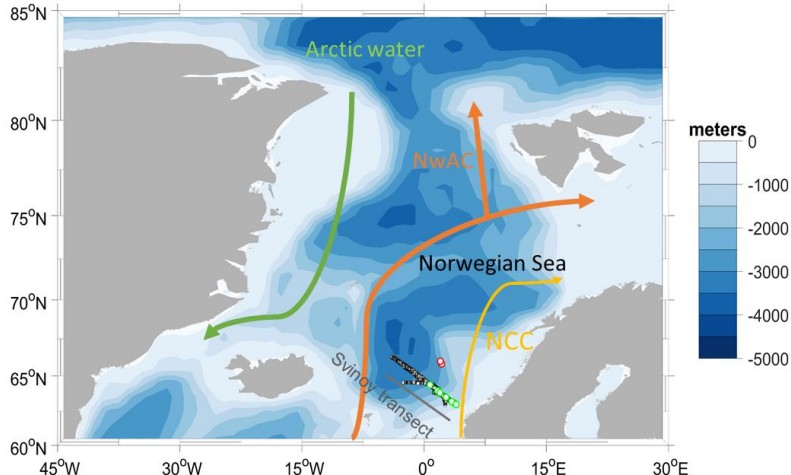

**Figure 1:** Map of the glider deployment and the main water masses. The black dots are the glider dives, the
green and the red dots are the water samples collected along the glider section and at OWSM, respectively. The
three main water masses (Skjelvan et al., 2008) are the Norwegian Coastal Current (yellow), the Norwegian
Atlantic Current (NwAC, orange) and Arctic Water (green).
**2.2 Discrete sampling**
During the glider deployment, 70 discrete water samples from various depths (5, 10, 20, 30, 50, 100, 300, 500
and 1000 m) were collected on 4 different cruises on the R/V Haakon Mosby along the southern half of the
glider transect on the 18 March, the 5 May, 6 and 14 June, and the 30 of October 2014. Samples for $C_T$ and $A_T$
were collected from 10 L Niskin bottles following the standard operational procedure (SOP) 1 of Dickson et al.
(2007). The $C_T$ and $A_T$ samples were preserved with saturated $HgCl_2$ solution (final $HgCl_2$ concentration: 15 mg
$dm^{-3}$). Nutrient samples from the same Niskin bottles were preserved with chloroform. $C_T$ and $A_T$ were analysed
on-shore according to SOP 2 and 3b (Dickson et al., 2007) using a VINDTA 3D (Marianda) with a CM5011
coulometer (UIC instruments) and a VINDTA 3S (Marianda), respectively. Nutrients were analysed on-shore
using a an Alpkem Auto Analyzer. In addition, 43 water samples were collected at Ocean Weather Station M
(OWSM) on 5 different cruises on the 22 March on R/V Haakon Mosby, the 9 May on R/V G.O. Sars, the 14
June on R/V Haakon Mosby, the 2 August and the 13 November on R/V Johan Hjort from 10, 30, 50, 100, 200,
500, 800 and 1000 m depth. The OWSM samples were preserved and analysed for $A_T$ and $C_T$ similar to the
Svinøy samples. No phosphate and silicate samples were collected from OSWM. Temperature ($\theta$) and salinity
($S$) profiles were measured at each station using a SeaBird 911 plus CTD. pH and $f(CO_2)$ were calculated using
the MATLAB toolbox CO2SYS (Van Heuven et al., 2011), with the following constants: $K_1$ and $K_2$ carbonic
acid dissociation constants of Lueker et al. (2000), $K(HSO_4^-/SO_4^{2-})$ bisulfate dissociation constant of Dickson





(1990) and borate to chlorinity ratio of Lee et al. (2010). In the OWSM calculations, we used nutrients collected
from the Svinøy section at a time as close as possible to the OWSM sampling as input. In the case of the glider,
we derived a parameterisation to derive phosphate and silicate concentration as a function of the discrete sample
depth and time. This parameterisation had an uncertainty of 1.3 and 0.13 µmol kg$^{-1}$, for silicate and phosphate
concentrations, respectively.
**2.3 Oxygen optode calibration**
The last oxygen optode calibration before the deployment was performed in 2012 as a two-point calibration at
9.91 °C in air-saturated water and at 20.37 °C in anoxic $Na_2SO_3$ solution. Oxygen optodes are known to be
affected by drift (Bittig et al., 2015), which is worse for the fast-response foils used in the 4330F optode for
glider deployments. It has been suggested to calibrate and drift correct the optode using discrete samples or in-
air measurements (Nicholson and Feen, 2017). Unfortunately, no discrete samples were collected at the glider
deployment or recovery.
To overcome this problem, we used archived data to correct for oxygen optode drift. These archived
concentration data (designated $c_C(O_2)$) were collected at OWSM between 2001 and 2007 (downloaded from
ICES data base) and from the deployment region between 2000 and 2018 (extracted from GLODAPv2; Olsen et
al., 2016). To apply the correction, we used the oxygen samples corresponding to a potential density $\sigma_0 > 1028$
kg m$^{-3}$ (corresponding to depths between 427 and 1000 m), because waters of these potential densities were
always well below the mixed layer and therefore subject to limited seasonal and interannual variability. The
salinity $S$ of these samples varied from 34.88 to 34.96, with a mean of 34.90±0.01; $\theta$ varied from 0.45 to –0.76
°C, with a mean of (–0.15±0.36) °C.
Figure 2 shows that the glider oxygen concentration ($c_G(O_2)$) where $\sigma_0 > 1028$ kg m$^{-3}$ was characterised by two
different water masses separated at a latitude of about 64° N. We used the samples collected north of 64° N to
derive the glider optode correction because this reflects the largest area covered by the glider. It was not possible
to use the southern region because it contained the archived samples from only 5 days. For each year-day with
archived samples, we calculated the median concentration of the glider and the archived samples. Figure 3
shows a plot of the ratio between $c_C(O_2)/c_G(O_2)$ against year-day and a linear fit, which is used to calibrate
$c_G(O_2)$ and correct for drift.





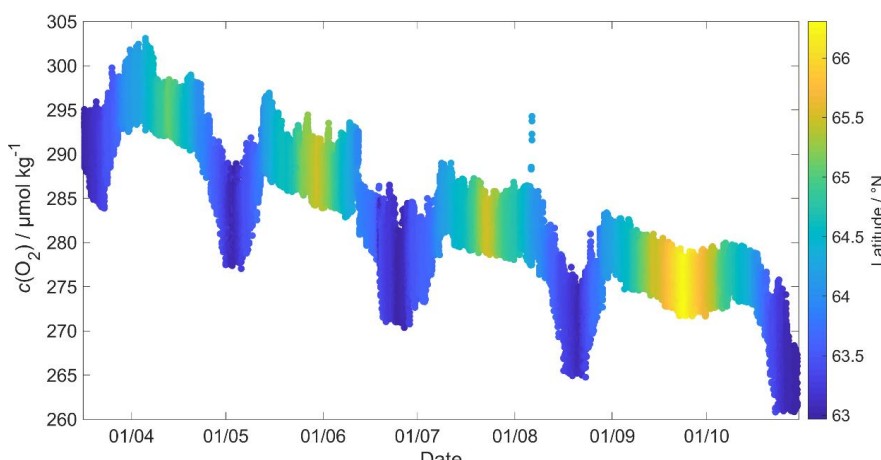


**Figure 2:** Glider oxygen concentration, $c_G(O_2)$, under $\sigma_0 = (1028\pm0.02)$ kg m$^{-3}$ coloured by latitude.


No lag correction was applied because the O$_2$ optode had a fast response foil and showed no detectable lag (<10
s), based on a comparison between descent and ascent profiles.

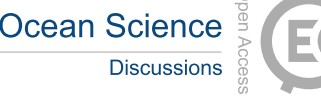
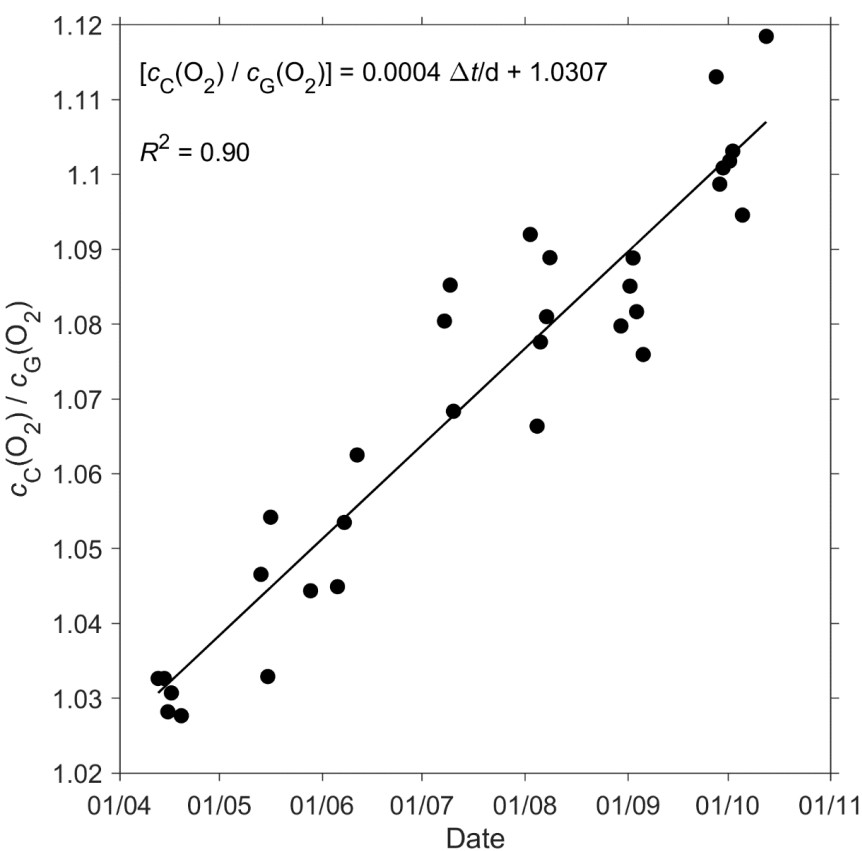

**Figure 3:**. The linear fit of the ratio between the daily median of the discrete oxygen samples ($c_C(O_2)$) and
glider oxygen data ($c_G(O_2)$) for $\sigma_0 > 1028$ kg m$^{-3}$ was used to derive the $c_G(O_2)$ drift and initial offset at
deployment. The time interval $\Delta t$ is calculated with respect to the deployment day of the 16th of March.

**2.4 CO$_2$ optode measurement principle**
The CO$_2$ optode consists of an optical and a temperature sensor incorporated into a pressure housing. The
optical sensor has a sensing foil comprising two fluorescence indicators (luminophores), of which one is
sensitive to pH changes and the other is not and thus used as a reference. The excitation and emission spectra of
the two fluorescence indicators overlap, but the reference indicator has a longer fluorescence lifetime than the
pH indicator. These two fluorescence lifetimes are combined using an approach known as Dual Lifetime
Referencing (DLR) (Klimant et al., 2001; von Bültzingslöwen et al., 2002). From the phase shift ($\varphi$), the partial
pressure of CO$_2$, $p(CO_2)$, is parameterised as an eight-degree polynomial (Atamanchuk et al., 2014):
$\log [p(CO_2)/\mu\text{atm}] = C_0 + C_1 \varphi + \ldots + C_8 \varphi^8$                                              (1)





where $C_0$ to $C_8$ are temperature-dependent coefficients.
The partial pressure of $CO_2$ is linked to the $CO_2$ concentration, $c(CO_2)$, and the fugacity of CO2, $f(CO_2)$, via the
following relationship:
$c(CO_2) = p(CO_2) / [1 − p(H_2O) / p]\ F(CO_2) = K_0(CO_2)\,f(CO_2)$                    (2)
where $F(CO_2)$ is the solubility function (Weiss and Price, 1980), $p(H_2O)$ is the water vapour pressure, $p$ is the
total gas tension (assumed to be near 1 atm) and $K_0(CO_2)$ is the solubility coefficient. $F$ and $K_0$ vary according to
temperature and salinity.
**2.5 $CO_2$ optode lag and drift correction**
The $CO_2$ optode was fully functional between dives 31 (the 21 March 2014) and 400 (the 24 July 2014). After
dive 400, the $CO_2$ optode stopped sampling in the top 150 m. Figure 4 shows the outcome of each calibration
step described in this section (steps 1 and 2) and section 2.6 (step 3): 0) uncalibrated optode output (blue dots),
1) drift correction (red dots), 2) lag correction (green dots) and 3) calibration using discrete water samples
(black dots).

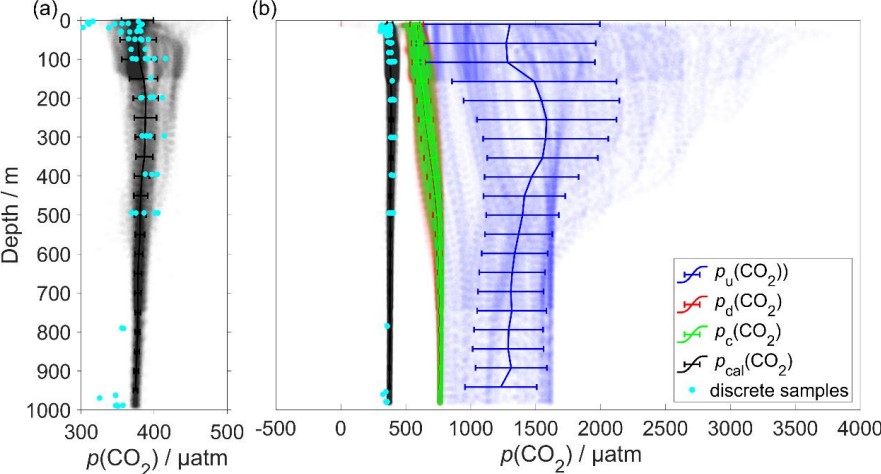

**Figure 4:** a) shows in black the calibrated $p(CO_2)$ ($p_{cal}(CO_2)$) and in azure the discrete samples b) $p(CO_2)$ versus
depth where the vertical continuous lines are the mean every 50 m and the error bars represent the standard
deviation. Blue colour shows $p_u(CO_2)$ without any correction; red shows $p_d(CO_2)$ corrected for drift, green
represents $p_c(CO_2)$ corrected for drift and lag; black shows $p_{cal}(CO_2)$ calibrated against water samples (azure
dots) collected during the deployment (section 2.6). $p_{cal}(CO_2)$ had a mean standard deviation of 22 µatm and a
mean bias of 8.4 µatm compared with the discrete samples.

In order to correct for the drift occurring during the glider mission, we selected the $CO_2$ optode measurements in
water with $\sigma_0 > 1028$ kg m$^{-3}$ (just as for $O_2$; section 2.3). We calculated the median of the raw optode phase shift





data ("CalPhase" $\varphi_{cal}$) for each Seaglider dive. Then, we calculated a drift coefficient ($m_i$) as the ratio between
the median $\varphi_{cal}$ for a given dive divided by the median $\varphi_{cal}$ of dive 31. Drift-corrected $\varphi_{cal,d}$ values were
calculated by dividing the raw $\varphi_{cal}$ by the specific $m_i$ for each dive.
The $CO_2$ optode was also affected by lag (Atamanchuk et al., 2014) caused by the slow response of the optode
to ambient $c(CO_2)$ changes in time and depth. The lag created a discrepancy between the depth profiles obtained
during glider ascents and descents. To correct for this lag we applied the method of Miloshevich et al. (2004),
which was previously used by Fiedler et al. (2013) and Atamanchuk et al. (2015b) to correct the lag of the
Contros HydroC $CO_2$ sensor (Fiedler et al., 2013; Saderne et al., 2013). This $CO_2$ sensor has a different
measurement principle (infrared absorption) than the $CO_2$ optode, but both rely on the diffusion of $CO_2$ through
a gas-permeable membrane.
To apply the lag correction, the sampling interval ($\Delta t$) needs to be sufficiently small compared to the sensor
response time ($\tau$) and the ambient variability (Miloshevich, 2004). Before the lag correction, $\varphi_{cal,d}$ was
rLOWESS-smoothed to remove any outliers and "kinks" in the profile. The smoothing function applies a local
regression every 9 points using a weighted robust linear least-squares fit. Subsequently, $\tau$ was determined such
that the following lag-correction equation (Miloshevich, 2004) minimised the $\varphi_{cal,d}$ difference between each
glider ascent and the following descent:
$$p_c(CO_2, t_1) = \frac{p_d(CO_2, t_1) - p_d(CO_2, t_0)\ e^{-\Delta t/\tau}}{1 - e^{-\Delta t/\tau}}$$    (3)
where $p_d(CO_2, t_0)$ is the drift-corrected value measured by the optode at time $t_0$, $p_m(CO_2, t_1)$ is the measured value
at time $t_1$, $\Delta t$ is the time between $t_0$ and $t_1$, $\tau$ is the response time, and $p_c(CO_2, t_1)$ is the lag-corrected value at $t_1$.
We calculated a $\tau$ value for each glider dive and used the median of $t_0$ (1384 s , 25[th] quartile: 1101 s; 75[th]
quartile: 1799 s) (Figure 5), which was larger than $\Delta t$ (258 s) and therefore met the requirement to apply the
Miloshevich (2004) method. This lag correction decreased the average difference between the glider ascent and
descent from (71±30) µatm to (21±26) µatm.



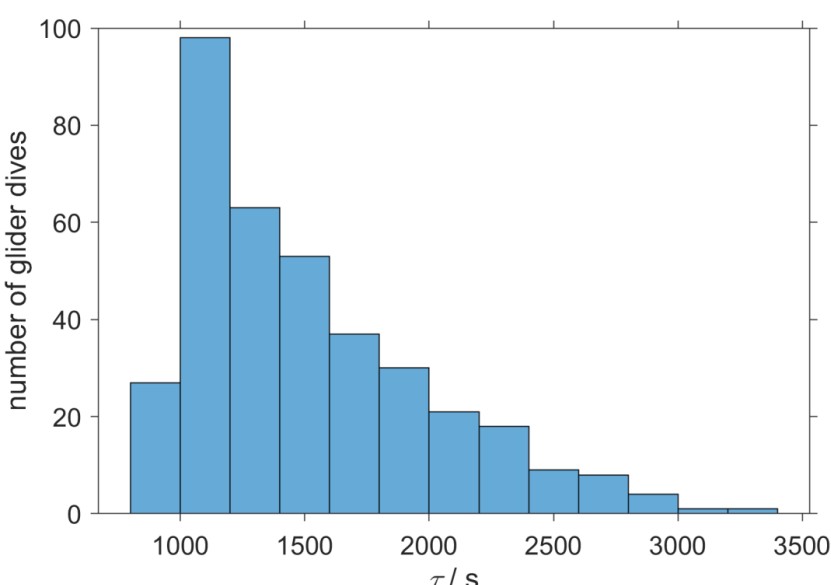


**Figure 5:** The histogram shows the distribution of the $\tau$ calculated from glider dive 31 to 400 to correct the $CO_2$
optode drift using the algorithm of Miloshevich (2004).



**2.6 $CO_2$ optode calibration**
The $CO_2$ optode output was calibrated using the discrete samples collected throughout the mission. Using the
discrete sample time and potential density $\sigma_0$, we selected the closest $CO_2$ optode output. Figure 6 shows a linear
regression between optode output and $c(CO_2)$ from the discrete samples ($c_{WS}(CO_2)$), which was used to calibrate
the optode output $p_c(CO_2)$ in terms of $c(CO_2)$. We used $c(CO_2)$ because it had a better correlation than $p(CO_2)$
($R^2 = 0.77$ vs. $R^2 = 0.02$). The residual difference in $c(CO_2)$ between glider and water samples had a standard
deviation of 1.3 µmol kg$^{-1}$.

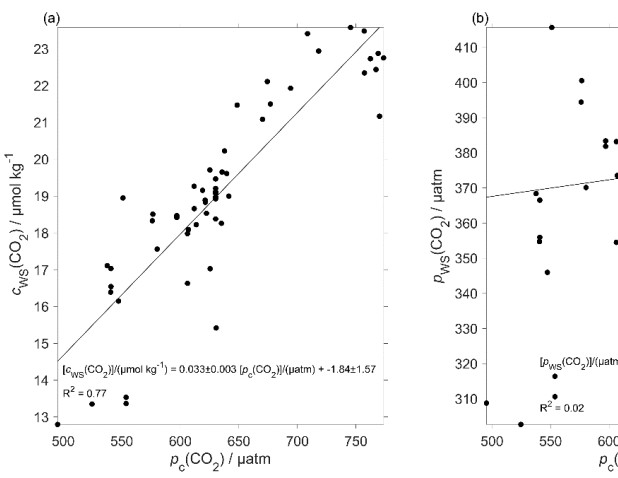






**Figure 6:** Calibration of the $CO_2$ optode using a) $CO_2$ concentration of the discrete samples ($c_{WS}(CO_2)$) against
the glider output with the linear regression line and b) $CO_2$ partial pressure of the discrete samples ($p_{WS}(CO_2)$)
against the glider output with the linear regression line.

**2.7 Regional algorithm to estimate $A_T$**
To calculate $C_T$, we used two variables: glider $c(CO_2)$ derived as described in section 2.6 and $A_T$ derived using a
regional algorithm that uses the top 1000 m $S$ and $\theta$. The algorithm followed the approach of Lee et al. (2006)
and was derived using 663 water samples collected at OWSM from 2004 to 2014 and GLODAPv2 (Olsen et al.,
2016) data from 2000 in the deployment region. Discrete samples with $S < 33$ were removed because these
values were lower than the minimum $S$ measured by the glider. The derived $A_T$ parameterisation is:
$A_{T,reg}$ / ($\mu$mol kg$^{-1}$) = 2317.03 + 33.12 ($S$–35) + 7.94 ($S$–35)$^2$ + 0.96 ($\theta$/°C–20) + 0.01 ($\theta$/°C–20)$^2$    (4)
The parameterisation has an uncertainty of 8.2 $\mu$mol kg$^{-1}$ calculated as the standard deviation of the residual
difference between actual and parameterised $A_T$.
To test this parameterisation, we compared the predicted $A_{T,reg}$ values with discrete measurements ($A_{T,WS}$)
collected close in terms of time, potential density ($\sigma_0$) and distance to the glider transect ($n = 60$). These discrete
samples and the glider had the mean temperature and salinity differences of (0.17±0.68) °C and 0.03±0.013,
respectively. The mean difference between $A_{T,WS}$ and $A_{T,reg}$ was  (2.1±6.5) $\mu$mol kg$^{-1}$.
This $A_T$ parameterisation was used in CO2SYS (Van Heuven et al., 2011) to calculate $C_T$ from $A_{T,reg}$ and the
calibrated $c(CO_2)$, $c_{G,cal}(CO_2)$. These calculated $C_{T,cal}$ values were compared with $C_{T,WS}$ of the same set of
discrete samples used to calibrate $c_{G,cal}(CO_2)$, the only difference being that instead of the actual total alkalinity
of the water sample ($A_{T,WS}$), we used  $A_{T,reg}$. The mean difference between $C_{T,cal}$ and $C_{T,reg}$ was (1.5±10) $\mu$mol
kg$^{-1}$, with the non-zero bias and the standard deviation due to the uncertainties in the $A_{Treg}$ parameterisation and
the $c_{G,cal}(CO_2)$ calibration.
**2.8 Quality control of other measurement variables**
The thermal lag of the glider conductivity sensor was corrected for (Gourcuff, 2014). Single-point outliers in
conductivity were removed and replaced by linear interpolation. The glider CTD salinity was affected by
presumed particulate matter stuck in the conductivity cell (Medeot et al., 2011) during dives 147, 234, 244, 251,
272, 279, 303, 320 and 397 and sensor malfunction caused a poor match between glider ascent and descent
during a dives 214, 215, 235 and 243. These dives were removed from the subsequent analysis.




Glider-reported chlorophyll concentrations, $c_{raw}$(Chl $a$), were affected by photochemical quenching during the
daytime dives. To correct for quenching, we used the method of Hemsley et al. (2015) based on the nighttime
relationship between fluorescence and optical backscatter. This relationship was established in the top 60 meters
and the nighttime values were selected between sunset and sunrise. We calculated a linear fit between $c_{raw}$(Chl
$a$) measured at night, $c_N$(Chl $a$), and the backscatter signal measured at night ($b_N$). The slope and the intercept
were then used to correct daytime $c_D$(Chl $a$). The glider-reported chlorophyll concentration has not been
calibrated against in situ samples and is not expected to be accurate, even after correction for quenching.
However, it should give an indication of the depth of the deep chlorophyll concentration maximum ($z_{DCM}$) and
the direction of chlorophyll concentration change (up/down). 8 day-means of $c_{raw}$(Chl $a$) were compared with
satellite 8 day-composite chlorophyll concentration (Figure 7) from Ocean Colour CCI (https://esa-
oceancolour-cci.org/) and gave a mean difference of (0.12±0.08) mg m$^{-3}$.

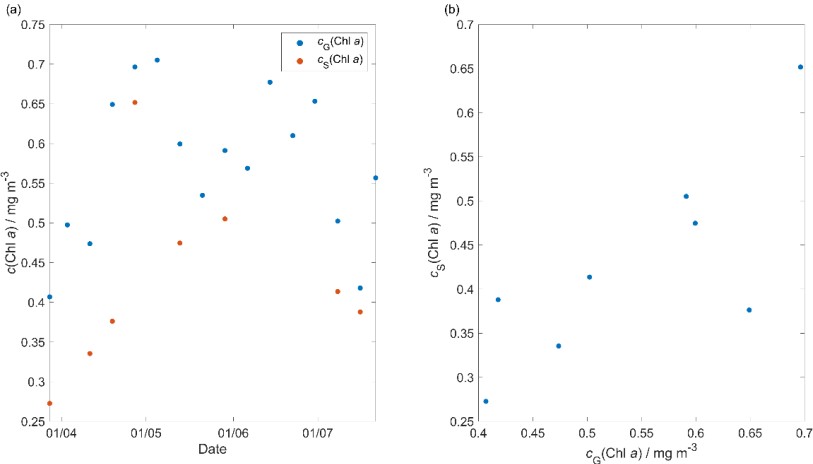


**Figure. 7:** comparison between the 8 days glider $c$(Chl $a$) ($c_G$(Chl $a$)) mean and the 8 days satellite $c$(Chl $a$)
($c_S$(Chl $a$)) download from Ocean Colour CCI (https://esa-oceancolour-cci.org/) where in a) $c_G$(Chl $a$) in red
and $c_S$(Chl $a$) in blue variability in time and in b) the direct comparison between $c_G$(Chl $a$) and $c_S$(Chl $a$).
**2.9 Calculation of oxygen-based net community production $N(O_2)$**
Calculating $N$ from glider data is challenging because the glider continuously moves through different water
masses. In particular, during the deployment, the glider sampled two different water masses: the Norwegian
Coastal Current (NCC) and the Norwegian Atlantic Current (NwAC) (Nilsen and Falck, 2006). During the
summer, the winds drive the NCC away from the coast (Skjelvan et al., 2008), and in July and August the glider
measured $S$ between 32 and 34 in the top 50 m. The two water masses were distinguished by different salinities;
the NwAC is saltier than the NCC (Nilsen and Falck, 2006; Swift, 1986). A threshold of $S$ = 35 was used to




distinguish between NCC and NwAC. We further subdivided the transect by binning the data into 0.1° latitude
intervals to derive $O_2$ concentration changes between transects.
We calculated $N(O_2)$ from the oxygen inventory changes ($\frac{\Delta I(O_2)}{\Delta t}$) corrected for air-sea exchange $\Phi(O_2)$
normalised to $z_{mix}$ when $z_{mix}$ was deeper than the integration depth of $z_{lim} = 45$ m and entrainment $E(O_2)$:
$$N(O_2) = \frac{\Delta I(O_2)}{\Delta t} + \Phi(O_2)\frac{\min(z_{lim}, z_{mix})}{z_{mix}} - E(O_2) \qquad (5)$$
The inventory changes were calculated as the difference between transects of the integrated $c(O_2)$ in the top 45
m. A constant integration depth of 45 m was chosen to capture the deepest extent of the deep chlorophyll
maximum ($z_{DCM}$) found during the deployment, which likely represents the lower bound for the euphotic zone.
The inventory changes were calculated using the following equation:
$$\frac{\Delta I(O_2)}{\Delta t} = \frac{\int_0^{45\,m} C_{n+1}dz - \int_0^{45\,m} C_n dz}{t_{n+1} - t_n} \qquad (6)$$
where $n$ is the transect number, $t$ is the year-day and $C$ is $c(O_2)$.
The air-sea flux of oxygen, $\Phi(O_2)$ was calculated for each glider dive using the median $c(O_2)$, $\theta$ and $S$ in the top
10 m. We followed the method of Woolf and Thorpe (1991) that includes the effect of bubble equilibrium
supersaturation in the calculations:
$$\Phi(O_2) = k_w(O_2) \{(c(O_2) - [1 + \Delta_{bub}(O_2)]c_{sat}(O_2)\} \qquad (7)$$
where $k_w(O_2)$ is the gas transfer coefficient, $\Delta_{bub}(O_2)$ is the increase of equilibrium saturation due to bubble
injection and $c_{sat}(O_2)$ is the oxygen saturation. $c_{sat}(O_2)$ was calculated from $S$ and $\theta$ using the solubility
coefficients of Benson and Krause Jr (1984), as fitted by Garcia and Gordon (1992). $\Delta_{bub}(O_2)$ was calculated
from the following equation:
$$\Delta_{bub}(O_2) = 0.01\left(\frac{U}{U_0}\right)^2 \qquad (8)$$
where $U$ is 10 m-wind speed with 6 hours resolution (ECMWF ERA InterimDaily,
https://apps.ecmwf.int/datasets/data/interim-full-daily/levtype=sfc/) and $U_0$ represents the wind speed
when the oxygen concentration is 1 % supersaturated and has a value of 9 m s$^{-1}$ (Woolf and Thorpe, 1991). $U$
has a spatial resolution of 0.75° latitude and 0.75° longitude and was interpolated to the glider position at the
beginning of the dive.
The transfer velocity $k_w(O_2)$ was calculated based on Wanninkhof (2014):

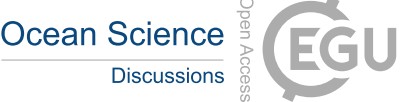



$\frac{k_{\mathrm{w}}(O_2)}{\mathrm{cm\,h^{-1}}} = 0.251 \left(\frac{Sc(O_2)}{660}\right)^{-0.5} \left(\frac{U}{\mathrm{m\,s^{-1}}}\right)^2$ (9)
The Schmidt number, $Sc(O_2)$, was calculated using the parameterisation of Wanninkhof (2014).
The entrainment flux, $E(O_2)$, was calculated as the oxygen flux when the mixed layer depth deepens in time and
is greater than $z_{\mathrm{lim}}$ at time $t_2$:
$E(O_2) = \frac{I(O_2,t_1,z_{\mathrm{mix}}(t_2))\frac{z_{\mathrm{lim}}}{z_{\mathrm{mix}(t_2)}} - I(O_2,t_1,z_{\mathrm{lim}})}{t_2 - t_1}$ (10)
where $t_2 - t_1$ represents the change in time, $z_{\mathrm{mix}}$ is the mixed layer depth, $I(O_2,t_1,z_{\mathrm{mix}}(t_2))$, is the expected
inventory that would result from a mixed layer deepening to $z_{\mathrm{mix}}(t_2)$ between $t_2$ and $t_1$, and $I(O_2,t_1,z_{\mathrm{lim}})$ is the
original inventory at $t_1$.
Then, we calculated $N(O_2)$ with $\frac{\Delta I(O_2)}{\Delta t}$ corrected for $\Phi(O_2)$ and $E(O_2)$ using equation 5.
**2.10 Calculation of dissolved inorganic carbon-based net community production, $N(C_T$**
$N(C_T)$ was calculated from the $C_T$ inventory changes $\frac{\Delta I(C_T)}{\Delta t}$, air-sea flux of $CO_2$, $\Phi F(CO_2)$, and entrainment
$E(C_T)$:
$N(C_T) = -\frac{\Delta I(C_T)}{\Delta t} - \Phi(CO_2)\frac{\min(z_{\mathrm{lim}},z_{\mathrm{mix}})}{z_{\mathrm{mix}}} + E(C_T)$ (11)
Firstly, $\Phi(CO_2)$ was calculated using the 10 m wind speed with 6 hours' resolution downloaded from ECMWF
ERA Interim Daily. As for oxygen, we selected the closest wind speed data point at the beginning of each glider
dive. We used the monthly mean atmospheric $CO_2$ dry mole fraction ($x(CO_2)$) downloaded from the Greenhouse
Gases Reference Network Site from (https://www.esrl.noaa.gov/gmd/ccgg/ggrn.php) the closest site to the
deployment at Mace Head, County Galway, Ireland (Dlugokencky et al., 2015). Using $x(CO_2)$ we calculated the
air-saturation concentration $c_{\mathrm{atm}}(CO_2)$:
$c_{\mathrm{atm}}(CO_2) = x(CO_2)\, p_{\mathrm{baro}}\, F(CO_2)$ (12)
where $p_{\mathrm{baro}}$ is the mean sea level pressure and $F(CO_2)$ is the $CO_2$ solubility function calculated from surface $\theta$
and $S$ (Weiss and Price, 1980).
The seawater $c(CO_2)$ at the surface was calculated using the median in the top 10 meters between the glider
ascent and descent of the following dive $c(CO_2)$. From this, $\Phi(CO_2)$ was calculated:
$\Phi(CO_2) = k(CO_2)\,[c(CO_2) - c_{\mathrm{atm}}(CO_2)]$. (13)





$k(CO_2)$ was calculated using the parameterisation of Wanninkhof (2014):
$$\frac{k(O_2)}{cm\ h^{-1}} = 0.251 \left(\frac{Sc(CO_2)}{660}\right)^{-0.5} \left(\frac{U}{m\ s^{-1}}\right)^2 \qquad (14)$$
$Sc(CO_2)$ is the dimensionless Schmidt number at the seawater temperature (Wanninkhof, 2014).
The inventory changes were calculated in the top 45 m with the following equation:
$$\frac{\Delta I(C)}{\Delta t} = \frac{\int_0^{45\ m} C_{n+1} dz - \int_0^{45\ m} C_n dz}{t_{n+1} - t_n} \qquad (15)$$
The entrainment flux, $E(C_T)$ was calculated as the oxygen flux when the mixed layer depth deepens in time and
is greater than $z_{lim}$ at time $t_2$:
$$E(C_T) = \frac{I(C, t_1, z_{mix}(t_2))\frac{z_{lim}}{z_{mix}(t_2)} - I(C, t_1, z_{lim})}{t_2 - t_1} \qquad (16)$$
The uncertainties in $N(C_T)$ and $N(O_2)$ were evaluated with a Monte-Carlo approach. The uncertainties of the
input variables are shown in Table 2; we repeated the analysis 1000 times. The total uncertainty in $N$ was
calculated as the standard deviation of the 1000 Monte-Carlo simulations.
**Table 2.** Uncertainty associated with $N(C_T)$ and $N(O_2)$ input variables calculated by a Monte Carlo approach

| Variable | Error | Reference/Method |
|---|---|---|
| $C_T$ | 10 µmol kg$^{-1}$ | Standard deviation vs the water samples. |
| $S$ | 0.01 | Standard deviation of glider salinities for $\sigma_0 > 1028$ kg m$^{-3}$ and latitude $> 64°$ N |
| $\theta$ | 0.3 °C | Standard deviation of glider temperature for $\sigma_0 > 1028$ kg m$^{-3}$ and latitude $> 64°$ N |
| $c_{atm}(CO_2)$ | 1.5 µmol kg$^{-1}$ | Standard deviation of $c_{atm}(CO_2)$ |
| $c(CO_2)$ | 1.3 µmol kg$^{-1}$ | Error is the standard deviation vs water samples. |
| $k(CO_2)$ | 20 % | (Wanninkhof, 2014) |
| $z_{mix}$ | 9 m | Standard deviation compared with $z_{mix}$ based on thresholds $\Delta T = 0.1$ °C (Sprintall and Roemmich, 1999), 0.2 °C (Thompson, 1976) and 0.8 °C (Kara et al., 2000). |
| $z_{mix}$ latitude | 0.32 m | Standard deviation compared with $z_{mix}$ based on thresholds $\Delta T = 0.1$ °C (Sprintall and Roemmich, 1999), 0.2 °C (Thompson, 1976) and 0.8 °C (Kara et al., 2000). |
| $c_G(O_2)$ | 2.4 µmol kg$^{-1}$ | Standard deviation of glider oxygen concentrations for $\sigma_0 > 1028$ kg m$^{-3}$ and latitude $> 64°$ N |


## 3 Results

The uncorrected temperature $\theta$, salinity $S$, $c(O_2)$, $p(CO_2)$ and $c_{raw}$(Chl $a$) presented in Figure 8 were analysed to
dive 400 dive(24 July 2014) because after the $CO_2$ optode stopped sampling in the first 150 m (Figure 8d). The
rawdata $c(O_2)$ data was calibrated and drift corrected and $c(CO_2)$ was drift, lag and recalibrated and then used to





quantify the temporal and spatial changes in $N$ and $\Phi$ together with the quenching corrected $c_{\mathrm{raw}}(\mathrm{Chl}\ a)$ to
evaluate the net community production changes.








**Figure 8:** Raw glider data for all 703 dives with latitude of the glider trajectory at the top (black: NwAC; red:
NCC, separated by a $S$ of 35). a) temperature $\theta$, b) salinity $S$, c) oxygen concentration $c(O_2)$, d) uncorrected $CO_2$
optode output $p_u(CO_2)$ and e) chlorophyll $a$ concentration $c_{raw}(Chl\ a)$. The white space means that the sensors
did not measure any data.

**3.1 $O_2$ optode calibration**
The uncorrected $c(O_2)$ continually decreased (Figure 8c). The ratio $c_C(O_2)/c_G(O_2)$ and against year-day used for
the drift correction had a good correlation with time ($R^2 = 0.90$), showing a continuous increase of 0.0004 $d^{-1}$
(Figure 3), equivalent to a decrease in the measured glider $O_2$ concentration of 0.11 µmol $kg^{-1}$ $d^{-1}$. It was
possible to apply the correction because $c_C(O_2)$ had low temporal variability. The discrete oxygen samples from
OWSM and GLODAPv2 had a mean of (304.6±3.1) µmol $kg^{-1}$, varying from 294 to 315 µmol $kg^{-1}$. The drift
correction reduced the variability of $c_G(O_2)$ in the selected potential density range from a standard deviation of
7.3 µmol $kg^{-1}$ to a standard deviation of 2.4 µmol $kg^{-1}$ (Figure 9).

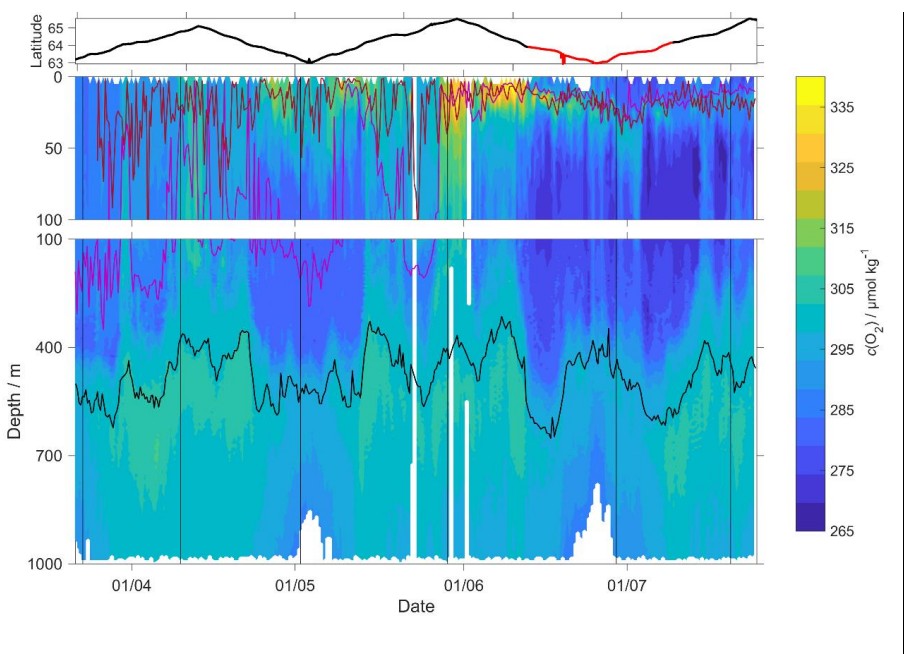

**Figure 9:** $c(O_2)$ contour plot with $z_{DCM}$ (red line) and the $z_{mix}$ (pink line) calculated using a threshold criterion of
$\Delta\theta = 0.5\ °C$ to median $\theta$ of the top 5 m of the glider profile (Obata et al., 1996; United States. National
Environmental Satellite and Information Service, Monterey and Levitus, 1997; Fo*ltz* et al., 2003), in black $\sigma_0 = $
1028 kg $m^{-3}$ and at the top the latitude trajectory of the glider in black NwAC and in red NCC.

**3.2 $CO_2$ optode calibration**
Following drift, lag and scale corrections, glider fugacity $f_G(CO_2)$ derived from Eq. 2 had a mean difference of
(8±22) µatm to the discrete samples ($n = 55$; not shown) and $C_T$ had a standard deviation of 10 µmol $kg^{-1}$ and a





mean difference of 1.5 µmol kg$^{-1}$ (Figure 10). $p(CO_2)$ and $f(CO_2)$ are almost the same numbers, specifically
$f(CO_2)$ takes into account of the non-ideal nature of the gas phase. The optode was able to capture the temporal
and spatial variability showing that NCC had a lower concentration of $C_T$ than NwAC. Restricting the discrete
samples $f(CO_2)$ to the top 10 m only gave a mean difference of (21±21) µatm ($n = 8$). We also compared glider
$f_G(CO_2)$ with SOCAT $f(CO_2)$ (Bakker et al., 2016) data in the region during the deployment (Figure 11). Until
the beginning of June, there was general agreement between $f_G(CO_2)$ and $f_{SOCAT}(CO_2)$. Afterwards, $f_G(CO_2)$
varied between 326 and 434 µatm while $f_{SOCAT}(CO_2)$ varied between 259 and 354 µatm (Figure 11).
Our results are in agreement with Jeansson et al. (2011) that found the surface NCC was the region with the
lowest $C_T$ values (2083 µmol kg$^{-1}$) in the Norwegian Sea. This was confirmed during our deployment because
$C_T$ was (2100±18) µmol kg$^{-1}$ in the NCC region and (2150±23) µmol kg$^{-1}$ in the NwAC region (Figure 10) and
$c(O_2)$ was >300 µmol kg$^{-1}$ in the NwAC and < 280 µmol kg$^{-1}$ in the NCC.

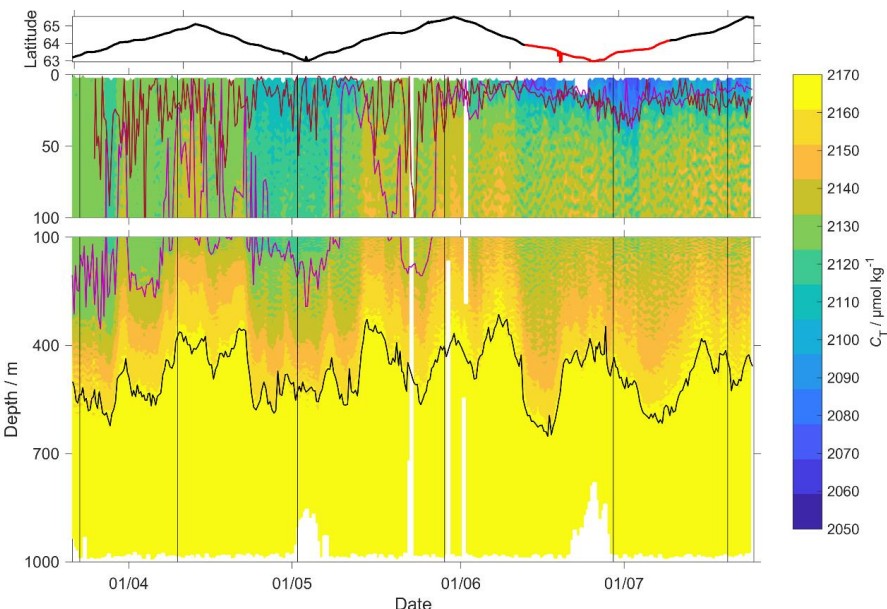

**Figure 10:** $C_T$ contour plot with $z_{DCM}$ (red line) and the $z_{mix}$ (pink line) calculated using a threshold criterion of
$\Delta\theta = 0.5$ °C to median $\theta$ of the top 5 m of the glider profile (Obata et al., 1996; United States. National
Environmental Satellite and Information Service, Monterey and Levitus, 1997; Fo*ltz* et al., 2003), in black $\sigma_0 =$
1028 kg m$^{-3}$ and at the top the latitude trajectory of the glider in black NwAC and in red NCC.



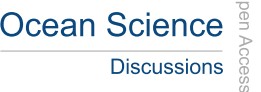

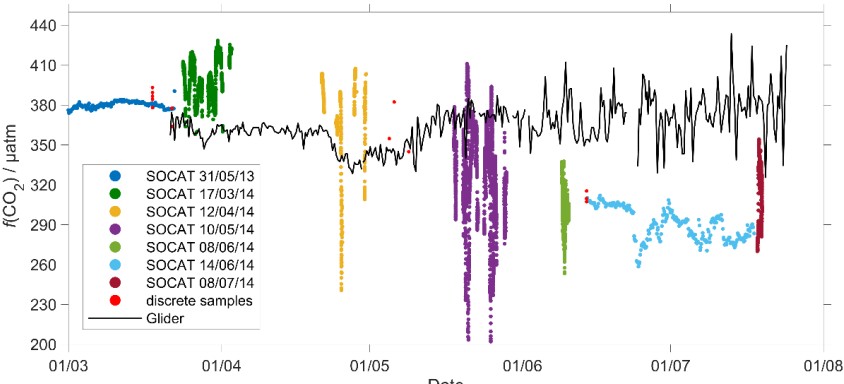

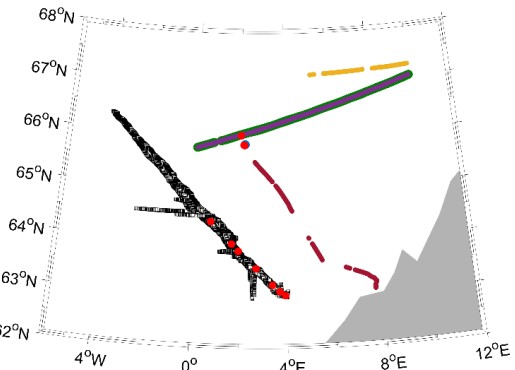

**Figure 11:** The plot represents the surface $f(CO_2)$ from 2014 SOCAT and from the glider. The black dots are the median of the glider $f(CO_2)$ in the top 10 meters calculated using the ascent of the single dive and the descent of the next dive. The red dots are the water samples collected during the deployment and the remaining dots are from the SOCAT cruises in the area during the deployment. On the bottom there is the map of the glider and SOCAT data positions.

### 3.3 Air-sea exchange

The surface water was supersaturated with oxygen all summer (Figure 12). From May this supersaturation drove

a continuous $O_2$ flux from the sea to the atmosphere. However, the flux varied throughout the deployment

having a median of 21 mmol m$^{-2}$ d$^{-1}$ (5$^{th}$ centile: -16 mmol m$^{-2}$ d$^{-1}$; 95$^{th}$ centile: 102 mmol m$^{-2}$ d$^{-1}$). Prior to the

spring period of increased Chl $a$ inventory, the supersaturation varied between 0 to 10 µmol kg$^{-1}$. $\Phi(O_2)$ had a

median of -0.2 mmol m$^{-2}$ d$^{-1}$ (5$^{th}$ centile: -57 mmol m$^{-2}$ d$^{-1}$; 95$^{th}$ centile: 12 mmol m$^{-2}$ d$^{-1}$). Then, during the

spring period of increased Chl $a$ inventory, the surface concentration increased by over 35 µmol kg$^{-1}$, causing a

peak in $\Phi(O_2)$ of 104 mmol m$^{-2}$ d$^{-1}$. A second period of increased Chl $a$ inventory was encountered in June and





had a larger $\Phi(O_2)$ up to 168 mmol m$^{-2}$ d$^{-1}$, driven by supersaturation of 68 µmol kg$^{-1}$. These larger fluxes during
the second period of increased Chl $a$ inventory were associated by an increase of $c_{raw}$(Chl a) from 2.5 mg m$^{-3}$ to
the summer maximum of 4.0 mg m$^{-3}$. However, prior to the spring period of increased Chl $a$ inventory, $\Phi(O_2)$
showed a few days of influx into seawater caused by a decrease of $\theta$ from 7.6 °C to 5.9 °C that increased
$c_{sat}(O_2)$. The influx at the beginning of the deployment is partly due to the $\Delta_{bub}(O_2)$ correction that increased [1+
$\Delta_{bub}(O_2)$]$c_{sat}(O_2)$ to values larger than $c(O_2)$ for $U > 10$ m s$^{-1}$.
The CO$_2$ flux from March to July was always from the air to the sea (Figure 13), with a median of -2.0 mmol m$^{-}$
$^2$ d$^{-1}$ (5$^{th}$ centile: -11 mmol m$^{-2}$ d$^{-1}$; 95$^{th}$ centile: 0.24 mmol m$^{-2}$ d$^{-1}$). An opposite flux direction is expected for
$\Phi(O_2)$ and $\Phi(CO_2)$ during the productive season when net community production is the main driver of
concentration changes. After the summer period of increased Chl $a$ inventory, the flux had a median of -0.32
mmol m$^{-2}$ d$^{-1}$ (5$^{th}$ centile: –2.4 mmol m$^{-2}$ d$^{-1}$; 95$^{th}$ centile: 1.1 mmol m$^{-2}$ d$^{-1}$). Positive fluxes (from water to air)
are in disagreement with previous studies that classified the Norwegian Sea as a CO$_2$ sink (Skjelvan et al., 2005;
Takahashi et al., 2002). Calculating $\Phi(CO_2)$ from the discrete samples from the 18 March to the 14 June ($n =$
13) the flux varied from 0.1 to -13 mmol m$^{-2}$ d$^{-1}$ with just one positive $\Phi(CO_2)$ in March.



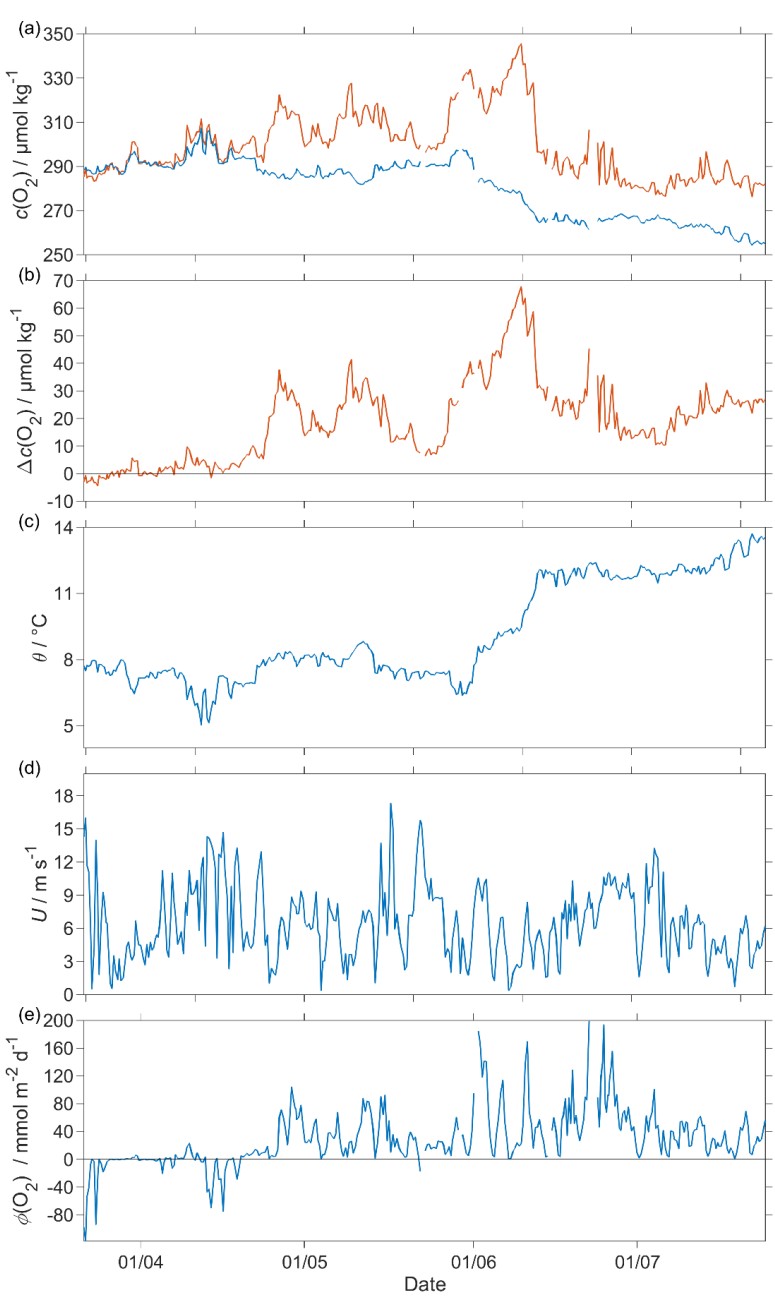

**Figure 12:** a) shows in blue $c_{sat}(O_2)$ and in red $c(O_2)$, b) the difference between $c(O_2)$ and $c_{sat}(O_2)$ ($\Delta c(O_2)$), c)
the surface $\theta$, d) 10 metre wind speed ($U$) and e) oxygen air-sea flux $\Phi(O_2)$ from sea to air for each glider dive.
Flux from sea to air is positive while that from air to sea is negative.



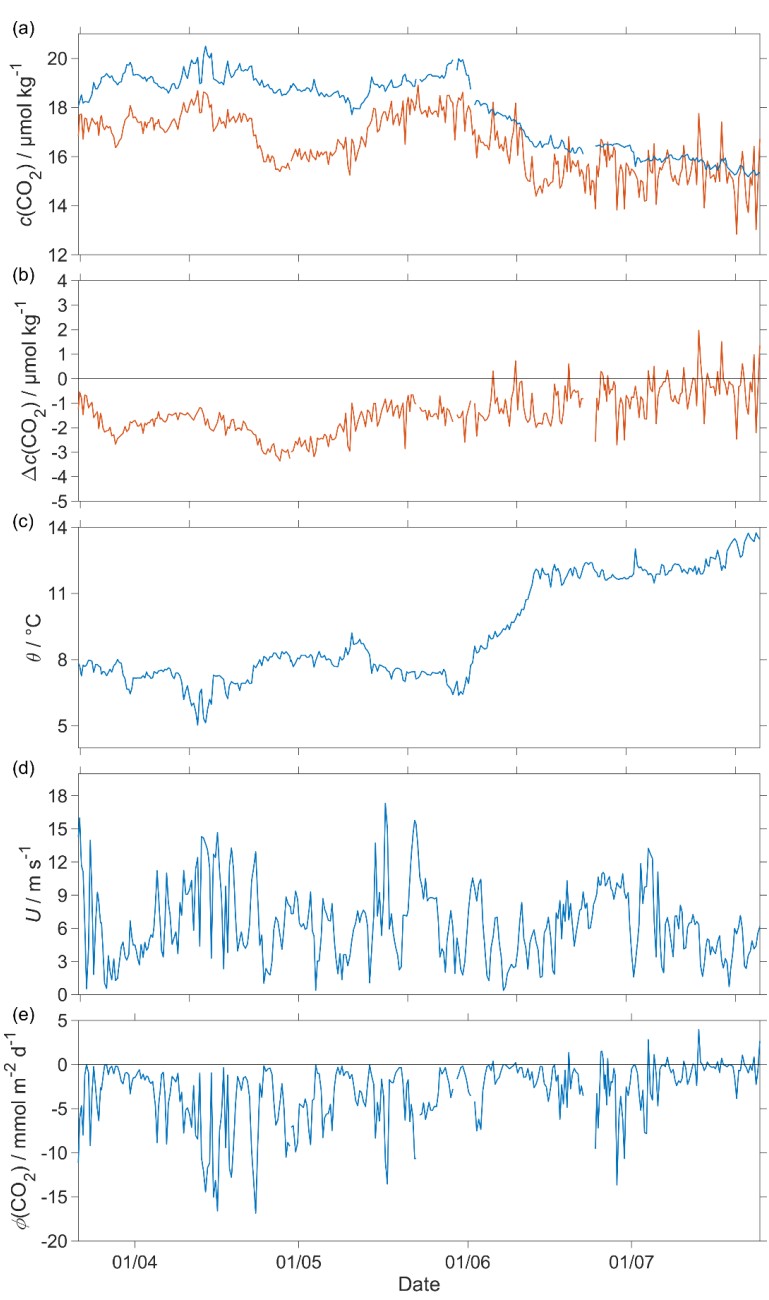

**Figure 13:** a) shows in blue $c_{sat}(CO_2)$ and in red $c(CO_2)$, b) the difference between $c(CO_2)$ and $c_{sat}(CO_2)$ ($\Delta c(CO_2)$), c) the surface $\theta$, d) 10 metre wind speed ($U$) and e) $CO_2$ air-sea flux $\Phi(CO_2)$ from sea to air for each glider dive. Flux from sea to air is positive while that from air to sea is negative.



**3.4 $N(O_2)$**
We calculated $N(O_2)$ and $N(C_T)$ using an integration depth of $z_{lim} = $ 45 m because the mean deep chlorophyll
maximum (DCM) depth was $z_{DCM} = (20\pm18$ m) (Figure 9). For comparison, the mixed layer depth was deeper
and varied more strongly and had a mean value of $z_{mix} = (73\pm74)$ m, using a threshold criterion of $\Delta\theta = 0.5$ °C to
the median $\theta$ value of the top 5 m of the glider profile (Obata et al., 1996; United States. National
Environmental Satellite and Information Service, Monterey and Levitus, 1997; Foltz et al., 2003).
The two $N$s were calculated as the difference in inventory changes between two transects when the glider was in
the same water mass. For that reason, Figures 14-16 starts from the second glider transect and in the second part
of no $N$s values were calculated because the glider started the first transect in NCC.
During the deployment, we sampled two periods of increased Chl $a$ inventory, the first one in May and a second
one in June. The Chloropyll $a$ inventory ($I_{raw,z_{lim}}(\text{Chl } a)$) was calculated integrating $c_{raw}(\text{Chl } a)$ to $z_{lim}$ The
fluorometer was not calibrated for that reason to remove any outliers we used a five points moving mean of
$I_{raw,z_{lim}}(\text{Chl } a)$.

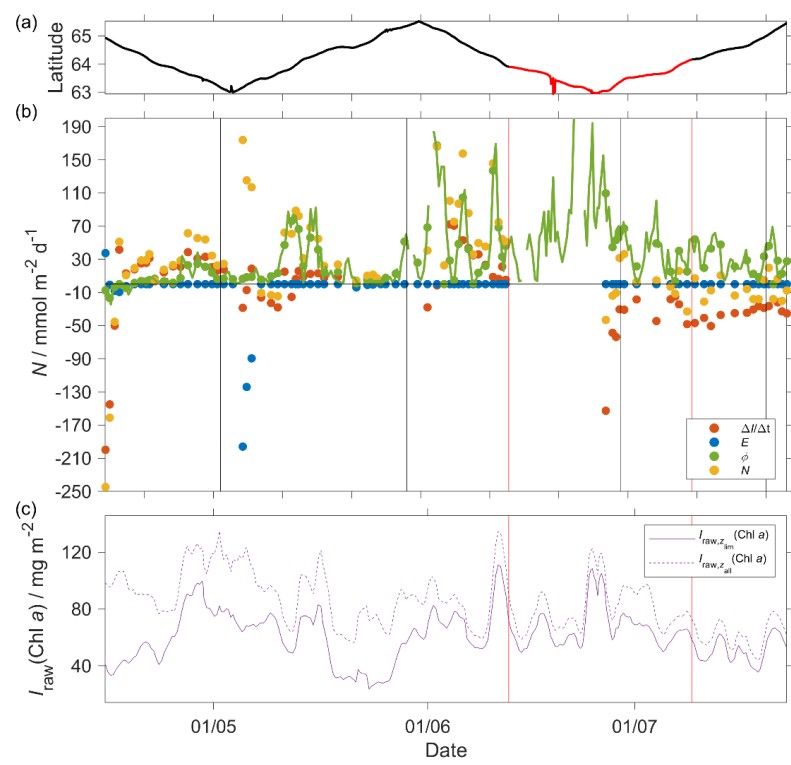




**Figure 14:** a) The trajectory in latitude of the glider where NwAC and NCC are in black and red, respectively .
b) Each component of the $N(O_2)$ calculation: in red $\frac{\Delta I(O_2)}{\Delta t}$, $E(O_2)$ in blue, $\Phi(O_2)$ in green dots and the green line
is $\Phi(O_2)$ continuous timeseries and in yellow $N(O_2) = \frac{\Delta I(O_2)}{\Delta t} + \Phi(O_2)\frac{min(z_{lim},z_{mix})}{z_{mix}} - E(O_2)$ c) the violet
continuous line is the $c_{raw}$(Chl $a$) inventory in the top 45 m, $z_{lim}$, $(I_{raw,z_{lim}}$(Chl $a$)) and the dotted line in all the
water columun, $z_{all}$, $(I_{raw,z_{all}}$(Chl $a$)). The black vertical lines represent each glider transect and between the two
vertical red lines the glider was in NCC.

During the summer $I_{raw,z_{lim}}$(Chl $a$) increased to 110 mg m$^{-2}$, which caused a sharp increase of $N(O_2)$ to
(166±75) mmol m$^{-2}$ d$^{-1}$. However, we were not able to see the end of this productive period because the glider
moved into NCC. The passage of the glider from NwAC to NCC accompanied by a drop of surface $c(O_2)$ from
330 to 280 µmol kg$^{-1}$ (Figure 9) that resulted in lower $\Phi(O_2)$ and $N(O_2)$ values (Figure 14).
At the beginning of May, $I_{raw,z_{lim}}$(Chl $a$) increased to 97 mg m$^{-2}$ and $N(O_2) = (61±14)$ mmol m$^{-2}$ d$^{-1}$. After this
period, $I_{raw,z_{lim}}$(Chl $a$) decreased to 49 mg m$^{-2}$ and $N(O_2) = (-15±27)$ mmol m$^{-2}$ d$^{-1}$. However, a deepening of
$z_{mix}$ from 123 m to 206 m caused a spike in the entrainment flux $E(O_2)$ of 190 mmol m$^{-2}$ d$^{-1}$ that drove the
maximum $N(O_2)$ to (174±72) mmol m$^{-2}$ d$^{-1}$.
Using the mean of $N(O_2)$ considering together NCC and NwAC and assuming an $N(O_2) = 0$ in the rest of the
year lead to an annual value of 10 mol m$^{-2}$ a$^{-1}$ (Table 3) discussed in section 4.3.
**Table 3.** $N$ estimates in the Norwegian Sea

| Study | $N(C_T)$ / mol m$^{-2}$ a$^{-1}$ | $N(O_2)$ / mol m$^{-2}$ a$^{-1}$ | $z_{lim}$ / m | Variables used to derive $N$ |
|---|---|---|---|---|
| (Falck and Anderson, 2005) | 3.4 | — | 100 | $c(NO_3^-)$, $c(PO_4^{3-})$, $C_T$ |
| (Skjelvan et al., 2001) | 2.0 | 2.6 | 300 | $c(O_2)$, $c(PO_4^{3-})$ |
| (Kivimäe, 2007) | 8.6 | 11 | $z_{mix}$ until 100 m | $c(O_2)$ |
| (Falck and Gade, 1999) | 3.0 | 3.9 | 30 | $c(O_2)$ |
| This study | 3.3 | 10 | 45 | $c(O_2)$, $C_T$ |


**3.5 $N(C_T)$**
Before the spring period of increased Chl $a$ inventory, $N(C_T)$ was (-15±14) mmol m$^{-2}$ d$^{-1}$ and increased to (23±5)
mmol m$^{-2}$ d$^{-1}$ during the period of increased Chl $a$ inventory (Figure 15). Later during the summer high Chl $a$
period $N(C_T)$ had its summer maximum in NwAC at (64±67) mmol m$^{-2}$ d$^{-1}$. The first transect in NCC was
characterised by an increase of $I_{raw,z_{lim}}$(Chl $a$) on the 11 June to 110 mg m$^{-2}$ and $N(C_T)$ to (75±58) mmol m$^{-2}$ d$^{-}$
$^{1}$. Later, on the 4 July $N(C_T)$ decreased to (-14±22) mmol m$^{-2}$ d$^{-1}$ when $I_{raw,z_{lim}}$(Chl $a$) dropped to 50 mg m$^{-2}$.





Using the mean of $N(C_T)$ in NCC and NwAC with the assumption that during the rest of year $N(C_T) = 0$, we

calculated the annual $N(C_T)$ of 3.3 mol m$^{-2}$ a$^{-1}$ (Table 3) that its implications are discussed in section 4.3.

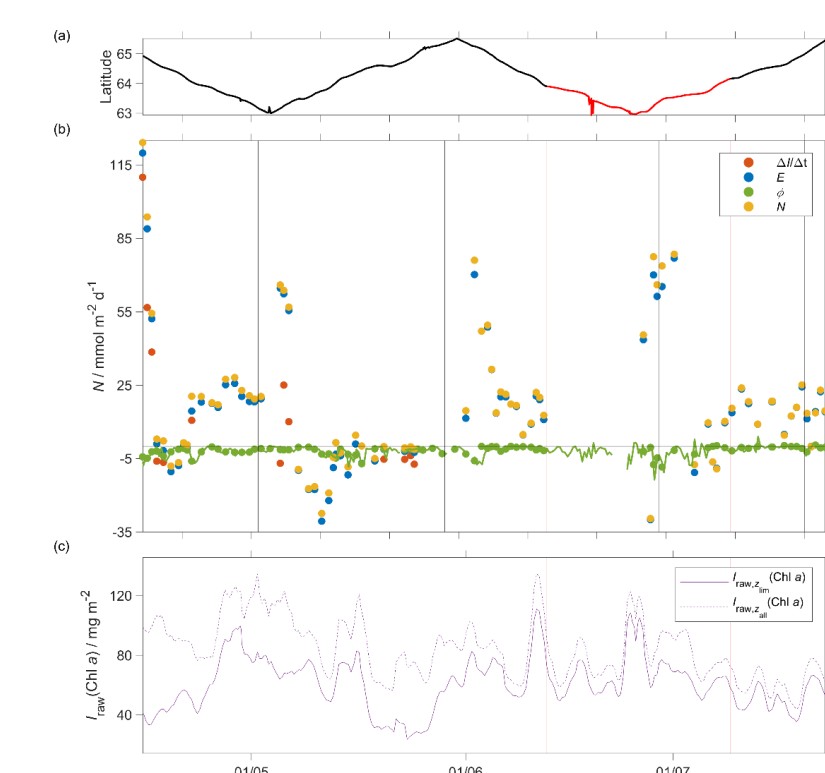

**Figure 15:** a) shows the trajectory in latitude of the glider where NwAC and NCC are in black and red, respectively. b) Each component of the $N(C_T)$ calculation: in red $\frac{\Delta I(C_T)}{\Delta t}$, $E(C_T)$ in blue, $\Phi(C_T)$ in green dots and the green line is $\Phi(O_2)$ continuous time-series and in yellow $N(C_T) = \frac{\Delta I(C_T)}{\Delta t} + \Phi(C_T)\frac{\min(z_{\mathrm{lim}},z_{\mathrm{mix}})}{z_{\mathrm{mix}}} - E(C_T)$ c) the violet continuous line is the $c_{\mathrm{raw}}$(Chl $a$) inventory in the top 45 m, $z_{\mathrm{lim}}$, $(I_{\mathrm{raw},z_{\mathrm{lim}}}$(Chl $a$)) and the dotted line in all the water columun, $z_{\mathrm{all}}$, $(I_{\mathrm{raw},z_{\mathrm{all}}}$(Chl $a$)). The black vertical lines represent each glider transect and between the two vertical red lines the glider was in NCC.

### 3.6 Comparison of $N(C_T)$ and $N(O_2)$

To compare $N(C_T)$ and $N(O_2)$, we divided $N(O_2)$ for the photosynthesis quotient (PQ) of 1.9 calculated as the

slope of the fitting between $N(C_T)$ and $N(O_2)$ using a geometric mean regression (Leng et al., 2007) excluding

the first 2 measurements, designated $N_C(O_2)$. The geometric mean regression was necessary because both the

variable had large errors. Figure 16 shows the changes of $N(C_T)$, $N_C(O_2)$ and $I_{\mathrm{raw}}$(Chl $a$) in time. During the

spring and summer periods of increased Chl $a$ inventory, the $N(C_T)$ and $N_C(O_2)$ increased simultaneously.





During the spring period of increased Chl *a* inventory $N(C_T)$ increased to $(23\pm5)$ mmol m$^{-2}$ d$^{-1}$ and $N(O_2)$ to
$(32\pm7)$ mmol m$^{-2}$ d$^{-1}$. After the period of increased Chl *a* inventory, $N(C_T)$ decreased to $(-80\pm107)$ mmol m$^{-2}$ d$^{-1}$
and $N(O_2)$ to $(-7\pm14)$ mmol m$^{-2}$ d$^{-1}$.
During the summer period of increased Chl *a* inventory, $N(C_T)$ and $N(O_2)$ reached the summer maximum in the
NwAC region at $(75\pm58)$ mmol m$^{-2}$ d$^{-1}$ for $N(C_T)$ and $(87\pm39)$ mmol m$^{-2}$ d$^{-1}$ for $N(O_2)$. Later in the NCC region,
the $I_{raw,z_{lim}}$(Chl *a*) reached a summer maximum of 110 mg m$^{-2}$, $N(C_T)$ increased to $(75\pm58)$ mmol m$^{-2}$ d$^{-1}$ and
$N(O_2)$ decreased to $(-23\pm66)$ mmol m$^{-2}$ d$^{-1}$.

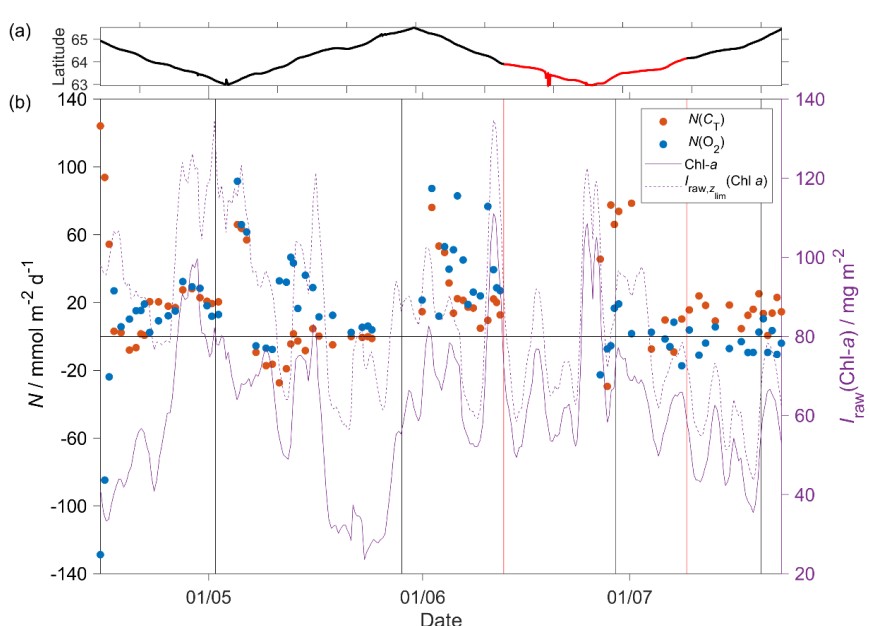

**Figure 16:** a) shows the trajectory in latitude of the glider where NwAC and NCC are in black and red,
respectively. b) Production estimates: in red $N(C)$ and in blue $N(O_2)$ divided per a photosynthesis quotient of
1.9. The violet line on the secondary y-axis shows the $c_{raw}$(Chl *a*) inventory in the top 45 m, $z_{lim}$,
$(I_{raw,z_{lim}}$(Chl *a*)) and the dotted line in all the water columun, $z_{all}$, $(I_{raw,z_{all}}$(Chl *a*)). Also, the black vertical
lines represent each glider transect and on the top there is the latitude glider trajectory with NwAC in black and
NCC in red.

**4 Discussion**
**4.1 Sensor performance**
This study presents data from the first glider deployment with a $CO_2$ optode. The initial uncalibrated $p(CO_2)$,
$p_U(CO_2)$, measured by the $CO_2$ optode had a median of 604 µatm (5$^{th}$ centile: 566 µatm; 95$^{th}$ centile: 768 µatm



when the $p(CO_2)$ of discrete samples varied from 302 to 421 µatm. This discrepancy was caused by sensor drift
prior to and during deployment of the optode.
We applied corrections for drift (using deep-water samples as a reference point), sensor lag and calibrated the
$CO_2$ optode against co-located discrete samples throughout the water column.
Atamanchuk (2014) reported that the sensor was affected by a lag that varied from 45 to 264 s depending on
temperature. These values were determined in an actively stirred beaker. However, in this study the sensor was
mounted on a glider and was not actively pumped, which increased the response time to (1384 s , 25[th] quartile:
1101 s; 75[th] quartile: 1799 s). Also, the optode was affected by a continuous drift from 637 to 5500 µatm that is
larger than the drift found by Atamanchuk et al. (2015a) that increased by 75 µatm after 7 months.
In this study, the drift- and lag-corrected sensor output showed a better correlation with the $CO_2$ concentration
$c(CO_2)$ than with $p(CO_2)$. The latter two quantities are related to each other by the solubility that varies with $\theta$
and $S$ (Weiss, 1974) (Eq. 2).
The calibrated optode output captured the $C_T$ changes in space and time with a standard deviation of 10 µmol
kg$^{-1}$ compared with the discrete samples. $C_T$ decreased from 2100 µmol kg$^{-1}$ to 2050 µmol kg$^{-1}$ and increased
with depth to 2170 µmol kg$^{-1}$. This shows the potential of the sensor for future studies that aim to analyse the
carbon cycle using a high-resolution dataset.
The optode-derived $CO_2$ fugacity $f_G(CO_2)$ had a mean bias of (8±22) µatm compared with the discrete samples.
These values are comparable with a previous study when the $CO_2$ optode was tested for 65 days on a wave-
powered Profiling crAWLER (PRAWLER) from 3 to 80 m (Chu et al., 2020), which had an uncertainty
between 35 and 72 µatm. The PRAWLER optode was affected by a continuous drift of 5.5 µatm d$^{-1}$ corrected
using a regional empirical algorithm that uses $c(O_2)$, $\theta$, $S$ and $\sigma_o$ to estimate $A_T$ and $C_T$.
**4.2 Norwegian Sea net community production**
Increases in $N(O_2)$ and $N(C_T)$ were associated with increases in depth-integrated $c_{raw}(\text{Chl a})$, designated as
periods of increased Chl $a$ inventory, at the beginning of May and in June. During the first period of increased
Chl $a$ inventory at the beginning of May surface $c_{raw}(\text{Chl } a)$ reached 3 mg m$^{-3}$. The second period of increased
Chl $a$ inventory in June lasted longer and $c_{raw}(\text{Chl } a)$ increased to 4 mg m$^{-3}$. Between the two periods of
increased Chl $a$ inventory $N(O_2)$ and $N(C_T)$ had negative values indicating that remineralisation of the high Chl
$a$ inventory material was a dominant process during this period. Even though they are uncalibrated, the spring
period of increased Chl $a$ inventory $c_{raw}(\text{Chl } a)$ values are in agreement with the study of Rey (2001) who found



$c_{raw}$(Chl $a$) = 3 mg m$^{-3}$ at the beginning of May. The largest period of increased Chl $a$ inventory when the top 50
m $\theta$ increased from 7 °C to 11 °C and $z_{mix}$ shoaled from 200 m to 20 m. During this period, $c(O_2)$ reached a
summer maximum of 340 µmol kg$^{-1}$ and $C_T$ decreased to the summer minimum at 2070 µmol kg$^{-1}$. In both cases,
the main components of the $N$ changes were the inventory and air-sea flux, while the smallest driver was the
entrainment. Also, the glider sampled two different water masses characterised by different $C_T$ and $c(O_2)$. This
led to smaller values of $N$ in NCC compared to NwAC.
Table 3 shows estimates of net community production ($N$) in the Norwegian Sea (Falck and Anderson, 2005;
Falck and Gade, 1999; Kivimäe, 2007; Skjelvan et al., 2001). All these studies used low-resolution datasets in
space and time. These datasets had data collected by several cruises in different years (e. g. 1955 to 1988 (Falck
and Gade, 1999)) in all the Norwegian Sea. The estimated $N$ in the 4 studies varies from 2.0 to 8.6 mol m$^{-2}$ a$^{-1}$
for $N(C_T)$ and from 2.6 to 11.1 mol m$^{-2}$ a$^{-1}$ for $N(O_2)$. In our study we obtained an annual $N$ in agreement with
these studies, with a $N(O_2)$ of 10 mol m$^{-2}$ a$^{-1}$ and a $N(C_T)$ of 3.3 mol m$^{-2}$ a$^{-1}$. The annual $N(C_T)$ and $N(O_2)$ that we
calculated is most likely an overestimation because it is ignoring the winter and autumn months where $N$ is
lower. In fact, for the Nordic Seas Falck and Gade (1999) found a negative $N(O_2)$ from October to March.
Some of the previous $N(C_T)$ estimates derived $C_T$ from other variables such as $c(O_2)$, $c(PO_4^{3-})$, $c(NO_3^-)$,
assuming Redfield ratios. Our $N(C_T)$ estimate was 3.3 mol m$^{-2}$ a$^{-1}$ and is similar to 3.4 mol m$^{-2}$ a$^{-1}$ estimated by
Falck and Anderson (2005) who used $C_T$ samples directly. The difference between our $N(C_T)$ and other studies
is likely due to their use of the Redfield ratio assumption (Redfield, 1963) to convert $N(O_2)$ to $N(C_T)$. The
carbon/nutrients ratios vary between water masses and during photosynthesis (Copin-Montégut, 2000;
Körtzinger et al., 2001; Osterroht and Thomas, 2000; Thomas et al., 1999). In deep waters, the release ratios
vary for $C_T$, $c(PO_4^{3-})$, $c(NO_3^-)$ and $c(O_2)$ leading to different concentrations than the traditional Redfield ratio
(Hupe, 2000; Hupe and Karstensen, 2000; Minster and Boulahdid, 1987; Shaffer, 1996). For example, during
remineralisation, $NO_3^-$ and $PO_4^{3-}$ are released faster than $C_T$ leading to a C:P remineralisation ratio of 90 ± 15 at
the base of the euphotic zone to about 125 ± 10 from to 1000 m to the bottom (Shaffer, 1996).
The difference of $N(O_2)$ is caused by the yearly variability of $N$ in the Norwegian Sea. In fact, Kivimäe (2007)
saw an annual variability of $N(O_2)$ from 1955 to 2005 of 4.7 mol m$^{-2}$ a$^{-1}$ to 18.3 mol m$^{-2}$ a$^{-1}$ and of $N(C_T)$ of 3.6
mol m$^{-2}$ a$^{-1}$ to 14.0 mol m$^{-2}$ a$^{-1}$. In order to understand what is causing these interannual changes, it is important
to use available high-resolution datasets. Also, this study showed that the Norwegian Sea spring and summer $N$
is strongly affected by time and location. For that reason, $N$ estimated from low-resolution datasets make the




result strongly dependant on the time and place of sampling. To quantify this interannual variability in $N$, more
high-resolution studies are needed.
**5 Conclusions**
This study was, to the best of our knowledge, the first glider deployment of a $CO_2$ optode. During the
deployment, the optode performance was affected by drift, lag, lack of sampling in the top 150 m after dive 400
(the 24 July 2014), and poor default calibration. We found that the optode response was better correlated with
$c(CO_2)$ than $p(CO_2)$. Nevertheless, the optode was able to capture the spatial and temporal changes in the
Norwegian Sea after recalibration with discrete samples collected along the glider section and nearby at OWSM
during the deployment.
$C_T$ estimated from glider data had a standard deviation of 10 µmol kg$^{-1}$ and a mean bias of 1.5 µmol kg$^{-1}$
compared with the discrete samples, while the $CO_2$ fugacity $f(CO_2)$ had a mean bias of ($8\pm23$) µatm. The dataset
was used to calculate net community production $N(O_2)$ and $N(C_T)$ from inventory changes, air-sea flux, and
entrainment. The two $N$ values had maxima during the summer period of increased Chl $a$ inventory of $N(C_T)$ =
($64\pm67$) mmol m$^{-2}$ d$^{-1}$ and $N(O_2)$ = ($166\pm75$) mmol m$^{-2}$ d$^{-1}$. At the beginning of April, we sampled a smaller
spring period of increased Chl $a$ inventory with a $N(C_T)$ = ($23\pm5$) mmol m$^{-2}$ d$^{-1}$ and $N(O_2)$ = ($61\pm14$) mmol m$^{-2}$
d$^{-1}$. After the period of increased Chl $a$ inventory, $N(C_T)$ decreased due to remineralisation to ($-80\pm107$) mmol
m$^{-2}$ d$^{-1}$, and $N(O_2)$ to ($-15\pm27$) mmol m$^{-2}$ d$^{-1}$. The glider monitored two water masses (NwAC and NCC). The
NCC-influenced one was characterised by a lower $c(O_2)$ and $C_T$ than the NwAC region. $N(O_2)$ decreased to ($-$
$43\pm127$) mmol m$^{-2}$ d$^{-1}$ driven by a decrease of $c(O_2)$ under 30 m from 300 to 290 µmol kg$^{-1}$ and increased for
$N(C_T)$ to ($75\pm58$) mmol m$^{-2}$ d$^{-1}$. In particular, the $N(O_2)$ changes were driven by the surface oxygen
supersaturation making the seawater a source of oxygen. In contrast, the ocean was a sink to inorganic carbon
during the summer, with a continuous $CO_2$ flux from the atmosphere into the water.
This deployment shows the potential of using small, low energy consuming $CO_2$ optodes on autonomous
observing platforms like Seagliders to quantify the interactions between biogeochemical processes and the
marine carbonate system at high spatiotemporal resolution.
*Acknowledgements.* Luca Possenti's PhD project is part of The Next Generation Unmanned Systems Science
(NEXUSS) Centre for Doctoral Training which is funded by the Natural Environment Research Council
(NERC) and the Engineering and Physical Science Research Council (EPSRC) [grant number NE/N012070/1].
We would like to thank the scientists, engineers, and crew that contributed to the glider mission and data
collection, as well as Michael Hemming and Bastien Queste for their initial contributions to the data analysis.





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
