# Peer review of "Norwegian Sea net community production estimated from O2"

_Ocean Science, 2020_

## Referee Comment (RC1) · Anonymous Referee #1 · 1 Sep 2020

Summary

The authors present oxygen, CO2, and chlorophyll fluorescence measurements obtain from a glider occupation of a transect in the Norwegian Sea. The observations are utilized to compute net community production (NCP) along the transect via temporal budgets of oxygen and DIC (calculated from CO2 measurements and an algorithmic estimate of total alkalinity). The data are novel as they may be the first collected from a CO2 optode deployed on a glider. As such, the results of this study have important implications for enhancing biogeochemical observations using autonomous measurements and, therefore, long-term observation capabilities of biological production and

ocean acidification. Thus, this study provides a potentially important contribution to the scientific literature. Given the importance of this study, the methods of sensor calibration and NCP calculation should be carefully scrutinized. There are some questions and concerns regarding the calibration and correction of the O2 and CO2 data. There are also a number of major concerns regarding the calculation of net community production that the authors must address and clarify in order to minimize doubt and error; chief among these concerns in an apparent lack of attempt to separate spatial and temporal variability in the O2 and CO2 measurements.

General comments

1) Parameterization for deriving phosphate and silicate concentrations along the glider track from 'spot' samples collected during four cruises over the deployment (March, May, June, and October). Sampling restricted to the southern half of the transect. And yet, the uncertainties were only 1.3 and 0.13 umol kg-1 for silicate and phosphate? I hope this parameterization is discussed in detail (in the text or an appendix). I also hope that some sensitivity analysis was completed regarding the impact of differing nutrient concentrations (within a reasonable range for the region & study period) on CO2SYS calculations. I'm also concerned about the use of chloroform to preserve nutrient samples.

2) For the lag correction to the CO2 optode, data from the glider ascents are compared against those from descents. However, there is significant horizontal distance between a glider ascent and descent, unlike what one might expect for a CTD cast from a ship. By minimizing the differences observed between glider ascents & descents, you are loosing information and I'm not sure the lag correction is necessarily reliable. I would suggest comparing potential temperature and salinity in glider ascents & descents. Do they match? If so, then perhaps this method is OK. If not, the authors may need to revise the lag correction method.

3) Why is the correlation between the discrete samples and optode output CO2 partial

pressure (Figure 6) so much better when using CO2 concentration vs. partial pressure (from the discrete samples)? The authors should at least offer some educated guesses or speculation.

4) I am concerned about the potential impact of advection on the NCP calculation. The study focuses on a SE-NW transect, in a region where waters are transported in a meridional direction along well known currents (NwAC and NCC, as shown in Fig. 1). Can the authors be certain that the time rate of change in O2 and DIC does not reflect advection of water through the transect? What steps did the authors take to ensure that changes in O2 and DIC were truly a function of time and not space? Differentiating temporal vs. spatial changes in measured variables from gliders is not a trivial task and prior studies have typically used repeating spatial patterns to form a 'box' in order to compute O2 and/or carbon budgets for the estimation of NCP. In this study, the glider did not survey a box but a transect in a region of potentially meandering currents and a frontal region separating two water mass regimes. The authors need to do a better job justifying their methods and eliminating (or at least minimizing) doubt that spatial variations and/or advection contribute significantly to the observed changes in oxygen over the study period.

5) The authors indicate a separation of the NCP calculation based on water masses with a cutoff at S = 35 that distinguishes between the two primary water masses influencing the study area: Norwegian Atlantic Current (NwAC) water and the Norwegian Coastal Current (NCC) water. It is also stated that salinities between 32 and 34 were encountered in the top 50 m, signifying influence of NCC water. I'm curious whether the authors took mixing into account between the two water masses in the region where NCC was encountered. How might this impact the NCP calculations? Also, I would have appreciated more information regarding the separation. Was NCP calculated separately for each of the two regions? Were they then averaged together to present a single NCP number for O2 and DIC?

6) Integration of oxygen & DIC over a specific depth range for the calculation of NCP

may be subject to vertical heaving of isopycnals. What steps did the authors take to ensure that such vertical displacement did not impact the calculations? What about vertical mixing from the bottom up? The authors calculate an entrainment flux that focuses on periods when the mixed layer depth exceeded the limit of integration (45 m), but do not discuss the possibility of mixing across the bottom boundary. Admittedly, this is probably minimal, unless there were periods of isopycnal heaving (which looks probable, from the temperature distribution shown in Fig. 8), but the possibility should have been investigated and (at least briefly) mentioned in the manuscript.

Specific comments

1) Please clarify units of N(CT) and N(O2). Are they both expressed as mmol C m-2 d-1 or do they differ (e.g., mmol C m-2 d-1 vs. mmol O2 m-2 d-1)? After getting to section 3.6, it's clear they were reported in different units, but readers shouldn't have to wait that long to be sure.

2) Preservation of nutrient samples with chloroform is not a recommended procedure. . .

3) Figure 2 indeed shows that, on average, the oxygen concentration at higher latitudes was greater (by 10-15 umol kg-1) than those measured at lower latitudes. However, the oxygen concentration decreases fairly linearly with time in both regions (lower and higher latitudes). Why is this the case? I wouldn't think it was short-term drift as such drift should be minimal in oxygen optodes. Does this results, perhaps, from a longitudinal gradient in oxygen concentrations?

Figure 3 shows a similar 'drift', or time rate of change, in the gain factor computed to correct the optode oxygen. I am surprised there is such an apparent, continued drift in the optode sensor response. I would have expected a large, initial drift ('storage' drift') but then would have thought the optode response to be relatively stable over a deployment period of ∼8 months. Can the authors show the individual, median oxygen concentrations and standard deviations from the discrete data? I'm curious how stable

the oxygen concentrations are in this density/depth range (∼427 to 1000 m).

4) Line 269: "The thermal lag of the glider conductivity sensor was corrected for..." What?

5) Can the authors please define cN(Chl a)? Is this the computed chlorophyll concentration, using factory-defined coefficients?

6) Line 363: "...because after this dive, the CO2 optode stopped sampling..."

7) Line 364: "...raw c(O2) data was calibrated and drift-corrected and c(CO2) was drift- and lag-corrected and recalibrated, then used to..."

I'm not going to focus my review on grammar corrections, so I suggest the authors carefully re-read the manuscript to avoid any additional grammar or spelling mistakes that should be addressed prior to publication.

8) Plot isopycnals on panels of Fig. 8. I'd also recommend plotting the mixed layer depth and highlighting zlim (dotted line?).

9) Line 375: What is "against year-day"? Please re-word this sentence.

10) Lines 456-457: Can the authors please expand on how NCP was calculated? It is stated that, "The two Ns were calculated as the difference in inventory changes between two transects when the glider was in the same water mass." Two transects? So, is one transect equivalent to the glider moving over the entire transect in one direction and the second transect is the glider moving back over the transect in the opposite direction?

Is the NCP calculated only for the NwAC water mass? So any changes within the NCC water mass are removed from the analysis?

11) It is important to compare NCP estimates with those of previous studies; however, it is difficult to know how comparable the numbers are in Table 3 because it is not clear where in the Norwegian Sea these various studies took place. It is also difficult because

zlim varies largely among the studies. The fact that three of the four compared studies used zlim >= 100 m also calls into question why exactly the current study decided on zlim = 45 m, particularly since the mixed layer depth varied so largely and often exceeded zlim.

─────────────────────────────

---

## Referee Comment (RC2) · Anonymous Referee #2 · 28 Sep 2020

This manuscript provides a detailed description of how net community production in the Norwegian Sea was calculated from glider data. Sensor calibration and drift correction are clearly presented. Therefore, I think this work is helpful to future net community production studies using glider data. Overall, this manuscript is well-organized. However, there are some short paragraphs that contain only one sentence. I suggest the authors consider re-organize some of the paragraphs.

My major concern is about oxygen optode calibration. It is unfortunate that discrete oxygen samples were not collected. But I am not convinced that archived oxygen data dated back to 2000 are suitable to be used for calibration even for deep water. The

authors may justify this by demonstrating that the changes in archived oxygen data over the past 20 years are minor. Otherwise, the most recent discrete oxygen data should be used for calibration.

One major advantage of glider is that it can survey the entire water column continuously. However, the major portion (sections 3.3-3.6) of the results section is on NCP data at an integration depth of zlim = 45 m (figures 14-16). This compromises the importance of using glider data.

Below are a few minor comments,

Lines 68-79, I think these two paragraphs belong to the method section.

Line 348, it should be k(CO2) rather than k(O2) in equation 14.

Line 609, change "a sink to" to "a sink of"

Figures 2, 3, 8, 9 Date on the x-axis is kind of misleading. It seems like Jan-04, Jan-05, etc. I think it is better to change 01/04, 01/05, . . ., 01/10 to April, May, . . ., October.

---

## Author Comment (AC1) · 9 Dec 2020

We would like to thank Reviewer 2 for providing constructive and insightful comments. We will incorporate their suggestions into a revised manuscript. Reviewer 1's comments have been reproduced below in black, with the authors' response in blue.

General comments

However, there are some short paragraphs that contain only one sentence. I suggest the authors consider re-organize some of the paragraphs. To solve this problem we merged section 2.5 and 2.6 that now is called: "CO2 optode lag and drift correction

and calibration" and section 3.1 and 3.2 than now is: "O2 and CO2 optode calibration".

My major concern is about oxygen optode calibration. It is unfortunate that discrete oxygen samples were not collected. But I am not convinced that archived oxygen data dated back to 2000 are suitable to be used for calibration even for deep water. The authors may justify this by demonstrating that the changes in archived oxygen data over the past 20 years are minor. Otherwise, the most recent discrete oxygen data should be used for calibration.

In the revised version we will add in the appendices Figure 1, the figure shows that the oxygen discrete samples variability is within the variability of the measured oxygen by the glider. For that reason, the interannual and seasonal variability of the discrete samples can be considered minimal. See also reply 9 to Reviewer 1. Also, Figure 2 shows that the oxygen discrete samples in this water mass do not change between 2000 and 2010. For example, c(O2) in 2000 varied from 299.5 to 314.3 $\mu$mol kg-1 and in 2009 from 300.6 to 312.7 $\mu$mol kg-1. For that reason, we can consider the oxygen concentration constant during the years and can be used as reference to correct the oxygen optode.

One major advantage of glider is that it can survey the entire water column continuously. However, the major portion (sections 3.3-3.6) of the results section is on NCP data at an integration depth of zlim = 45 m (figures 14-16). This compromises the importance of using glider data.

In all the manuscript we correct and show the entire profiles for oxygen, CO2, temperature, salinity and chlorophyll. The net community production was calculated using an integration depth of 45 m because it was the mean depth of the euphotic zone. The two Ns were 4.6 and 0.5 mol m-2 a-1 for N(O2) and N(CT), respectively. For comparison, we calculated net community production using integration depths of 30 and 100 m. The derived net community production was the same for N(O2) at the different integration depths and similar for N(CT). In particular, N(CT; 30 m) was 0.6 mol m-2 a-1 and N(CT;
100 m) was -0.04 mol m-2 a-1; N(O2; 30 m) was 4.6 mol m-2 a-1 and N(O2; 100 m) was 4.3 mol m-2 a-1.

Specific comments

Lines 68-79, I think these two paragraphs belong to the method section. We moved the two paragraphs to the method section, creating a new introductory section (2.8) to the net community production.

Line 348, it should be k(CO2) rather than k(O2) in equation 14. We changed the equation replacing k(O2) with k(CO2).

Line 609, change "a sink to" to "a sink of" We changed "a sink to" to "a sink of".

Figures 2, 3, 8, 9 Date on the x-axis is kind of misleading. It seems like Jan-04, Jan-05, etc. I think it is better to change 01/04, 01/05, . . ., 01/10 to April, May, . . ., October. We changed the date to the suggested date format in figures: 2, 3 and 6 to 16.

Please also note the supplement to this comment:
https://os.copernicus.org/preprints/os-2020-72/os-2020-72-AC1-supplement.pdf
* * *
[Figure]

**Fig. 1.**

**Fig. 2.**

**Supplement:**

**Response to Reviewer 2**

We would like to thank Reviewer 2 for providing constructive and insightful comments. We will incorporate their suggestions into a revised manuscript. Reviewer 1's comments have been reproduced below in black, with the authors' response in blue.

**General comments**

However, there are some short paragraphs that contain only one sentence. I suggest the authors consider re-organize some of the paragraphs.

To solve this problem we merged section 2.5 and 2.6 that now is called: "$CO_2$ optode lag and drift correction and calibration" and section 3.1 and 3.2 than now is: "$O_2$ and $CO_2$ optode calibration"

My major concern is about oxygen optode calibration. It is unfortunate that discrete oxygen samples were not collected. But I am not convinced that archived oxygen data dated back to 2000 are suitable to be used for calibration even for deep water. The authors may justify this by demonstrating that the changes in archived oxygen data over the past 20 years are minor. Otherwise, the most recent discrete oxygen data should be used for calibration.

In the revised version we will add in the appendices Figure 1, the figure shows that the oxygen discrete samples variability is within the variability of the measured oxygen by the glider. For that reason, the interannual and seasonal variability of the discrete samples can be considered minimal. See also reply 9 to Reviewer 1.

[Figure]

**Figure 1:** a) Discrete samples $c_C(O_2)$ (yellow), raw glider oxygen $c_G(O_2)$ (blue) and drift corrected glider oxygen $c_{G,cal}(O_2)$ (red) using water density > 1028 kg m$^{-3}$.

Also, Figure 2 shows that the oxygen discrete samples in this water mass do not change between 2000 and 2010. For example, $c(O_2)$ in 2000 varied from 299.5 to 314.3 µmol kg$^{-1}$ and in 2009 from 300.6 to 312.7 µmol kg$^{-1}$. For that reason, we can consider the oxygen concentration constant during the years and can be used as reference to correct the oxygen optode.

[Figure]

**Figure 2:** oxygen discrete samples used as reference to calibrate the oxygen optode output. All the samples were collected in the latitude and longitude range of the deployment area for a water density > 1028 kg m⁻³.

One major advantage of glider is that it can survey the entire water column continuously. However, the major portion (sections 3.3-3.6) of the results section is on NCP data at an integration depth of zlim = 45 m (figures 14-16). This compromises the importance of using glider data.

In all the manuscript we correct and show the entire profiles for oxygen, $CO_2$, temperature, salinity and chlorophyll. The net community production was calculated using an integration depth of 45 m because it was the mean depth of the euphotic zone. The two $N$s were 4.6 and 0.5 mol m⁻² a⁻¹ for $N(O_2)$ and $N(C_T)$, respectively. For comparison, we calculated net community production using integration depths of 30 and 100 m. The derived net community production was the same for $N(O_2)$ at the different integration depths and similar for $N(C_T)$. In particular, $N(C_T; 30$ m$)$ was 0.6 mol m⁻² a⁻¹ and $N(C_T; 100$ m$)$ was -0.04 mol m⁻² a⁻¹; $N(O_2; 30$ m$)$ was 4.6 mol m⁻² a⁻¹ and $N(O_2; 100$ m$)$ was 4.3 mol m⁻² a⁻¹.

**Specific comments**

Lines 68-79, I think these two paragraphs belong to the method section.

We moved the two paragraphs to the method section, creating a new introductory section (2.8) to the net community production.

Line 348, it should be k(CO2) rather than k(O2) in equation 14.

We changed the equation replacing $k(O_2)$ with $k(CO_2)$.

Line 609, change "a sink to" to "a sink of"

We changed "a sink to" to "a sink of".

Figures 2, 3, 8, 9 Date on the x-axis is kind of misleading. It seems like Jan-04, Jan-05, etc. I think it is better to change 01/04, 01/05, . . ., 01/10 to April, May, . . ., October.

We changed the date to the suggested date format in figures: 2, 3 and 6 to 16.

---

## Author Comment (AC2) · 9 Dec 2020

We would like to thank Reviewer 1 for providing constructive and insightful comments. We will incorporate their suggestions into a revised manuscript. Reviewer 1's comments have been reproduced below in black, with the authors' response in blue.

General comments

Parameterization for deriving phosphate and silicate concentrations along the glider track from 'spot' samples collected during four cruises over the deployment (March, May, June, and October). Sampling restricted to the southern half of the transect. And

yet, the uncertainties were only 1.3 and 0.13 umol kg-1 for silicate and phosphate? I hope this parameterization is discussed in detail (in the text or an appendix). I also hope that some sensitivity analysis was completed regarding the impact of differing nutrient concentrations (within a reasonable range for the region & study period) on CO2SYS calculations. I'm also concerned about the use of chloroform to preserve nutrient samples.

The reviewer's comment highlighted that our method description has been too brief on nutrient analysis. In fact, we collected 58 discrete samples along the glider transect in March, May, June and October, which were analysed for total alkalinity (AT), dissolved inorganic carbon (CT) as well as nutrient concentrations. In addition, we used 52 CT and AT discrete samples collected at a weather station (OWSM) located 270 km northeast of the glider transect (Figure 1 of the discussion paper) in March, May, June, August and November. For the latter samples, no nutrient measurements were undertaken. Instead, we filled in these gaps with the nutrient concentration nearest in time and depth from the 58 along-transect samples, which we expected to be sufficiently close to the actual concentration at OWSM. We then derived the CO2 concentration (c(CO2)) using the MATLAB toolbox CO2SYS (Van Heuven et al., 2011), using the phosphate and silicate concentrations to account for their contributions to AT. This c(CO2) was used as reference to calibrate the glider output (Figure 1, in red the samples collected along the transect and in black at OWSM). The nutrient concentrations were not used in any further calculations. To assess the uncertainty of the final silicate and phosphate concentration we calculated the uncertainty in the calculation of the OWSM c(CO2) using the interpolation uncertainty 1.1 and 0.12 $\mu$mol kg-1 for the silicate and phosphate concentration, respectively. The uncertainty was calculated as the root mean square error of the interpolation, which estimates the standard deviation of the error distribution. See the interpolation for phosphate and silicate in the supplement PDF. The derived nutrient concentration led to a mean error of 0.04 $\mu$mol kg-1 in the calculation of c(CO2). We followed the procedure described by Hagebo and Rey (1984) to collect and preserve the nutrients. We are not aware of any problems

with the use of chloroform to preserve nutrient samples. The 2019 GO-SHIP Repeat Hydrography Nutrient Manual only discourages the use of acid (which would require neutralisation before analysis) and mercuric chloride (a long-term environmental hazard) (Becker et al., 2020).

For the lag correction to the CO2 optode, data from the glider ascents are compared against those from descents. However, there is significant horizontal distance between a glider ascent and descent, unlike what one might expect for a CTD cast from a ship. By minimizing the differences observed between glider ascents & descents, you are loosing information and I'm not sure the lag correction is necessarily reliable. I would suggest comparing potential temperature and salinity in glider ascents & descents. Do they match? If so, then perhaps this method is OK. If not, the authors may need to revise the lag correction method.

The CO2 optode lag was corrected using the algorithm of Miloshevich, (2004) that uses the sensor response time ($\tau$). The $\tau$ was calculated minimising the difference between each glider ascent and descent (see Figure 5 of the discussion paper). In the equation, we used the median of all the $\tau$ values (1384 s). The correction decreased the difference between the glider and descent of the raw p(CO2) from (71$\pm$30) to (21$\pm$26) $\mu$atm. Using the median $\tau$ is a robust indicator for the lag time, even there were short-term variations (e.g. due to internal tides or waves) between the descent and ascent. To assess if the glider was in the same water mass between ascent and descent we looked at potential temperature and salinity. The mean difference between the descent and ascent was (0.13$\pm$0.33) $^\circ$C for potential temperature and 0.02$\pm$0.04 for salinity. These small mean differences for potential temperature and salinity show that the method is sufficiently robust.

Why is the correlation between the discrete samples and optode output CO2 partial pressure (Figure 6) so much better when using CO2 concentration vs. partial pressure (from the discrete samples)? The authors should at least offer some educated guesses or speculation.

The better correlation with c(CO2) was probably related due to an inadequate temperature-parameterisation of the sensor calibration function. The sensor output depends on the changes in pH that are directly related to the changes of c(CO2) in the membrane and – indirectly – p(CO2), via Henry's Law. Sensor and external water p(CO2) should be in equilibrium. The calibration is supposed to correct for the temperature-dependence of the sensor output (Atamanchuk et al, 2014). The observation that the sensor output correlated better with c(CO2) than p(CO2) is perhaps due to a fortuitous cancellation of an inadequate temperature-parameterisation and the temperature-dependence of the Henry's Law relationship between c(CO2) than p(CO2).

I am concerned about the potential impact of advection on the NCP calculation. The study focuses on a SE-NW transect, in a region where waters are transported in a meridional direction along well known currents (NwAC and NCC, as shown in Fig. 1). Can the authors be certain that the time rate of change in O2 and DIC does not reflect advection of water through the transect? What steps did the authors take to ensure that changes in O2 and DIC were truly a function of time and not space? Differentiating temporal vs. spatial changes in measured variables from gliders is not a trivial task and prior studies have typically used repeating spatial patterns to form a 'box' in order to compute O2 and/or carbon budgets for the estimation of NCP. In this study, the glider did not survey a box but a transect in a region of potentially meandering currents and a frontal region separating two water mass regimes. The authors need to do a better job justifying their methods and eliminating (or at least minimizing) doubt that spatial variations and/or advection contribute significantly to the observed changes in oxygen over the study period.

We have assumed that the main processes controlling the surface dissolved inorganic carbon and oxygen concentrations are biological production and respiration as well as air-sea gas exchange and vertical transport. Even though there are well-known currents, horizontal gradients are reduced due to constant stirring from winds and tides

and therefore net advective fluxes are likely to be small (Gislefoss et al., 1998; Falck and Gade, 1999). Previous estimates of net community production in the Norwegian Sea have also neglected advective fluxes (Falck and Gade, 1999; Skjelvan, Falck, Leif G. Anderson, et al., 2001; Falck and Anderson, 2005; Kivimäe, 2007). For example, Gislefoss et al, (1998) considered minimal the effect of horizontal advection on N(CT) during the summer because CT changes were largely controlled by biology and air-sea interactions. For NCP estimates on shorter timescales (days to a few weeks), advective fluxes and water-mass movement would have to be taken into account, but this would require a different survey design, involving multiple platforms (Alkire et al., 2014), beyond the scope of the present study. However, the query from the reviewer prompted us to revisit our NCP calculation, which showed that at the glider turn-around points, inventory changes were calculated over relatively short time-scales of a few days. Therefore, to minimise the effect of horizontal advection in the new version of the manuscript we will extend the time interval used to calculate the inventory changes from less than a week to an average of 50 days. This was achieved by calculating the concentration difference between two transects when the glider moved in the same direction (e.g. transects 1-3, 2-4 and 3-5 all in N-S direction) instead of two consecutive transects.

The authors indicate a separation of the NCP calculation based on water masses with a cutoff at S = 35 that distinguishes between the two primary water masses influencing the study area: Norwegian Atlantic Current (NwAC) water and the Norwegian Coastal Current (NCC) water. It is also stated that salinities between 32 and 34 were encountered in the top 50 m, signifying influence of NCC water. I'm curious whether the authors took mixing into account between the two water masses in the region where NCC was encountered. How might this impact the NCP calculations? Also, I would have appreciated more information regarding the separation. Was NCP calculated separately for each of the two regions? Were they then averaged together to present a single NCP number for O2 and DIC?

The daily value of the net community production (N) was calculated separately for NCC and NwAC. The annual N was then calculated combining the two water masses to be consistent with the previous studies (Falck and Gade, 1999; Skjelvan, Falck, Leif G Anderson, et al., 2001; Falck and Anderson, 2005; Kivimäe, 2007). Here again, the reviewer's comment inspired a revision of our calculation method and for the new version of the paper, we will calculate daily and annual N without separating NCC from NwAC. We change the methodology to be consistent with previous studies, to extend and homogenise the time difference used to calculate inventory and entrainment and to minimise the impact of horizontal advection.

Integration of oxygen & DIC over a specific depth range for the calculation of NCP may be subject to vertical heaving of isopycnals. What steps did the authors take to ensure that such vertical displacement did not impact the calculations? What about vertical mixing from the bottom up? The authors calculate an entrainment flux that focuses on periods when the mixed layer depth exceeded the limit of integration (45 m), but do not discuss the possibility of mixing across the bottom boundary. Admittedly, this is probably minimal, unless there were periods of isopycnal heaving (which looks probable, from the temperature distribution shown in Fig. 8), but the possibility should have been investigated and (at least briefly) mentioned in the manuscript.

We thank the reviewer for the interesting suggestion. Vertical heaving has an effect entraining the water from below the integration. We do already consider the effect of entrainment in the calculation of the net community production, in the form of terms $E(O_2)$ (Eq. 10) and $E(CT)$ (Eq. 16). In response to the reviewer's comment, we have also estimated the diapycnal mixing flux and will incorporate this into the new version of the manuscript. The diapycnal mixing flux ($F_v$) was calculated from the vertical oxygen concentration gradient. In the calculation, we used a vertical eddy diffusivity ($K_z$) of 10–5 m s-2 derived for the Nordic Seas by Naveira Garabato et al. (2004). The effect of $F_v$ for $O_2$ and $CT$ was calculated at $z_{mix}$ when it was deeper than the integration depth $z_{lim}$ and at $z_{lim}$ when $z_{mix}$ was shallower than $z_{lim}$. See the

equations in the supplement PDF. A positive sign of Fv(CT) means a decrease of the dissolved inorganic concentration in the layer of interest between surface and zlim; a negative sign corresponds to an increase. In the new version of the manuscript, we will add a new Figure 2 that shows Fv as a function of time during the glider deployment. The results show that Fv is negligibly small: Fv(CT) = (0.05±0.3) mmol m-2 d-1 and (-0.02±0.33) mmol m-2 d-1 for O2. For that reason, diapycnal mixing will not be used to calculate N.

Specific comments

Please clarify units of N(CT) and N(O2). Are they both expressed as mmol C m-2 d-1 or do they differ (e.g., mmol C m-2 d-1 vs. mmol O2 m-2 d-1)? After getting to section 3.6, it's clear they were reported in different units, but readers shouldn't have to wait that long to be sure.

Both N(O2) and N(CT) are expressed in mmol m-2 d-1. In the case of N(O2), this is a flux of O2, in the case of N(CT), a flux of inorganic carbon.

Preservation of nutrient samples with chloroform is not a recommended procedure. . .

See our answer on page 1.

Figure 2 indeed shows that, on average, the oxygen concentration at higher latitudes was greater (by 10-15 umol kg-1) than those measured at lower latitudes. However, the oxygen concentration decreases fairly linearly with time in both regions (lower and higher latitudes). Why is this the case? I wouldn't think it was short-term drift as such drift should be minimal in oxygen optodes. Does this results, perhaps, from a longitudinal gradient in oxygen concentrations? Figure 3 shows a similar 'drift', or time rate of change, in the gain factor computed to correct the optode oxygen. I am surprised there is such an apparent, continued drift in the optode sensor response. I would have expected a large, initial drift ('storage' drift) but then would have thought the optode response to be relatively stable over a deployment period of around 8 months. Can the

authors show the individual, median oxygen concentrations and standard deviations from the discrete data? I'm curious how stable the oxygen concentrations are in this density/depth range (around 427 to 1000 m).

The oxygen concentrations for $\sigma 0 > 1028$ kg m−3 decreased linearly in both regions because the oxygen optode drifted continuously during the deployment (Figure 2 of the paper and Figure 3 where in red is the uncorrected oxygen, in blue the corrected oxygen and in yellow the discrete sample used as reference). In the new version of the manuscript, we will add a figure with all the samples collected and the glider data before and after the correction showing how the corrected glider oxygen is within the variability of the discrete samples and how stable the O2 concentration is in this depth range. It was possible to use waters of these potential densities because were always well below the mixed layer depth and therefore subject to limited seasonal and interannual variability. The salinity of the discrete samples varied from 34.88 to 34.96, with a mean of (34.90±0.01) and the temperature varied from 0.45 to -0.76 °C with a mean of (-0.15±0.36) °C. Variations are due to differences in deep-water masses. Therefore, we only used the glider and discrete samples collected at latitudes north of 64° N because this reflects the largest part of the transect. Also, the region south of 64° N contained just 5 days of archived samples. See also reply 2 to Reviewer 2. We added in the appendices a plot with all the discrete samples and the glider oxygen before and after the correction.

Line 269: "The thermal lag of the glider conductivity sensor was corrected for. . ." What?

The correct phrase should be "The thermal lag of the glider conductivity sensor was corrected using the method of Gourcuff (2014)."

Can the authors please define cN(Chl a)? Is this the computed chlorophyll concentration, using factory-defined coefficients?

Yes, in the conversion from the raw chlorophyll to the chlorophyll concentration, we

used the factory-based coefficients.

Line 363: ". . .because after this dive, the CO2 optode stopped sampling. . ."

We meant "For the subsequent dives, the CO2 optode stopped sampling in the first 150 m (Figure 2.8d)."

Line 364: ". . .raw c(O2) data was calibrated and drift-corrected and c(CO2) was drift-tand lag-corrected and recalibrated, then used to. . ." I'm not going to focus my review on grammar corrections, so I suggest the authors carefully re-read the manuscript to avoid any additional grammar or spelling mistakes that should be addressed prior to publication.

Apologies if the sentence structure was unclear. We meant to say that "The raw c(O2) data were drift-corrected and calibrated. The CO2 output was drift and lag-corrected and then calibrated against cC(CO2) from nearby discrete samples. The calibrated glider cG(O2) and cG(CO2) were used to calculate inventory changes and air-sea exchange fluxes ($\Phi$) to evaluate the net community production changes."

Plot isopycnals on panels of Fig. 8. I'd also recommend plotting the mixed layer depth and highlighting zlim (dotted line?).

We changed figure 8 adding the mixed layer depth, zlim and the isopycnals (Figure 4).

Line 375: What is "against year-day"? Please re-word this sentence.

Year-day means day of the year and varies from 1 to 365. We will change all occurrences of year-day in the manuscript to "day of the year".

Lines 456-457: Can the authors please expand on how NCP was calculated? It is stated that, "The two Ns were calculated as the difference in inventory changes between two transects when the glider was in the same water mass." Two transects? So, is one transect equivalent to the glider moving over the entire transect in one direction and the second transect is the glider moving back over the transect in the opposite direction? Is the NCP calculated only for the NwAC water mass? So any changes within the NCC water mass are removed from the analysis?

Yes, it is correct, that we used one transect with the glider moving in one direction and the following transect with the glider moving in the opposite direction. To calculate the net community production (N) the data were binned into 0.1° latitude intervals and the inventory changes were calculated as the difference of the integrated c(O2) and CT every time the glider was in the latitude bin. The air-sea flux was the instantaneous flux when the glider was in the bin and the entrainment was considered as the concentration changes when the mixed layer deepened between two transects in the same latitude bin. The daily N was calculated separating the two water masses (NCC and NwAC) and the annual N was calculated as the mean of the daily N considering the two water masses together. Following the prompt for the reviewer on the possible influence of horizontal advection (see above), the revised version of the manuscript will use an amended methodology to calculate net community production will. We will use the difference of CT and O2 between two transects when the glider moved in the same direction (e.g. southeast to northwest). We will not use two consecutive transects. This means that inventory changes will be calculated based on a similar time difference between the two samples. For two consecutive transects, the time difference between the two samples would be smaller at the beginning and the end of the transect and larger in the middle. Also, to correct for the variability of the wind speed, we will use flux-weighted gas transfer velocities for O2 and CO2 (Reuer et al, 2007), rather than instantaneous fluxes (as before). kw(O2) and kw(CO2) will be normalised using the daily wind speed in the latitude bin in the time interval used to calculate the inventory changes. The time interval is the time between two samples used to calculate the inventory changes and entrainment. The air-sea flux is based on the concentration measured at the time of the second transect used to calculate the inventory changes and the entrainment flux.

It is important to compare NCP estimates with those of previous studies; however, it

is difficult to know how comparable the numbers are in Table 3 because it is not clear where in the Norwegian Sea these various studies took place. It is also difficult because zlim varies largely among the studies. The fact that three of the four compared studies used zlim >= 100 m also calls into question why exactly the current study decided on zlim = 45 m, particularly since the mixed layer depth varied so largely and often exceeded zlim.

We used zlim = 45 m because this corresponds to the average depth of the euphotic zone, which is the region of interest for net community production from a biogeochemical and ecological point of view. Previous studies may have used zlim = 100 m for operational reasons (e.g. constrained by discrete sampling depths). To show the influence of zlim on N, we calculated N for zlim = 30 m and 100 m. N(CT; 30 m) was 0.6 and N(CT; 100 m) was –0.04 mol m-2 a-1. N(O2; 30 m) was 4.6 and N(O2; 100 m) was 4.3 mol m-2 a-1. In the case of N(CT), the derived two values are lower to the previous studies where N(CT) varied from 8.6 to 2.0 mol m-2 a-1. N(CT; 100 m) was negative because the deep integration depth included water below the euphotic zone where the remineralisation of organic matter can increase CT. This signal is not present in N(O2) because the changes were largely controlled by $\Phi$(O2) that was always positive. The calculated N(O2) is in agreement with previous studies, which gave results between 2.6 and 11 mol m-2 a-1. In the discussion, we will add a section where we explain the location and the period when the previous studies took place. Falck and Anderson (2005) used historical data from 1960 to 2000 collected in the area from 62 to 70° N and from 1991 to 1994 collected at OWSM. Skjelvan et al (2001) used data collected from 67.5° N 9° E to 71.5° N 1° E and along 74.5° N from 7 to 15° E from 1957 to 1970 and 1991 to 1998. Kivimäe (2007) used the oxygen measured at OWSM from 1955 to 2005 and Falck and Gade (1999) used data collected in all the Norwegian Sea from 1955 to 1988. The glider in the transect moved from to 66.3 °N 4 °W to 63 °N 4 °E (Figure 5 where the black dots are the glider dives, the green box the region used by Falck and Gade (1999), the yellow lines the transects used by Skjelvan et al. (2001), the azure box the region used by Falck and Anderson (2005) and the red dot the location that corresponds to the Ocean Weather Station M (OWSM) used by Kivimäe (2007)).

Bibliography

Alkire, M. B. et al. (2014) 'Net community production and export from Seaglider measurements in the North Atlantic after the spring bloom', Journal of Geophysical Research: Oceans. Wiley Online Library, 119(9), pp. 6121–6139.

Atamanchuk, D. et al. (2014) 'Performance of a lifetime-based optode for measuring partial pressure of carbon dioxide in natural waters', Limnology and Oceanography: Methods, 12(2), pp. 63–73. doi: 10.4319/lom.2014.12.63.

Becker, S. et al. (2020) 'GO-SHIP repeat hydrography nutrient manual: the precise and accurate determination of dissolved inorganic nutrients in seawater, using continuous flow analysis methods', Frontiers in Marine Science. Frontiers, 7, p. 908.

Falck, E. and Anderson, L. G. (2005) 'The dynamics of the carbon cycle in the surface water of the Norwegian Sea', 94, pp. 43–53. doi: 10.1016/j.marchem.2004.08.009.

Falck, E. and Gade, G. (1999) 'Net community production and oxygen fluxes in the Nordic Seas based on O2 budget calculations', 13(4), pp. 1117–1126.

Foltz, G. R. et al. (2003) 'Seasonal mixed layer heat budget of the tropical Atlantic Ocean', Journal of Geophysical Research: Oceans. Wiley Online Library, 108(C5).

Gislefoss, J. S. et al. (1998) 'Carbon time series in the Norwegian sea', 45, pp. 433–460.

Gourcuff, C. (2014) 'ANFOG Slocum CTD data correction', (March).

Hagebo, M. and Rey, F. (1984) 'Storage of seawater for nutrients analysis', Fisken Hav., 4, 1, 12. Van Heuven, S. et al. (2011) 'MATLAB program developed for CO2 system calculations', ORNL/CDIAC-105b. Carbon Dioxide Information Analysis Center, Oak Ridge National Laboratory, US Department of Energy, Oak Ridge, Tennessee, 530.

Kivimäe, C. (2007) 'Carbon and oxygen fluxes in the Barents and Norwegian Seas: production, air-sea exchange and budget calculations'. The University of Bergen.

Miloshevich, L. (2004) 'Development and Validation of a Time-Lag Correction for Vaisala Radiosonde Humidity Measurements', pp. 1305–1328.

Naveira Garabato, A. C. et al. (2004) 'Turbulent diapycnal mixing in the Nordic seas', Journal of Geophysical Research: Oceans. Wiley Online Library, 109(C12).

Obata, A., Ishizaka, J. and Endoh, M. (1996) 'Global verification of critical depth theory for phytoplankton bloom with climatological in situ temperature and satellite ocean color data', Journal of Geophysical Research: Oceans. Wiley Online Library, 101(C9), pp. 20657–20667.

Reuer, M. K. et al. (2007) 'New estimates of Southern Ocean biological production rates from O2/Ar ratios and the triple isotope composition of O2', Deep Sea Research Part I: Oceanographic Research Papers. Elsevier, 54(6), pp. 951–974.

Skjelvan, I., Falck, E., Anderson, Leif G, et al. (2001) 'Oxygen fluxes in the Norwegian Atlantic current', Marine chemistry. Elsevier, 73(3–4), pp. 291–303.

Skjelvan, I., Falck, E., Anderson, Leif G., et al. (2001) 'Oxygen fluxes in the Norwegian Atlantic Current', Marine Chemistry, 73(3–4), pp. 291–303. doi: 10.1016/S0304-4203(00)00112-2.

Please also note the supplement to this comment:
https://os.copernicus.org/preprints/os-2020-72/os-2020-72-AC2-supplement.pdf

[Figure]

**Fig. 1.**

[Figure]

**Fig. 2.**

[Figure]

**Fig. 3.**

[Figure]

**Fig. 4.**

[Figure]

**Fig. 5.**

**Supplement:**

**Response to Reviewer 1**

We would like to thank Reviewer 1 for providing constructive and insightful comments. We will incorporate their suggestions into a revised manuscript. Reviewer 1's comments have been reproduced below in black, with the authors' response in blue.

**General comments**

Parameterization for deriving phosphate and silicate concentrations along the glider track from 'spot' samples collected during four cruises over the deployment (March, May, June, and October). Sampling restricted to the southern half of the transect. And yet, the uncertainties were only 1.3 and 0.13 umol kg-1 for silicate and phosphate? I hope this parameterization is discussed in detail (in the text or an appendix). I also hope that some sensitivity analysis was completed regarding the impact of differing nutrient concentrations (within a reasonable range for the region & study period) on CO2SYS calculations. I'm also concerned about the use of chloroform to preserve nutrient samples.

The reviewer's comment highlighted that our method description has been too brief on nutrient analysis. In fact, we collected 58 discrete samples along the glider transect in March, May, June and October, which were analysed for total alkalinity ($A_T$), dissolved inorganic carbon ($C_T$) as well as nutrient concentrations. In addition, we used 52 $C_T$ and $A_T$ discrete samples collected at a weather station (OWSM) located 270 km northeast of the glider transect (Figure 1 of the discussion paper) in March, May, June, August and November. For the latter samples, no nutrient measurements were undertaken. Instead, we filled in these gaps with the nutrient concentration nearest in time and depth from the 58 along-transect samples, which we expected to be sufficiently close to the actual concentration at OWSM. We then derived the $CO_2$ concentration ($c(CO_2)$) using the MATLAB toolbox CO2SYS (Van Heuven *et al.*, 2011), using the phosphate and silicate concentrations to account for their contributions to $A_T$. This $c(CO_2)$ was used as reference to calibrate the glider output (Figure 1, in red the samples collected along the transect and in black at OWSM). The nutrient concentrations were not used in any further calculations.

[Figure]

**Figure 1:** Calibration of the $CO_2$ optode using the samples collected along the glider transect (red) and at OWSM (black) a) $CO_2$ concentration of the discrete samples ($c_{WS}(CO_2)$) against the glider output with the linear regression line and b) $CO_2$ partial pressure of the discrete samples ($p_{WS}(CO_2)$) against the glider output with the linear regression line.

To assess the uncertainty of the final silicate and phosphate concentration we calculated the uncertainty in the calculation of the OWSM $c(CO_2)$ using the interpolation uncertainty 1.1 and 0.12 µmol kg$^{-1}$ for the silicate and phosphate concentration, respectively. The uncertainty was calculated as

the root mean square error of the interpolation, which estimates the standard deviation of the error distribution. The interpolation was the following for phosphate:

$$c(PO_4) = 0.0003 \times z/m + -0.0003 \times t + 220.18 \tag{1}$$

where $z$ is the depth and $t$ the date. In the case of silicate the equation was:

$$c(Si) = 0.0054 \times z/m + -0.0049 \times t + 3626.3. \tag{2}$$

The derived nutrient concentration led to a mean error of 0.04 µmol kg$^{-1}$ in the calculation of $c(CO_2)$.

We followed the procedure described by Hagebo and Rey (1984) to collect and preserve the nutrients. We are not aware of any problems with the use of chloroform to preserve nutrient samples. The 2019 GO-SHIP Repeat Hydrography Nutrient Manual only discourages the use of acid (which would require neutralisation before analysis) and mercuric chloride (a long-term environmental hazard) (Becker *et al.*, 2020).

For the lag correction to the CO2 optode, data from the glider ascents are compared against those from descents. However, there is significant horizontal distance between a glider ascent and descent, unlike what one might expect for a CTD cast from a ship. By minimizing the differences observed between glider ascents & descents, you are loosing information and I'm not sure the lag correction is necessarily reliable. I would suggest comparing potential temperature and salinity in glider ascents & descents. Do they match? If so, then perhaps this method is OK. If not, the authors may need to revise the lag correction method.

The $CO_2$ optode lag was corrected using the algorithm of Miloshevich, (2004) that uses the sensor response time ($\tau$). The $\tau$ was calculated minimising the difference between each glider ascent and descent (see Figure 5 of the discussion paper). In the equation, we used the median of all the $\tau$ values (1384 s). The correction decreased the difference between the glider and descent of the raw $p(CO_2)$ from (71±30) to (21±26) µatm. Using the median $\tau$ is a robust indicator for the lag time, even there were short-term variations (e.g. due to internal tides or waves) between the descent and ascent.

To assess if the glider was in the same water mass between ascent and descent we looked at potential temperature and salinity. The mean difference between the descent and ascent was (0.13±0.33) °C for potential temperature and 0.02±0.04 for salinity. These small mean differences for potential temperature and salinity show that the method is sufficiently robust.

Why is the correlation between the discrete samples and optode output CO2 partial pressure (Figure 6) so much better when using CO2 concentration vs. partial pressure (from the discrete samples)? The authors should at least offer some educated guesses or speculation.

The better correlation with $c(CO_2)$ was probably related due to an inadequate temperature-parameterisation of the sensor calibration function. The sensor output depends on the changes in pH that are directly related to the changes of $c(CO_2)$ in the membrane and – indirectly – $p(CO_2)$, via Henry's Law. Sensor and external water $p(CO_2)$ should be in equilibrium. The calibration is supposed to correct for the temperature-dependence of the sensor output (Atamanchu*k et al*, 2014). The observation that the sensor output correlated better with $c(CO_2)$ than $p(CO_2)$ is perhaps due to a fortuitous cancellation of an inadequate temperature-parameterisation and the temperature-dependence of the Henry's Law relationship between $c(CO_2)$ than $p(CO_2)$.

I am concerned about the potential impact of advection on the NCP calculation. The study focuses on a SE-NW transect, in a region where waters are transported in a meridional direction along well known currents (NwAC and NCC, as shown in Fig. 1). Can the authors be certain that the time rate of change in O2 and DIC does not reflect advection of water through the transect? What steps did the authors take to ensure that changes in O2 and DIC were truly a function of time and not space? Differentiating temporal vs. spatial changes in measured variables from gliders is not a trivial task and prior studies have typically used repeating spatial patterns to form a 'box' in order to compute O2 and/or carbon budgets for the estimation of NCP. In this study, the glider did not survey a box but a transect in a region of potentially meandering currents and a frontal region separating two water mass

regimes. The authors need to do a better job justifying their methods and eliminating (or at least minimizing) doubt that spatial variations and/or advection contribute significantly to the observed changes in oxygen over the study period.

We have assumed that the main processes controlling the surface dissolved inorganic carbon and oxygen concentrations are biological production and respiration as well as air-sea gas exchange and vertical transport. Even though there are well-known currents, horizontal gradients are reduced due to constant stirring from winds and tides and therefore net advective fluxes are likely to be small (Gislefoss et al., 1998; Falck and Gade, 1999). Previous estimates of net community production in the Norwegian Sea have also neglected advective fluxes (Falck and Gade, 1999; Skjelvan, Falck, Leif G. Anderson, et al., 2001; Falck and Anderson, 2005; Kivimäe, 2007). For example, Gislefoss et al, (1998) considered minimal the effect of horizontal advection on $N(C_T)$ during the summer because $C_T$ changes were largely controlled by biology and air-sea interactions. For NCP estimates on shorter timescales (days to a few weeks), advective fluxes and water-mass movement would have to be taken into account, but this would require a different survey design, involving multiple platforms (Alkire et al., 2014), beyond the scope of the present study.

However, the query from the reviewer prompted us to revisit our NCP calculation, which showed that at the glider turn-around points, inventory changes were calculated over relatively short time-scales of a few days. Therefore, to minimise the effect of horizontal advection in the new version of the manuscript we will extend the time interval used to calculate the inventory changes from less than a week to an average of 50 days. This was achieved by calculating the concentration difference between two transects when the glider moved in the same direction (e.g. transects 1-3, 2-4 and 3-5 all in N-S direction) instead of two consecutive transects.

The authors indicate a separation of the NCP calculation based on water masses with a cutoff at S = 35 that distinguishes between the two primary water masses influencing the study area: Norwegian Atlantic Current (NwAC) water and the Norwegian Coastal Current (NCC) water. It is also stated that salinities between 32 and 34 were encountered in the top 50 m, signifying influence of NCC water. I'm curious whether the authors took mixing into account between the two water masses in the region where NCC was encountered. How might this impact the NCP calculations? Also, I would have appreciated more information regarding the separation. Was NCP calculated separately for each of the two regions? Were they then averaged together to present a single NCP number for O2 and DIC?

The daily value of the net community production ($N$) was calculated separately for NCC and NwAC. The annual $N$ was then calculated combining the two water masses to be consistent with the previous studies (Falck and Gade, 1999; Skjelvan, Falck, Leif G Anderson, et al., 2001; Falck and Anderson, 2005; Kivimäe, 2007).

Here again, the reviewer's comment inspired a revision of our calculation method and for the new version of the paper, we will calculate daily and annual $N$ without separating NCC from NwAC. We change the methodology to be consistent with previous studies, to extend and homogenise the time difference used to calculate inventory and entrainment and to minimise the impact of horizontal advection.

Integration of oxygen & DIC over a specific depth range for the calculation of NCP may be subject to vertical heaving of isopycnals. What steps did the authors take to ensure that such vertical displacement did not impact the calculations? What about vertical mixing from the bottom up? The authors calculate an entrainment flux that focuses on periods when the mixed layer depth exceeded the limit of integration (45 m), but do not discuss the possibility of mixing across the bottom boundary. Admittedly, this is probably minimal, unless there were periods of isopycnal heaving (which looks probable, from the temperature distribution shown in Fig. 8), but the possibility should have been investigated and (at least briefly) mentioned in the manuscript.

We thank the reviewer for the interesting suggestion. Vertical heaving has an effect entraining the water from below the integration. We do already consider the effect of entrainment in the calculation of the net community production, in the form of terms $E(O_2)$ (Eq. 10) and $E(C_T)$ (Eq. 16).

In response to the reviewer's comment, we have also estimated the diapycnal mixing flux and will incorporate this into the new version of the manuscript.

The diapycnal mixing flux ($F_v$) was calculated from the vertical oxygen concentration gradient. In the calculation, we used a vertical eddy diffusivity ($K_z$) of $10^{-5}$ m s$^{-2}$ derived for the Nordic Seas by Naveira Garabato et al. (2004). The effect of $F_v$ for $O_2$ was calculated at $z_{mix}$ when it was deeper than the integration depth $z_{lim}$ and at $z_{lim}$ when $z_{mix}$ was shallower than $z_{lim}$, using the following equation:

$$F_v(O_2) = -K_z \frac{\partial c(O_2)}{\partial z} \tag{3}$$

The net community production ($N$) incorporating $F_v(O_2)$ (scaled in the same way as the flux at the air-sea boundary) would be:

$$N(O_2) = \frac{\Delta I(O_2)}{\Delta t} + \Phi(O_2) \frac{\min(z_{lim}, z_{mix})}{z_{mix}} - E(O_2) + F_v(O_2) \frac{\min(z_{lim}, z_{mix})}{z_{mix}} \tag{4}$$

A positive sign of $F_v(O_2)$ means a decrease of the oxygen concentration in the layer of interest between surface and $z_{lim}$; a negative sign corresponds to an increase.

In the case of $C_T$, $F_V$ was calculated using the equivalent equations:

$$F_v(C_T) = -K_z \frac{\partial c(C_T)}{\partial z} \tag{5}$$

$$N(C_T) = -\frac{\Delta I(C_T)}{\Delta t} - \Phi(CO_2) \frac{\min(z_{lim}, z_{mix})}{z_{mix}} + E(C_T) - F_v(C_T) \frac{\min(z_{lim}, z_{mix})}{z_{mix}} \tag{6}$$

A positive sign of $F_v(C_T)$ means a decrease of the dissolved inorganic concentration in the layer of interest between surface and $z_{lim}$;  a negative sign corresponds to an increase.

In the new version of the manuscript, we will add a new Figure 2 that shows $F_v$ as a function of time during the glider deployment.

[Figure]

**Figure 2:** Diapycnal mixing ($F_v$) calculated for the glider descent and ascent for a) $C_T$ and b) $O_2$ at the mixed layer depth ($z_{mix}$) when deeper than 45 m ($z_{lim}$) and at $z_{lim}$ when $z_{mix}$ was shallower than 45 m. In the calculations, we used a vertical eddy diffusivity ($K_z$) of $10^{-5}$ m s$^{-2}$ (Naveira Garabato *et al.*, 2004).

The results show that $F_v$ is negligibly small: $F_v(C_T) = (0.05\pm0.3)$ mmol m$^{-2}$ d$^{-1}$ and $(-0.02\pm0.33)$ mmol m$^{-2}$ d$^{-1}$ for $O_2$. For that reason, diapycnal mixing will not be used to calculate $N$.

**Specific comments**

Please clarify units of N(CT) and N(O2). Are they both expressed as mmol C m-2 d-1 or do they differ (e.g., mmol C m-2 d-1 vs. mmol O2 m-2 d-1)? After getting to section 3.6, it's clear they were reported in different units, but readers shouldn't have to wait that long to be sure.

Both $N(O_2)$ and $N(C_T)$ are expressed in mmol m$^{-2}$ d$^{-1}$. In the case of $N(O_2)$, this is a flux of $O_2$, in the case of $N(C_T)$, a flux of inorganic carbon.

Preservation of nutrient samples with chloroform is not a recommended procedure. . .

See our answer on page 1.

Figure 2 indeed shows that, on average, the oxygen concentration at higher latitudes was greater (by 10-15 umol kg-1) than those measured at lower latitudes. However, the oxygen concentration decreases fairly linearly with time in both regions (lower and higher latitudes). Why is this the case? I wouldn't think it was short-term drift as such drift should be minimal in oxygen optodes. Does this results, perhaps, from a longitudinal gradient in oxygen concentrations? Figure 3 shows a similar 'drift', or time rate of change, in the gain factor computed to correct the optode oxygen. I am surprised there is such an apparent, continued drift in the optode sensor response. I would have expected a large, initial drift ('storage' drift') but then would have thought the optode response to be relatively stable over a deployment period of ~8 months. Can the authors show the individual, median oxygen concentrations and standard deviations from the discrete data? I'm curious how stable the oxygen concentrations are in this density/depth range (~427 to 1000 m).

The oxygen concentrations for $\sigma_0 > 1028$ kg m$^{-3}$ decreased linearly in both regions because the oxygen optode drifted continuously during the deployment (Figure 2 of the paper and Figure 3 where in red is the uncorrected oxygen, in blue the corrected oxygen and in yellow the discrete sample used as reference). In the new version of the manuscript, we will add a figure with all the samples collected and the glider data before and after the correction showing how the corrected glider oxygen is within the variability of the discrete samples and how stable the $O_2$ concentration is in this depth range. It was possible to use waters of these potential densities because were always well below the mixed layer depth and therefore subject to limited seasonal and interannual variability. The salinity of the discrete samples varied from 34.88 to 34.96, with a mean of $(34.90\pm0.01)$ and the temperature varied from 0.45 to -0.76 °C with a mean of $(-0.15\pm0.36)$ °C. Variations are due to differences in deep-water masses. Therefore, we only used the glider and discrete samples collected at latitudes north of 64° N because this reflects the largest part of the transect. Also, the region south of 64° N contained just 5 days of archived samples. See also reply 2 to Reviewer 2.

We added in the appendices a plot with all the discrete samples and the glider oxygen before and after the correction:

[Figure]

**Figure 3:** a) Discrete samples $c_C(O_2)$ (yellow), raw glider oxygen $c_G(O_2)$ (blue) and drift corrected glider oxygen $c_{G,cal}(O_2)$ (red) using water density > 1028 kg m$^{-3}$.

Line 269: "The thermal lag of the glider conductivity sensor was corrected for. . ." What?

The correct phrase should be "The thermal lag of the glider conductivity sensor was corrected using the method of Gourcuff (2014)."

Can the authors please define cN(Chl a)? Is this the computed chlorophyll concentration, using factory-defined coefficients?

Yes, in the conversion from the raw chlorophyll to the chlorophyll concentration, we used the factory-based coefficients.

Line 363: ". . .because after this dive, the CO2 optode stopped sampling. . ."

We meant "For the subsequent dives, the $CO_2$ optode stopped sampling in the first 150 m (Figure 2.8d)."

Line 364: ". . .raw c(O2) data was calibrated and drift-corrected and c(CO2) was driftand lag-corrected and recalibrated, then used to. . ." I'm not going to focus my review on grammar corrections, so I suggest the authors carefully re-read the manuscript to avoid any additional grammar or spelling mistakes that should be addressed prior to publication.

Apologies if the sentence structure was unclear. We meant to say that "The raw $c(O_2)$ data were drift-corrected and calibrated. The $CO_2$ output was drift and lag-corrected and then calibrated against $c_C(CO_2)$ from nearby discrete samples. The calibrated glider $c_G(O_2)$ and $c_G(CO_2)$ were used to calculate inventory changes and air-sea exchange fluxes ($\Phi$) to evaluate the net community production changes."

Plot isopycnals on panels of Fig. 8. I'd also recommend plotting the mixed layer depth and highlighting zlim (dotted line?).

We changed figure 8 adding the mixed layer depth, $z_{lim}$ and the isopycnals (Figure 4).

[Figure]

**Figure 4:** Raw glider data for all 703 dives with latitude of the glider trajectory at the top (black: NwAC; red: NCC, separated by a $S$ of 35). a) temperature $\theta$, b) salinity $S$, c) oxygen concentration $c(O_2)$, d) uncorrected $CO_2$ optode output $p_u(CO_2)$ and e) chlorophyll $a$ concentration $c_{raw}(Chl\ a)$. The white space means that the sensors did not measure any data. The pink line is $z_{mix}$ calculated using a threshold criterion of $\Delta\theta = 0.5$ °C to median $\theta$ of the top 5 m of the glider profile (Obata et al., 1996; United States. National Environmental Satellite and Information Service, Monterey and Levitus, 1997; Fo*ltz* et al., 2003), the black dotted line $z_{lim}$ used as depth limit to calculate the net community production ($N$) and black contour lines are the isopycnals.

Line 375: What is "against year-day"? Please re-word this sentence.

Year-day means day of the year and varies from 1 to 365.

We will change all occurrences of year-day in the manuscript to "day of the year".

Lines 456-457: Can the authors please expand on how NCP was calculated? It is stated that, "The two Ns were calculated as the difference in inventory changes between two transects when the glider was in the same water mass." Two transects? So, is one transect equivalent to the glider moving over the entire transect in one direction and the second transect is the glider moving back over the transect in the opposite direction? Is the NCP calculated only for the NwAC water mass? So any changes within the NCC water mass are removed from the analysis?

Yes, it is correct, that we used one transect with the glider moving in one direction and the following transect with the glider moving in the opposite direction. To calculate the net community production ($N$) the data were binned into 0.1° latitude intervals and the inventory changes were calculated as the difference of the integrated $c(O_2)$ and $C_T$ every time the glider was in the latitude bin. The air-sea flux was the instantaneous flux when the glider was in the bin and the entrainment was considered as the concentration changes when the mixed layer deepened between two transects in the same latitude bin.

The daily $N$ was calculated separating the two water masses (NCC and NwAC) and the annual $N$ was calculated as the mean of the daily $N$ considering the two water masses together.

Following the prompt for the reviewer on the possible influence of horizontal advection (see above), the revised version of the manuscript will use an amended methodology to calculate net community production will. We will use the difference of $C_T$ and $O_2$ between two transects when the glider moved in the same direction (e.g. southeast to northwest). We will not use two consecutive transects. This means that inventory changes will be calculated based on a similar time difference between the two samples. For two consecutive transects, the time difference between the two samples would be smaller at the beginning and the end of the transect and larger in the middle.

Also, to correct for the variability of the wind speed, we will use flux-weighted gas transfer velocities for $O_2$ and $CO_2$ (Re*uer et al*, 2007), rather than instantaneous fluxes (as before). $k_w(O_2)$ and $k_w(CO_2)$ will be normalised using the daily wind speed in the latitude bin in the time interval used to calculate the inventory changes. The time interval is the time between two samples used to calculate the inventory changes and entrainment. The air-sea flux is based on the concentration measured at the time of the second transect used to calculate the inventory changes and the entrainment flux.

It is important to compare NCP estimates with those of previous studies; however, it is difficult to know how comparable the numbers are in Table 3 because it is not clear where in the Norwegian Sea these various studies took place. It is also difficult because zlim varies largely among the studies. The fact that three of the four compared studies used zlim >= 100 m also calls into question why exactly the current study decided on zlim = 45 m, particularly since the mixed layer depth varied so largely and often exceeded zlim.

We used $z_{lim} = 45$ m because this corresponds to the average depth of the euphotic zone, which is the region of interest for net community production from a biogeochemical and ecological point of view.

Previous studies may have used $z_{lim}$ = 100 m for operational reasons (e.g. constrained by discrete sampling depths).

To show the influence of $z_{lim}$ on $N$, we calculated $N$ for $z_{lim}$ = 30 m and 100 m. $N(C_T;\ 30\ m)$ was 0.6 and $N(C_T;\ 100\ m)$ was –0.04 mol m$^{-2}$ a$^{-1}$. $N(O_2;\ 30\ m)$ was 4.6 and $N(O_2;\ 100\ m)$ was 4.3 mol m$^{-2}$ a$^{-1}$. In the case of $N(C_T)$, the derived two values are lower to the previous studies where $N(C_T)$ varied from 8.6 to 2.0 mol m$^{-2}$ a$^{-1}$. $N(C_T;\ 100\ m)$ was negative because the deep integration depth included water below the euphotic zone where the remineralisation of organic matter can increase $C_T$. This signal is not present in $N(O_2)$ because the changes were largely controlled by $\Phi(O_2)$ that was always positive. The calculated $N(O_2)$ is in agreement with previous studies, which gave results between 2.6 and 11 mol m$^{-2}$ a$^{-1}$.

In the discussion, we will add a section where we explain the location and the period when the previous studies took place. Falck and Anderson (2005) used historical data from 1960 to 2000 collected in the area from 62 to 70° N and from 1991 to 1994 collected at OWSM. Skjelvan et al (2001) used data collected from 67.5° N 9° E to 71.5° N 1° E and along 74.5° N from 7 to 15° E from 1957 to 1970 and 1991 to 1998. Kivimäe (2007) used the oxygen measured at OWSM from 1955 to 2005 and Falck and Gade (1999) used data collected in all the Norwegian Sea from 1955 to 1988. The glider in the transect moved from to 66.3 °N 4 °W to 63 °N 4 °E (Figure 5 where the black dots are the glider dives, the green box the region used by Falck and Gade (1999), the yellow lines the transects used by Skjelvan *et al.* (2001), the azure box the region used by Falck and Anderson (2005) and the red dot the location that corresponds to the Ocean Weather Station M (OWSM) used by Kivimäe (2007)).

[Figure]

**Figure 5:** Map of the glider deployment showing the previous studies that estimated the net community production in the Norwegian Sea. The black dots are the glider dives, the green box the region used by Falck and Gade (1999), the yellow lines the transects used by Skjelvan *et al.* (2001), the azure box the region used by Falck and Anderson (2005) and the red dot the location that corresponds to the Ocean Weather Station M (OWSM) used by Kivimäe (2007).

**Bibliography**

Alkire, M. B. *et al.* (2014) 'Net community production and export from Seaglider measurements in the North Atlantic after the spring bloom', *Journal of Geophysical Research: Oceans*. Wiley Online Library, 119(9), pp. 6121–6139.

Atamanchuk, D. *et al.* (2014) 'Performance of a lifetime-based optode for measuring partial pressure of carbon dioxide in natural waters', *Limnology and Oceanography: Methods*, 12(2), pp. 63–73. doi: 10.4319/lom.2014.12.63.

Becker, S. *et al.* (2020) 'GO-SHIP repeat hydrography nutrient manual: the precise and accurate determination of dissolved inorganic nutrients in seawater, using continuous flow analysis methods', *Frontiers in Marine Science*. Frontiers, 7, p. 908.

Falck, E. and Anderson, L. G. (2005) 'The dynamics of the carbon cycle in the surface water of the Norwegian Sea', 94, pp. 43–53. doi: 10.1016/j.marchem.2004.08.009.

Falck, E. and Gade, G. (1999) 'Net community production and oxygen fluxes in the Nordic Seas based on O2 budget calculations', 13(4), pp. 1117–1126.

Foltz, G. R. *et al.* (2003) 'Seasonal mixed layer heat budget of the tropical Atlantic Ocean', *Journal of Geophysical Research: Oceans*. Wiley Online Library, 108(C5).

Gislefoss, J. S. *et al.* (1998) 'Carbon time series in the Norwegian sea', 45, pp. 433–460.

Gourcuff, C. (2014) 'ANFOG Slocum CTD data correction', (March).

Hagebo, M. and Rey, F. (1984) 'Storage of seawater for nutrients analysis', *Fisken Hav., 4, 1*, 12.

Van Heuven, S. *et al.* (2011) 'MATLAB program developed for CO2 system calculations', *ORNL/CDIAC-105b. Carbon Dioxide Information Analysis Center, Oak Ridge National Laboratory, US Department of Energy, Oak Ridge, Tennessee*, 530.

Kivimäe, C. (2007) 'Carbon and oxygen fluxes in the Barents and Norwegian Seas: production, air-sea exchange and budget calculations'. The University of Bergen.

Miloshevich, L. (2004) 'Development and Validation of a Time-Lag Correction for Vaisala Radiosonde Humidity Measurements', pp. 1305–1328.

Naveira Garabato, A. C. *et al.* (2004) 'Turbulent diapycnal mixing in the Nordic seas', *Journal of Geophysical Research: Oceans*. Wiley Online Library, 109(C12).

Obata, A., Ishizaka, J. and Endoh, M. (1996) 'Global verification of critical depth theory for phytoplankton bloom with climatological in situ temperature and satellite ocean color data', *Journal of Geophysical Research: Oceans*. Wiley Online Library, 101(C9), pp. 20657–20667.

Reuer, M. K. *et al.* (2007) 'New estimates of Southern Ocean biological production rates from O2/Ar ratios and the triple isotope composition of O2', *Deep Sea Research Part I: Oceanographic Research Papers*. Elsevier, 54(6), pp. 951–974.

Skjelvan, I., Falck, E., Anderson, Leif G, *et al.* (2001) 'Oxygen fluxes in the Norwegian Atlantic current', *Marine chemistry*. Elsevier, 73(3–4), pp. 291–303.

Skjelvan, I., Falck, E., Anderson, Leif G., *et al.* (2001) 'Oxygen fluxes in the Norwegian Atlantic Current', *Marine Chemistry*, 73(3–4), pp. 291–303. doi: 10.1016/S0304-4203(00)00112-2.

United States. National Environmental Satellite  and Information Service, D., Monterey, G. I. and Levitus, S. (1997) *Seasonal variability of mixed layer depth for the world ocean*. US Department of Commerce, National Oceanic and Atmospheric Administration ….

---

## Author Response (AR1)

**Response to Reviewer 1**

We would like to thank Reviewer 1 for providing constructive and insightful comments. We incorporated their suggestions into a revised manuscript. Reviewer 1's comments have been reproduced below in black, with the authors' response in blue.

**General comments**

Parameterization for deriving phosphate and silicate concentrations along the glider track from 'spot' samples collected during four cruises over the deployment (March, May, June, and October). Sampling restricted to the southern half of the transect. And yet, the uncertainties were only 1.3 and 0.13 umol kg-1 for silicate and phosphate? I hope this parameterization is discussed in detail (in the text or an appendix). I also hope that some sensitivity analysis was completed regarding the impact of differing nutrient concentrations (within a reasonable range for the region & study period) on CO2SYS calculations. I'm also concerned about the use of chloroform to preserve nutrient samples.

The reviewer's comment highlighted that our method description has been too brief on nutrient analysis. In fact, we collected 58 discrete samples along the glider transect in March, May, June and October, which were analysed for total alkalinity ($A_T$), dissolved inorganic carbon ($C_T$) as well as nutrient concentrations. In addition, we used 52 $C_T$ and $A_T$ discrete samples collected at a weather station (OWSM) located 270 km northeast of the glider transect (Figure 1 of the discussion paper) in March, May, June, August and November. For the latter samples, no nutrient measurements were undertaken. Instead, we filled in these gaps with the nutrient concentration nearest in time and depth from the 58 along-transect samples, which we expected to be sufficiently close to the actual concentration at OWSM. We then derived the $CO_2$ concentration ($c(CO_2)$) using the MATLAB toolbox CO2SYS (Van Heuven et al., 2011), using the phosphate and silicate concentrations to account for their contributions to $A_T$. This $c(CO_2)$ was used as reference to calibrate the glider output (Figure 1, in red the samples collected along the transect and in black at OWSM). The nutrient concentrations were not used in any further calculations.

[Figure]

**Figure 1:** Calibration of the $CO_2$ optode using the samples collected along the glider transect (red) and at OWSM (black) a) $CO_2$ concentration of the discrete samples ($c_{WS}(CO_2)$) against the glider output with the linear regression line and b) $CO_2$ partial pressure of the discrete samples ($p_{WS}(CO_2)$) against the glider output with the linear regression line.

To assess the uncertainty of the final silicate and phosphate concentration we calculated the uncertainty in the calculation of the OWSM $c(CO_2)$ using the interpolation uncertainty 1.1 and 0.12 $\mu$mol kg$^{-1}$ for the silicate and phosphate concentration, respectively. The uncertainty was calculated as

the root mean square error of the interpolation, which estimates the standard deviation of the error distribution. The interpolation was the following for phosphate:

$$c(PO_4) = 0.0003 \times z/m + -0.0003 \times t + 220.18 \tag{1}$$

where $z$ is the depth and $t$ the date. In the case of silicate the equation was:

$$c(Si) = 0.0054 \times z/m + -0.0049 \times t + 3626.3. \tag{2}$$

The derived nutrient concentration led to a mean error of 0.04 µmol kg$^{-1}$ in the calculation of $c(CO_2)$.

We followed the procedure described by Hagebo and Rey (1984) to collect and preserve the nutrients. We are not aware of any problems with the use of chloroform to preserve nutrient samples. The 2019 GO-SHIP Repeat Hydrography Nutrient Manual only discourages the use of acid (which would require neutralisation before analysis) and mercuric chloride (a long-term environmental hazard) (Becker *et al.*, 2020).

We changed line 164-167: "This parameterisation had an uncertainty of 1.3 and 0.13 µmol kg$^{-1}$ and a $R^2$ of 0.6 and 0.4, for silicate and phosphate concentrations, respectively. The uncertainty was calculated as the root mean square difference between measured and parameterised concentrations. This nutrient concentration uncertainty contributed an uncertainty of 0.04 µmol kg$^{-1}$ in the calculation of $c(CO_2)$, which is negligible".

For the lag correction to the CO2 optode, data from the glider ascents are compared against those from descents. However, there is significant horizontal distance between a glider ascent and descent, unlike what one might expect for a CTD cast from a ship. By minimizing the differences observed between glider ascents & descents, you are loosing information and I'm not sure the lag correction is necessarily reliable. I would suggest comparing potential temperature and salinity in glider ascents & descents. Do they match? If so, then perhaps this method is OK. If not, the authors may need to revise the lag correction method.
The CO$_2$ optode lag was corrected using the algorithm of Miloshevich, (2004) that uses the sensor response time ($\tau$). The $\tau$ was calculated minimising the difference between each glider ascent and descent (see Figure 5 of the discussion paper). In the equation, we used the median of all the $\tau$ values (1384 s). The correction decreased the difference between the glider and descent of the raw $p(CO_2)$ from (71±30) to (21±26) µatm. Using the median $\tau$ is a robust indicator for the lag time, even there were short-term variations (e.g. due to internal tides or waves) between the descent and ascent.

To assess if the glider was in the same water mass between ascent and descent we looked at potential temperature and salinity. The mean difference between the descent and ascent was (0.13±0.33) °C for potential temperature and 0.02±0.04 for salinity. These small mean differences for potential temperature and salinity show that the method is sufficiently robust.

Why is the correlation between the discrete samples and optode output CO2 partial pressure (Figure 6) so much better when using CO2 concentration vs. partial pressure (from the discrete samples)? The authors should at least offer some educated guesses or speculation.

The better correlation with $c(CO_2)$ was probably related due to an inadequate temperature-parameterisation of the sensor calibration function. The sensor output depends on the changes in pH that are directly related to the changes of $c(CO_2)$ in the membrane and – indirectly – $p(CO_2)$, via Henry's Law. Sensor and external water $p(CO_2)$ should be in equilibrium. The calibration is supposed to correct for the temperature-dependence of the sensor output (Atamanchuk et al, 2014). The observation that the sensor output correlated better with $c(CO_2)$ than $p(CO_2)$ is perhaps due to a fortuitous cancellation of an inadequate temperature-parameterisation and the temperature-dependence of the Henry's Law relationship between $c(CO_2)$ than $p(CO_2)$.

We added in section 4.1, line 712-718: "The better correlation with $c(CO_2)$ was probably related due to an inadequate temperature-parameterisation of the sensor calibration function. The sensor output depends on the changes in pH that are directly related to the changes of $c(CO_2)$ in the membrane and – indirectly – $p(CO_2)$, via Henry's Law. The calibration is supposed to correct for the temperaturedependence of the sensor output (Atamanchuk et al, 2014). So the fact, that the sensor output correlated better with $c(CO_2)$ than $p(CO_2)$ is perhaps due to a fortuitous cancellation of an inadequate temperature-parameterisation and the Henry's Law relationship between $c(CO_2)$ than $p(CO_2)$."

I am concerned about the potential impact of advection on the NCP calculation. The study focuses on a SE-NW transect, in a region where waters are transported in a meridional direction along well known currents (NwAC and NCC, as shown in Fig. 1). Can the authors be certain that the time rate of change in O2 and DIC does not reflect advection of water through the transect? What steps did the authors take to ensure that changes in O2 and DIC were truly a function of time and not space? Differentiating temporal vs. spatial changes in measured variables from gliders is not a trivial task and prior studies have typically used repeating spatial patterns to form a 'box' in order to compute O2 and/or carbon budgets for the estimation of NCP. In this study, the glider did not survey a box but a transect in a region of potentially meandering currents and a frontal region separating two water mass regimes. The authors need to do a better job justifying their methods and eliminating (or at least minimizing) doubt that spatial variations and/or advection contribute significantly to the observed changes in oxygen over the study period.

We have assumed that the main processes controlling the surface dissolved inorganic carbon and oxygen concentrations are biological production and respiration as well as air-sea gas exchange and vertical transport. Even though there are well-known currents, horizontal gradients are reduced due to constant stirring from winds and tides and therefore net advective fluxes are likely to be small (Gislefoss et al., 1998; Falck and Gade, 1999). Previous estimates of net community production in the Norwegian Sea have also neglected advective fluxes (Falck and Gade, 1999; Skjelvan, Falck, Leif G. Anderson, et al., 2001; Falck and Anderson, 2005; Kivimäe, 2007). For example, Gislefoss et al, (1998) considered minimal the effect of horizontal advection on $N(C_T)$ during the summer because $C_T$ changes were largely controlled by biology and air-sea interactions. For NCP estimates on shorter timescales (days to a few weeks), advective fluxes and water-mass movement would have to be taken into account, but this would require a different survey design, involving multiple platforms (Alkire et al., 2014), beyond the scope of the present study.

However, the query from the reviewer prompted us to revisit our NCP calculation, which showed that at the glider turn-around points, inventory changes were calculated over relatively short time-scales of a few days. Therefore, to minimise the effect of horizontal advection in the new version of the manuscript we extended the time interval used to calculate the inventory changes from less than a week to an average of 50 days. This was achieved by calculating the concentration difference between two transects when the glider moved in the same direction (e.g. transects 1-3, 2-4 and 3-5 all in N-S direction) instead of two consecutive transects.

We changed section 2.9, line 438-442: "The contribution of horizontal advection to $N(C_T)$ was considered minimal over the timescales we calculated inventory changes because previous studies have shown that changes in $C_T$ during summer are mainly controlled by biology and air-sea interactions (Gislefoss et al., 1998). For that reason, previous studies that estimated $N$ in the Norwegian Sea have also neglected advective fluxes (Falck and Anderson, 2005; Falck and Gade, 1999; Kivimäe, 2007; Skjelvan et al., 2001)."

The authors indicate a separation of the NCP calculation based on water masses with a cutoff at S = 35 that distinguishes between the two primary water masses influencing the study area: Norwegian Atlantic Current (NwAC) water and the Norwegian Coastal Current (NCC) water. It is also stated that salinities between 32 and 34 were encountered in the top 50 m, signifying influence of NCC water. I'm curious whether the authors took mixing into account between the two water masses in the region where NCC was encountered. How might this impact the NCP calculations? Also, I would have appreciated more information regarding the separation. Was NCP calculated separately for each of the two regions? Were they then averaged together to present a single NCP number for O2 and DIC?

The daily value of the net community production ($N$) was calculated separately for NCC and NwAC. The annual $N$ was then calculated combining the two water masses to be consistent with the previous

studies (Falck and Gade, 1999; Skjelvan, Falck, Leif G Anderson, et al., 2001; Falck and Anderson, 2005; Kivimäe, 2007).

Here again, the reviewer's comment inspired a revision of our calculation method and for the new version of the paper, we calculated daily and annual $N$ without separating NCC from NwAC. We changed the methodology to be consistent with previous studies, to extend and homogenise the time difference used to calculate inventory and entrainment and to minimise the impact of horizontal advection.

We changed section 3.3, line 595-599: "The two $N$ values were calculated as the difference in inventory changes between two transects when the glider moved in the same direction. This method was used in order to have similar time interval between repeat occupations of the same transect position to calculate the inventory changes and entrainment. "

Integration of oxygen & DIC over a specific depth range for the calculation of NCP may be subject to vertical heaving of isopycnals. What steps did the authors take to ensure that such vertical displacement did not impact the calculations? What about vertical mixing from the bottom up? The authors calculate an entrainment flux that focuses on periods when the mixed layer depth exceeded the limit of integration (45 m), but do not discuss the possibility of mixing across the bottom boundary. Admittedly, this is probably minimal, unless there were periods of isopycnal heaving (which looks probable, from the temperature distribution shown in Fig. 8), but the possibility should have been investigated and (at least briefly) mentioned in the manuscript.

We thank the reviewer for the interesting suggestion. Vertical heaving has an effect entraining the water from below the integration. We do already consider the effect of entrainment in the calculation of the net community production, in the form of terms $E(O_2)$ (Eq. 10) and $E(C_T)$ (Eq. 17).

In response to the reviewer's comment, we have also estimated the diapycnal mixing flux and incorporated this into the new version of the manuscript.

The diapycnal mixing flux ($F_v$) was calculated from the vertical oxygen concentration gradient. In the calculation, we used a vertical eddy diffusivity ($K_z$) of $10^{-5}$ m s$^{-2}$ derived for the Nordic Seas by Naveira Garabato et al. (2004). The effect of $F_v$ for $O_2$ was calculated at $z_{mix}$ when it was deeper than the integration depth $z_{lim}$ and at $z_{lim}$ when $z_{mix}$ was shallower than $z_{lim}$, using the following equation:

$$F_v(O_2) = K_z \frac{\partial c(O_2)}{\partial z} \tag{3}$$

The net community production ($N$) incorporating $F_v(O_2)$ (scaled in the same way as the flux at the air-sea boundary) would be:

$$N(O_2) = \frac{\Delta I(O_2)}{\Delta t} + \Phi(O_2) \frac{\min(z_{lim}, z_{mix})}{z_{mix}} - E(O_2) + F_v(O_2) \frac{\min(z_{lim}, z_{mix})}{z_{mix}} \tag{4}$$

A positive sign of $F_v(O_2)$ means a decrease of the oxygen concentration in the layer of interest between surface and $z_{lim}$; a negative sign corresponds to an increase.

In the case of $C_T$, $F_V$ was calculated using the equivalent equations:

$$F_v(C_T) = K_z \frac{\partial c(C_T)}{\partial z} \tag{5}$$

$$N(C_T) = -\frac{\Delta I(C_T)}{\Delta t} - \Phi(CO_2) \frac{\min(z_{lim}, z_{mix})}{z_{mix}} + E(C_T) - F_v(C_T) \frac{\min(z_{lim}, z_{mix})}{z_{mix}} \tag{6}$$

A positive sign of $F_v(C_T)$ means a decrease of the dissolved inorganic concentration in the layer of interest between surface and $z_{lim}$; a negative sign corresponds to an increase.

In the new version of the manuscript, we added a new Figure 2 that shows $F_v$ as a function of time during the glider deployment.

[Figure]

**Figure 2:** Diapycnal mixing ($F_v$) calculated for the glider descent and ascent for a) $C_T$ and b) $O_2$ at the mixed layer depth ($z_{mix}$) when deeper than 45 m ($z_{lim}$) and at $z_{lim}$ when $z_{mix}$ was shallower than 45 m. In the calculations, we used a vertical eddy diffusivity ($K_z$) of $10^{-5}$ m s$^{-2}$ (Naveira Garabato et al., 2004).

The results show that $F_v$ is negligibly small: $F_v(C_T) = (0.05 \pm 0.3)$ mmol m$^{-2}$ d$^{-1}$ and ($-0.02 \pm 0.33$) mmol m$^{-2}$ d$^{-1}$ for $O_2$.

We added in section 2.8, line 396-401: "The effect of diapycnal eddy diffusion ($F_v$) was calculated at $z_{mix}$ when it was deeper than $z_{lim}$ and at $z_{lim}$ when $z_{mix}$ was shallower than $z_{lim}$, using the following equation:

$$F_v(O_2) = K_z \frac{\partial c(O_2)}{\partial z} \tag{11}$$

for a vertical eddy diffusivity ($K_z$) of $10^{-5}$ m s$^{-2}$ (Naveira Garabato et al., 2004). The effect of $F_v(O_2)$ on $N(O_2)$ was negligible (Figure A2b) with a median of ($-0.06 \pm 0.34$) mmol m$^{-2}$ d$^{-1}$."

And in section 2.9, line 433-437: "As for oxygen, the effect of diapycnal eddy diffusion ($F_v$) was calculated at $z_{mix}$ when it was deeper than $z_{lim}$ and at $z_{lim}$ when $z_{mix}$ was shallower than $z_{lim}$, using the following equation:

$$F_v(C_T) = K_z \frac{\partial c(C_T)}{\partial z} \tag{18}$$

for a $K_z$ of $10^{-5}$ m s$^{-2}$ (Naveira Garabato et al., 2004). The effect of $F_v(C_T)$ was negligible (Figure A2a) with a median of ($0.07 \pm 0.3$) mmol m$^{-2}$ d$^{-1}$. "

**Specific comments**

Please clarify units of N(CT) and N(O2). Are they both expressed as mmol C m-2 d-1 or do they differ (e.g., mmol C m-2 d-1 vs. mmol O2 m-2 d-1)? After getting to section 3.6, it's clear they were reported in different units, but readers shouldn't have to wait that long to be sure.

Both $N(O_2)$ and $N(C_T)$ are expressed in mmol m$^{-2}$ d$^{-1}$. In the case of $N(O_2)$, this is a flux of $O_2$, in the case of $N(C_T)$, a flux of inorganic carbon.

Added in section 2.8 line 355-357: " We calculated $N(O_2)$ (in mmol m$^{-2}$ d$^{-1}$) from the oxygen inventory changes ($I(O_2)$) corrected for air-sea exchange $\Phi(O_2)$, normalised to $z_{mix}$ when $z_{mix}$ was deeper than the integration depth of $z_{lim} = 45$ m, entrainment $E(O_2)$ and diapycnal eddy diffusion $F_v(O_2)$:"

And in section 2.9, line 403-404: "$N(C_T)$ was expressed in mmol m$^{-2}$ d$^{-1}$ and was calculated from the $C_T$ inventory changes $I(C_T)$, air-sea flux of $CO_2$, $\Phi(CO_2)$, entrainment $E(C_T)$ and diapycnal diffusion $F_v(C_T)$:"

Preservation of nutrient samples with chloroform is not a recommended procedure. . .

See our answer on page 1.

Figure 2 indeed shows that, on average, the oxygen concentration at higher latitudes was greater (by 10-15 umol kg-1) than those measured at lower latitudes. However, the oxygen concentration decreases fairly linearly with time in both regions (lower and higher latitudes). Why is this the case? I wouldn't think it was short-term drift as such drift should be minimal in oxygen optodes. Does this results, perhaps, from a longitudinal gradient in oxygen concentrations? Figure 3 shows a similar 'drift', or time rate of change, in the gain factor computed to correct the optode oxygen. I am surprised there is such an apparent, continued drift in the optode sensor response. I would have expected a large, initial drift ('storage' drift') but then would have thought the optode response to be relatively stable over a deployment period of ~8 months. Can the authors show the individual, median oxygen concentrations and standard deviations from the discrete data? I'm curious how stable the oxygen concentrations are in this density/depth range (~427 to 1000 m).

The oxygen concentrations for $\sigma_0 > 1028$ kg m$^{-3}$ decreased linearly in both regions because the oxygen optode drifted continuously during the deployment (Figure 2 of the paper and Figure 3 where in red is the uncorrected oxygen, in blue the corrected oxygen and in yellow the discrete sample used as reference). In the new version of the manuscript, we added a figure with all the samples collected and the glider data before and after the correction showing how the corrected glider oxygen is within the variability of the discrete samples and how stable the $O_2$ concentration is in this depth range. It was possible to use waters of these potential densities because were always well below the mixed layer depth and therefore subject to limited seasonal and interannual variability. The salinity of the discrete samples varied from 34.88 to 34.96, with a mean of (34.90±0.01) and the temperature varied from 0.45 to -0.76 °C with a mean of (-0.15±0.36) °C. Variations are due to differences in deep-water masses. Therefore, we only used the glider and discrete samples collected at latitudes north of 64° N because this reflects the largest part of the transect. Also, the region south of 64° N contained just 5 days of archived samples. See also reply 2 to Reviewer 2.

We added in the appendices a plot with all the discrete samples and the glider oxygen before and after the correction:

[Figure]

**Figure 3:** a) Discrete samples $c_C(O_2)$ (yellow), raw glider oxygen $c_G(O_2)$ (blue) and drift corrected glider oxygen $c_{G,cal}(O_2)$ (red) using water density > 1028 kg m$^{-3}$.

Line 269: "The thermal lag of the glider conductivity sensor was corrected for. . ." What?

The correct phrase should be "The thermal lag of the glider conductivity sensor was corrected using the method of Gourcuff (2014)."

Can the authors please define cN(Chl a)? Is this the computed chlorophyll concentration, using factory-defined coefficients?

Yes, in the conversion from the raw chlorophyll to the chlorophyll concentration, we used the factory-based coefficients.

We added in section 2.7, line 321-322: "Glider-reported chlorophyll concentrations, $c_{raw}(Chl\ a)$, were computed using the factory coefficients. $c_{raw}(Chl\ a)$ was affected by photochemical quenching during the daytime dives."

Line 363: ". . .because after this dive, the CO2 optode stopped sampling. . ."

We meant "For the subsequent dives, the $CO_2$ optode stopped sampling in the first 150 m (Figure 2.8d)."

Line 364: ". . .raw c(O2) data was calibrated and drift-corrected and c(CO2) was driftand lag-corrected and recalibrated, then used to. . ." I'm not going to focus my review on grammar corrections, so I suggest the authors carefully re-read the manuscript to avoid any additional grammar or spelling mistakes that should be addressed prior to publication.

Apologies if the sentence structure was unclear. We meant to say that "The raw $c(O_2)$ data were drift-corrected and calibrated. The $CO_2$ output was drift and lag-corrected and then calibrated against $c_C(CO_2)$ from nearby discrete samples. The calibrated glider $c_G(O_2)$ and $c_G(CO_2)$ were used to calculate inventory changes and air-sea exchange fluxes ($\Phi$) to evaluate the net community production changes."

Plot isopycnals on panels of Fig. 8. I'd also recommend plotting the mixed layer depth and highlighting zlim (dotted line?).

We changed figure 8 adding the mixed layer depth, $z_{lim}$ and the isopycnals (Figure 4).

[Figure]

**Figure 4:** Raw glider data for all 703 dives with latitude of the glider trajectory at the top (black: NwAC; red: NCC, separated by a $S$ of 35). a) temperature $\theta$, b) salinity $S$, c) oxygen concentration $c(O_2)$, d) uncorrected $CO_2$ optode output $p_u(CO_2)$ and e) chlorophyll $a$ concentration $c_{raw}(Chl\ a)$. The white space means that the sensors did not measure any data. The pink line is $z_{mix}$ calculated using a threshold criterion of $\Delta\theta = 0.5\ °C$ to median $\theta$ of the top 5 m of the glider profile (Obata et al., 1996; United States. National Environmental Satellite and Information Service, Monterey and Levitus, 1997; Foltz et al., 2003), the black dotted line $z_{lim}$ used as depth limit to calculate the net community production ($N$) and black contour lines are the isopycnals.

Line 375: What is "against year-day"? Please re-word this sentence.

Year-day means day of the year and varies from 1 to 365.

We changed all occurrences of year-day in the manuscript to "day of the year".

Lines 456-457: Can the authors please expand on how NCP was calculated? It is stated that, "The two Ns were calculated as the difference in inventory changes between two transects when the glider was in the same water mass." Two transects? So, is one transect equivalent to the glider moving over the entire transect in one direction and the second transect is the glider moving back over the transect in the opposite direction? Is the NCP calculated only for the NwAC water mass? So any changes within the NCC water mass are removed from the analysis?

Yes, it is correct, that we used one transect with the glider moving in one direction and the following transect with the glider moving in the opposite direction. To calculate the net community production ($N$) the data were binned into 0.1° latitude intervals and the inventory changes were calculated as the difference of the integrated $c(O_2)$ and $C_T$ every time the glider was in the latitude bin. The air-sea flux was the instantaneous flux when the glider was in the bin and the entrainment was considered as the concentration changes when the mixed layer deepened between two transects in the same latitude bin.

The daily $N$ was calculated separating the two water masses (NCC and NwAC) and the annual $N$ was calculated as the mean of the daily $N$ considering the two water masses together.

Following the prompt for the reviewer on the possible influence of horizontal advection (see above), the revised version of the manuscript used an amended methodology to calculate net community production. We used the difference of $C_T$ and $O_2$ between two transects when the glider moved in the same direction (e.g. southeast to northwest). We did not use two consecutive transects. This means that inventory changes were calculated based on a similar time difference between the two samples. For two consecutive transects, the time difference between the two samples would be smaller at the beginning and the end of the transect and larger in the middle.

Also, to correct for the variability of the wind speed, we used flux-weighted gas transfer velocities for $O_2$ and $CO_2$ (Reuer et al, 2007), rather than instantaneous fluxes (as before). $k_w(O_2)$ and $k_w(CO_2)$ were normalised using the daily wind speed in the latitude bin in the time interval used to calculate the inventory changes. The time interval is the time between two samples used to calculate the inventory changes and entrainment. The air-sea flux is based on the concentration measured at the time of the second transect used to calculate the inventory changes and the entrainment flux.

We changed section 2.8, line 349-354: "Calculating net community production $N$ from glider data is challenging because the glider continuously moves through different water masses. For that reason we subdivided the transect by binning the data into 0.1° latitude intervals to derive $O_2$ concentration changes every two transects. The changes were calculated between transects in the same direction of glider travel (e.g. transects 1 and 3, both in N-S direction) to have approximately the same time difference (40-58 days) at every latitude. If instead we had used two consecutive transects, this would lead to a highly variable time difference of near-0 to about 50 days along the transect."

And section 2.8, line 386-389: "To account for wind speed variability, $k_w(O_2)$ applied to calculate $N(O_2)$ was a weighted mean based on the varying daily-mean wind speed $U$ in the time interval

between $t_n$ and $t_{n+1}$ ($\Delta t$) used to calculate $\frac{\Delta I(O_2)}{\Delta t}$ using a points median $z_{mix}$ and for 50 days to calculate $\Phi(O_2)$ (section 3.2) (Reuer et al., 2007)."

And section 2.9, line 423-426: "To account for wind speed variability, $k_w(CO_2)$ applied to calculate $N(O_2)$ was a weighted mean based on the varying daily-mean wind speed $U$ in the time interval between $t_n$ and $t_{n+1}$ ($\Delta t$) used to calculate $\frac{\Delta I(C_T)}{\Delta t}$ and for 50 days to calculate $\Phi(CO_2)$ (section 3.2) (Reuer et al., 2007)."

It is important to compare NCP estimates with those of previous studies; however, it is difficult to know how comparable the numbers are in Table 3 because it is not clear where in the Norwegian Sea these various studies took place. It is also difficult because zlim varies largely among the studies. The fact that three of the four compared studies used zlim >= 100 m also calls into question why exactly the current study decided on zlim = 45 m, particularly since the mixed layer depth varied so largely and often exceeded zlim.

We used $z_{lim}$ = 45 m because this corresponds to the average depth of the euphotic zone, which is the region of interest for net community production from a biogeochemical and ecological point of view. Previous studies may have used $z_{lim}$ = 100 m for operational reasons (e.g. constrained by discrete sampling depths).

To show the influence of $z_{lim}$ on $N$, we calculated $N$ for $z_{lim}$ = 30 m and 100 m. $N(C_T;$ 30 m) was 1 and $N(C_T;$ 100 m) was 0.4 mol m$^{-2}$ a$^{-1}$. $N(O_2;$ 30 m) was 4 and $N(O_2;$ 100 m) was 3.7 mol m$^{-2}$ a$^{-1}$. In the case of $N(C_T)$, the derived two values are lower to the previous studies where $N(C_T)$ varied from 8.6 to 2.0 mol m$^{-2}$ a$^{-1}$. $N(C_T;$ 100 m) was negative because the deep integration depth included water below the euphotic zone where the remineralisation of organic matter can increase $C_T$. This signal is not present in $N(O_2)$ because the changes were largely controlled by $\Phi(O_2)$ that was always positive. The calculated $N(O_2)$ is in agreement with previous studies, which gave results between 2.6 and 11 mol m$^{-2}$ a$^{-1}$.

In the discussion, we added a section where we explain the location and the period when the previous studies took place. Falck and Anderson (2005) used historical data from 1960 to 2000 collected in the area from 62 to 70° N and from 1991 to 1994 collected at OWSM. Skjelvan et al (2001) used data collected from 67.5° N 9° E to 71.5° N 1° E and along 74.5° N from 7 to 15° E from 1957 to 1970 and 1991 to 1998. Kivimäe (2007) used the oxygen measured at OWSM from 1955 to 2005 and Falck and Gade (1999) used data collected in all the Norwegian Sea from 1955 to 1988. The glider in the transect moved from to 66.3 °N 4 °W to 63 °N 4 °E (Figure 5 where the black dots are the glider dives, the green box the region used by Falck and Gade (1999), the yellow lines the transects used by Skjelvan et al. (2001), the azure box the region used by Falck and Anderson (2005) and the red dot the location that corresponds to the Ocean Weather Station M (OWSM) used by Kivimäe (2007)).

[Figure]

**Figure 5:** Map of the glider deployment showing the previous studies that estimated the net community production in the Norwegian Sea. The black dots are the glider dives, the green box the region used by Falck and Gade (1999), the yellow lines the transects used by Skjelvan *et al.* (2001), the azure box the region used by Falck and Anderson (2005) and the red dot the location that corresponds to the Ocean Weather Station M (OWSM) used by Kivimäe (2007).

We added in section 4.2, line 760-773: "The estimated $N$ in the 4 studies varies from 2.0 to 8.6 mol m$^{-2}$ a$^{-1}$ for $N(C_T)$ and from 2.6 to 11.1 mol m$^{-2}$ a$^{-1}$ for $N(O_2)$. In our study, we obtained an annual $N(C_T)$ of 0.9 mol m$^{-2}$ a$^{-1}$ and a $N(O_2)$ of 4 mol m$^{-2}$ a$^{-1}$ in agreement with these studies. The larger $N(O_2)$ compared with $N(C_T)$ should be attributed to the large $\Phi(O_2)$ that had an absolute median of 47 mmol m$^{-2}$ d$^{-1}$ compared with $\Phi(CO_2)$ absolute median of 1.7 mmol m$^{-2}$ d$^{-1}$. Instead, the inventory changes were similar for $N(O_2)$ and $N(C_T)$ had a median of 12 mmol m$^{-2}$ d$^{-1}$ and 7.6 mmol m$^{-2}$ d$^{-1}$, respectively. To compare our results with previous studies we used the same $z_{lim}$ of 30 m(Falck and Gade, 1999) and 100 m (Falck and Anderson, 2005; Kivimäe, 2007). The calculated $N(C_T; 30$ m) was 1 mol m$^{-2}$ a$^{-1}$, $N(C_T; 100$ m) was 0.4 mol m$^{-2}$ a$^{-1}$, $N(O_2; 30$ m) was 4 mol m$^{-2}$ a$^{-1}$ and $N(O_2; 100$ m) was 3.7 mol m$^{-2}$ a$^{-1}$. In the case of $N(C_T; 30$ m) and $N(C_T; 100$ m) the values calculated were smaller to the previous studies where $N(C_T)$ varied from 2 to 8.6 mol m$^{-2}$ a$^{-1}$. The smallest value was for $N(C_T; 100$ m) because it included the not productive layer located under the euphotic zone and the $z_{mix}$ where the remineralisation of the organic matter can increase $C_T$. The calculated $N(O_2)$ was not affected by the selection of $z_{lim}$ because the changes are largely controlled by $\Phi(O_2)$. However, the calculated $N(O_2)$ was in agreement with the previous studies where varied from 2.6 to 11 mol m$^{-2}$ a$^{-1}$. The annual $N(C_T)$ and $N(O_2)$ that we calculated are most likely an understimation because we set the net community production to 0 for the rest of the year."

And line 754-760: "All these studies used low-resolution datasets in space and time. These datasets had data collected by several cruises in different years, Falck and Anderson (2005) used historical data from 1960 to 2000 collected in the area from 62 to 70° N and from 1991 to 1994 collected all the year at OWSM. Skjelvan et al., (2001) used data collected from 67.5° N 9° E to 71.5° N 1° E and along 74.5° N from 7 to 15° E from 1957 to 1970 and from 1991 to 1998. Kivimäe (2007) used the oxygen measured all the year at OWSM from 1955 to 2005 and Falck and Gade (1999) used data collected all the year in all the Norwegian Sea from 1955 to 1988."

**Bibliography**

Alkire, M. B. *et al.* (2014) 'Net community production and export from Seaglider measurements in the North Atlantic after the spring bloom', *Journal of Geophysical Research: Oceans*. Wiley Online

Library, 119(9), pp. 6121–6139.

Atamanchuk, D. *et al.* (2014) 'Performance of a lifetime-based optode for measuring partial pressure of carbon dioxide in natural waters', *Limnology and Oceanography: Methods*, 12(2), pp. 63–73. doi: 10.4319/lom.2014.12.63.

Becker, S. *et al.* (2020) 'GO-SHIP repeat hydrography nutrient manual: the precise and accurate determination of dissolved inorganic nutrients in seawater, using continuous flow analysis methods', *Frontiers in Marine Science*. Frontiers, 7, p. 908.

Falck, E. and Anderson, L. G. (2005) 'The dynamics of the carbon cycle in the surface water of the Norwegian Sea', 94, pp. 43–53. doi: 10.1016/j.marchem.2004.08.009.

Falck, E. and Gade, G. (1999) 'Net community production and oxygen fluxes in the Nordic Seas based on O2 budget calculations', 13(4), pp. 1117–1126.

Foltz, G. R. *et al.* (2003) 'Seasonal mixed layer heat budget of the tropical Atlantic Ocean', *Journal of Geophysical Research: Oceans*. Wiley Online Library, 108(C5).

Gislefoss, J. S. *et al.* (1998) 'Carbon time series in the Norwegian sea', 45, pp. 433–460.

Gourcuff, C. (2014) 'ANFOG Slocum CTD data correction', (March).

Hagebo, M. and Rey, F. (1984) 'Storage of seawater for nutrients analysis', *Fisken Hav., 4, 1*, 12.

Van Heuven, S. *et al.* (2011) 'MATLAB program developed for CO2 system calculations', *ORNL/CDIAC-105b. Carbon Dioxide Information Analysis Center, Oak Ridge National Laboratory, US Department of Energy, Oak Ridge, Tennessee*, 530.

Kivimäe, C. (2007) 'Carbon and oxygen fluxes in the Barents and Norwegian Seas: production, air-sea exchange and budget calculations'. The University of Bergen.

Miloshevich, L. (2004) 'Development and Validation of a Time-Lag Correction for Vaisala Radiosonde Humidity Measurements', pp. 1305–1328.

Naveira Garabato, A. C. *et al.* (2004) 'Turbulent diapycnal mixing in the Nordic seas', *Journal of Geophysical Research: Oceans*. Wiley Online Library, 109(C12).

Obata, A., Ishizaka, J. and Endoh, M. (1996) 'Global verification of critical depth theory for phytoplankton bloom with climatological in situ temperature and satellite ocean color data', *Journal of Geophysical Research: Oceans*. Wiley Online Library, 101(C9), pp. 20657–20667.

Reuer, M. K. *et al.* (2007) 'New estimates of Southern Ocean biological production rates from O2/Ar ratios and the triple isotope composition of O2', *Deep Sea Research Part I: Oceanographic Research Papers*. Elsevier, 54(6), pp. 951–974.

Skjelvan, I., Falck, E., Anderson, Leif G, *et al.* (2001) 'Oxygen fluxes in the Norwegian Atlantic current', *Marine chemistry*. Elsevier, 73(3–4), pp. 291–303.

Skjelvan, I., Falck, E., Anderson, Leif G., *et al.* (2001) 'Oxygen fluxes in the Norwegian Atlantic Current', *Marine Chemistry*, 73(3–4), pp. 291–303. doi: 10.1016/S0304-4203(00)00112-2.

United States. National Environmental Satellite and Information Service, D., Monterey, G. I. and Levitus, S. (1997) *Seasonal variability of mixed layer depth for the world ocean*. US Department of Commerce, National Oceanic and Atmospheric Administration ….

**Response to Reviewer 2**

We would like to thank Reviewer 2 for providing constructive and insightful comments. We incorporated their suggestions into a revised manuscript. Reviewer 2's comments have been reproduced below in black, with the authors' response in blue.

**General comments**

However, there are some short paragraphs that contain only one sentence. I suggest the authors consider re-organize some of the paragraphs.

To solve this problem we merged section 2.5 and 2.6 that now is called: "$CO_2$ optode lag and drift correction and calibration" and section 3.1 and 3.2 than now is: "$O_2$ and $CO_2$ optode calibration"

My major concern is about oxygen optode calibration. It is unfortunate that discrete oxygen samples were not collected. But I am not convinced that archived oxygen data dated back to 2000 are suitable to be used for calibration even for deep water. The authors may justify this by demonstrating that the changes in archived oxygen data over the past 20 years are minor. Otherwise, the most recent discrete oxygen data should be used for calibration.

In the revised version we added in the appendices Figure 1, the figure shows that the oxygen discrete samples variability is within the variability of the measured oxygen by the glider. For that reason, the interannual and seasonal variability of the discrete samples can be considered minimal. See also reply 9 to Reviewer 1.

[Figure]

**Figure 1:** a) Discrete samples $c_C(O_2)$ (yellow), raw glider oxygen $c_G(O_2)$ (blue) and drift corrected glider oxygen $c_{G,cal}(O_2)$ (red) using water density > 1028 kg m$^{-3}$.

Also, Figure 2 shows that the oxygen discrete samples in this water mass do not change between 2000 and 2010. For example, $c(O_2)$ in 2000 varied from 299.5 to 314.3 µmol kg$^{-1}$ and in 2009 from 300.6 to 312.7 µmol kg$^{-1}$. For that reason, we can consider the oxygen concentration constant during the years and can be used as reference to correct the oxygen optode.

[Figure]

**Figure 2:** oxygen discrete samples used as reference to calibrate the oxygen optode output. All the samples were collected in the latitude and longitude range of the deployment area for a water density > 1028 kg m$^{-3}$.

One major advantage of glider is that it can survey the entire water column continuously. However, the major portion (sections 3.3-3.6) of the results section is on NCP data at an integration depth of zlim = 45 m (figures 14-16). This compromises the importance of using glider data.

In all the manuscript we correct and show the entire profiles for oxygen, $CO_2$, temperature, salinity and chlorophyll. The net community production was calculated using an integration depth of 45 m because it was the mean depth of the euphotic zone. The two $N$s were 4.6 and 0.5 mol m$^{-2}$ a$^{-1}$ for $N(O_2)$ and $N(C_T)$, respectively. For comparison, we calculated net community production using integration depths of 30 and 100 m. The derived net community production was the same for $N(O_2)$ at the different integration depths and similar for $N(C_T)$. In particular, $N(C_T; 30$ m$)$ was 1 mol m$^{-2}$ a$^{-1}$ and $N(C_T; 100$ m$)$ was 0.4 mol m$^{-2}$ a$^{-1}$; $N(O_2; 30$ m$)$ was 4 mol m$^{-2}$ a$^{-1}$ and $N(O_2; 100$ m$)$ was 3.7 mol m$^{-2}$ a$^{-1}$.

We added in section 4.2, line 760-773: "The estimated $N$ in the 4 studies varies from 2.0 to 8.6 mol m$^{-2}$ a$^{-1}$ for $N(C_T)$ and from 2.6 to 11.1 mol m$^{-2}$ a$^{-1}$ for $N(O_2)$. In our study, we obtained an annual $N(C_T)$ of 0.9 mol m$^{-2}$ a$^{-1}$ and a $N(O_2)$ of 4 mol m$^{-2}$ a$^{-1}$ in agreement with these studies. The larger $N(O_2)$ compared with $N(C_T)$ should be attributed to the large $\Phi(O_2)$ that had an absolute median of 47 mmol m$^{-2}$ d$^{-1}$ compared with $\Phi(CO_2)$ absolute median of 1.7 mmol m$^{-2}$ d$^{-1}$. Instead, the inventory changes were similar for $N(O_2)$ and $N(C_T)$ had a median of 12 mmol m$^{-2}$ d$^{-1}$ and 7.6 mmol m$^{-2}$ d$^{-1}$, respectively.

To compare our results with previous studies we used the same $z_{\text{lim}}$ of 30 m(Falck and Gade, 1999) and 100 m (Falck and Anderson, 2005; Kivimäe, 2007). The calculated $N(C_T; 30 \text{ m})$ was 1 mol m$^{-2}$ a$^{-1}$, $N(C_T; 100 \text{ m})$ was 0.4 mol m$^{-2}$ a$^{-1}$, $N(O_2; 30 \text{ m})$ was 4 mol m$^{-2}$ a$^{-1}$ and $N(O_2; 100 \text{ m})$ was 3.7 mol m$^{-2}$ a$^{-1}$. In the case of $N(C_T; 30 \text{ m})$ and $N(C_T; 100 \text{ m})$ the values calculated were smaller to the previous studies where $N(C_T)$ varied from 2 to 8.6 mol m$^{-2}$ a$^{-1}$. The smallest value was for $N(C_T; 100 \text{ m})$ because it included the not productive layer located under the euphotic zone and the $z_{mix}$ where the remineralisation of the organic matter can increase $C_T$. The calculated $N(O_2)$ was not affected by the selection of $z_{\text{lim}}$ because the changes are largely controlled by $\Phi(O_2)$. However, the calculated $N(O_2)$ was in agreement with the previous studies where varied from 2.6 to 11 mol m$^{-2}$ a$^{-1}$. The annual $N(C_T)$ and $N(O_2)$ that we calculated are most likely an understimation because we set the net community production to 0 for the rest of the year."

**Specific comments**

Lines 68-79, I think these two paragraphs belong to the method section.

We changed the these lines to follow the reviewer comment.

Line 348, it should be k(CO2) rather than k(O2) in equation 14.

We changed the equation replacing $k(O_2)$ with $k(CO_2)$.

Line 609, change "a sink to" to "a sink of"

We changed "a sink to" to "a sink of".

Figures 2, 3, 8, 9 Date on the x-axis is kind of misleading. It seems like Jan-04, Jan-05, etc. I think it is better to change 01/04, 01/05, . . ., 01/10 to April, May, . . ., October.

We changed the date to the suggested date format in figures: 2, 3 and 6 to 16.

---

## Author Response (AR2)

**Response to the Reviewer**

We would like to thank the Reviewer for providing constructive and insightful comments. We incorporated the suggestions into a revised manuscript. Reviewer 1's comments have been reproduced below in black, with the authors' response in blue.

I might recommend the authors include a key at the beginning of the paper that lists all the abbreviations and their definitions. There is a lot to keep track of here.

We added a list of symbols at beginning of the Methods section:

| 2.1 List of symbols (unit)                                   |  |  |  |  |  |  |
|--------------------------------------------------------------|--|--|--|--|--|--|
| total alkalinity (µmol kg -1 )                    |  |  |  |  |  |  |
| backscatter signal (engineering units)                       |  |  |  |  |  |  |
| amount content ( $\mu$ mol kg -1 )                |  |  |  |  |  |  |
| amount concentration (mmol m -3 )                 |  |  |  |  |  |  |
| chlorophyll a                                                |  |  |  |  |  |  |
| dissolved inorganic carbon                                   |  |  |  |  |  |  |
| entrainment flux (mmol m -2 d -1 )     |  |  |  |  |  |  |
| diapycnal eddy diffusion flux (mmol $m^{-2} d^{-1}$ )        |  |  |  |  |  |  |
| fugacity of CO 2 (µatm)                           |  |  |  |  |  |  |
| inventory (mmol m -2 )                            |  |  |  |  |  |  |
| diapycnal eddy diffusivity (m 2 s -1 ) |  |  |  |  |  |  |
| net community production (mmol $m^{-2} d^{-1}$ )             |  |  |  |  |  |  |
| partial pressure of CO 2 (µatm)                   |  |  |  |  |  |  |
| practical salinity (1)                                       |  |  |  |  |  |  |
| time (s)                                                     |  |  |  |  |  |  |
| wind speed (m $s^{-1}$ )                                     |  |  |  |  |  |  |
| dry mole fraction (mol mol -1 )                   |  |  |  |  |  |  |
| depth of the deep chlorophyll maximum (m)                    |  |  |  |  |  |  |
| inventory integration depth (m)                              |  |  |  |  |  |  |
| mixed layer depth (m)                                        |  |  |  |  |  |  |
| air-sea flux (mmol $m^{-2} d^{-1}$ )                         |  |  |  |  |  |  |
| CO 2 optode CalPhase (°)                          |  |  |  |  |  |  |
| potential density (kg m -3 )                      |  |  |  |  |  |  |
| Celsius temperature (°C)                                     |  |  |  |  |  |  |
| response time (s)                                            |  |  |  |  |  |  |
|                                                              |  |  |  |  |  |  |

Line 375: Shouldn't the units for eddy diffusivity be m2 s-1? We corrected the units to  $m^2 s^{-1}$  here and also in the caption of Figure A2.

Figure 11: Discrete samples (red dots) plotted after 01/06 agree with SOCAT data from 14/06/14 but both these measurements are lower than the Seaglider fCO2 by ~90 uatm. Is the result of a difference in where the "discrete samples" were collected (Seaglider transect vs. SOCAT transect)? Please clarify. We changed to a new calibration method including temperature to better match the SOCAT data. In the previous manuscript, the CO2 optode signal ( $p_c(CO_2)$ ) was calibrated just using a regression against the CO2 concentrations of the discrete water samples ( $c_{WS}(CO_2)$ ). Plotting the regression residuals ( $c_r(CO_2)$ , calculated as  $c_{WS}(CO_2)$  minus the value predicted by the regression) revealed a quadratic correlation between the regression residuals and water temperature ( $\theta$ ) (Figure R1).

Figure R1: Residuals of the regression  $c_{WS}(CO_2)$  versus  $p_c(CO_2)$  plotted against water temperature ( $\theta$ ).

We have therefore included  $\theta$  and  $\theta^2$  in the optode calibration (revised Figure 6a below). This new calibration increased the correlation coefficient  $R^2$  from 0.77 to 0.90 and decreased the standard deviation of the regression residuals from 1.3 to 0.8 µmol kg-1. Even with this calibration, the CO2 optode response remained more closely related to  $c(CO_2)$  than  $p(CO_2)$  (Figure 6b).

**Figure 6:** Regression (black lines, reg1) of the CO2 optode output  $p_c(CO_2)$  against a) co-located concentration  $c_{WS}(CO_2)$  that has an uncertainty of 0.28 µmol kg-1 b) and partial pressure  $p_{WS}(CO_2)$  of CO2 in discrete water samples (black dots). Also shown are the values predicted by including  $\theta$  and  $\theta^2$  in the regression used for optode calibration (red dots, reg2). The regression equations are: a) reg1:  $c_{WS}(CO_2)/(\mu mol kg^{-1}) = (0.033\pm 0.003)p_c(CO_2)/\mu atm - 1.8\pm 1.6$  ( $R^2 = 0.77$ )

a) reg2:  $c_{WS}(CO_2)/(\mu mol \text{ kg}^{-1}) = (0.12\pm0.14)\theta/^{\circ}C - (0.071\pm0.011)(\theta/^{\circ}C)^2 + (0.0094\pm0.0048)p_c(CO_2)/\mu atm + 16\pm4 (R^2 = 0.90).$

b) reg1:  $p_{WS}(CO_2)/\mu atm = (0.05\pm0.05)p_c(CO_2)/\mu atm + 344\pm33 \ (R^2 = 0.02)$

b) reg2:  $p_{WS}(CO_2)/\mu atm] = (21\pm3)\theta/^{\circ}C - (1.9\pm0.2)(\theta/^{\circ}C)^2 + (0.2\pm0.1)p_{c}(CO_2)/\mu atm + 209\pm76 \ (R^2 = 0.60)."$

With the new calibration, our estimate of CO2 optode-derived fugacity  $f_t(CO_2)$  follows SOCAT  $f_{SOCAT}(CO_2)$  much better, including the decrease of  $f(CO_2)$  in the middle of June from 380 to 300 µatm (Figure 11).

---

## Author Response (AR3)

**Response to the Editor**

We would like to thank the Editor for his guidance and further constructive comments. We incorporated his suggestions into the manuscript. We have also looked into the suggestion whether salinity S could be used to improve the CO2 optode calibration further.

To recapitulate, we calibrated the CO2 optode output,  $p_c(CO_2)$ , against the CO2 concentrations measurements derived from discrete water samples,  $c_{WS}(CO_2)$ , using a quadratic parameterisation in terms of temperature  $\theta$ :

 $c_{reg2}(CO_2)/(\mu mol kg^{-1}) = (0.12\pm0.14)\theta/^{\circ}C - (0.071\pm0.011)(\theta/^{\circ}C)^2 + (0.0094\pm0.0048)p_c(CO_2)/\mu atm + 16\pm4$

This regression gave a mean residual of 0.8  $\mu$ mol kg-1.

As per the Editor's suggestion, we attempted a third regression, adding a quadratic parameterisation in terms of salinity. This gave the following fit:

 $c_{\text{reg3}}(\text{CO}_2)/(\mu\text{mol kg}^{-1}) = (0.10\pm0.11)\theta/^{\circ}\text{C} - (0.088\pm0.009)(\theta/^{\circ}\text{C})^2 + (4.2\pm1.0)(S-35) + (6.2\pm1.3)(S-35)^2 + (0.0084\pm0.0034)p_c(\text{CO}_2)/\mu\text{atm} + 17\pm3$

This regression gave a reduced mean residual of 0.6 µmol kg-1.

However, as Figure A below shows, our parameterisation is not well constrained in terms of salinity (*S*) outside the range of the salinities of the discrete water samples. Only one discrete sample had S

**Figure A:** Comparison between surface  $f(CO_2)$  from 2014 SOCAT and CO2 optode on the glider. Top panel:  $f_t(CO_2)$  corresponds to  $c_{reg2}(CO_2)$ ; and  $f_s(CO_2)$  to  $c_{reg3}(CO_2)$ , which includes salinity in the parameterisation. Bottom panel: Glider surface salinity.